# CLIPGaussian: Universal and Multimodal Style Transfer Based on Gaussian Splatting

**Kornel Howil**[1,3*]  **Joanna Waczyńska**[1,2*]  **Piotr Borycki**[1,2]  **Tadeusz Dziarmaga**[1]

**Marcin Mazur**[1]  **Przemysław Spurek**[1,3]

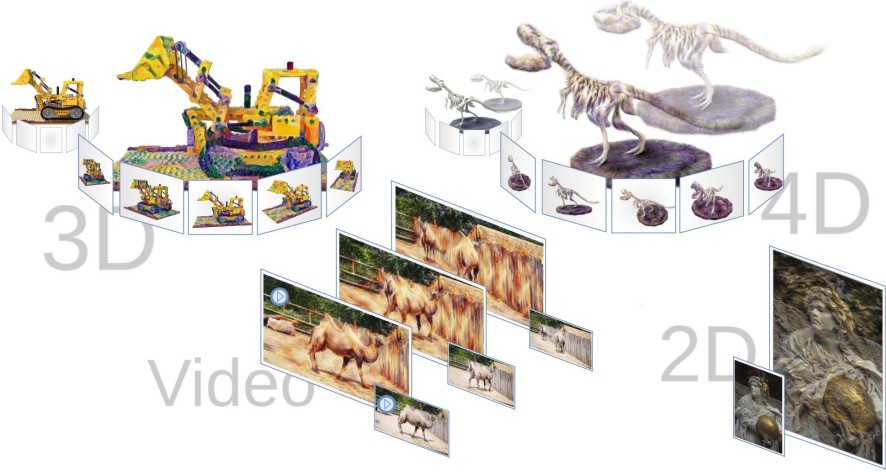

Figure 1: We present CLIPGaussian, a universal model for style transfer that supports a wide range of data modalities, including images, videos, 3D objects, and 4D dynamic scenes. Style transfer in CLIPGaussian can be guided using an image or a text prompt. Our method leverages a Gaussian Splatting representation to model both color and geometric aspects of style transfer.

## Abstract

Gaussian Splatting (GS) has recently emerged as an efficient representation for rendering 3D scenes from 2D images and has been extended to images, videos, and dynamic 4D content. However, applying style transfer to GS-based representations, especially beyond simple color changes, remains challenging. In this work, we introduce CLIPGaussian, the first unified style transfer framework that supports text- and image-guided stylization across multiple modalities: 2D images, videos, 3D objects, and 4D scenes. Our method operates directly on Gaussian primitives and integrates into existing GS pipelines as a plug-in module, without requiring large generative models or retraining from scratch. The CLIPGaussian approach enables joint optimization of color and geometry in 3D and 4D settings, and achieves temporal coherence in videos, while preserving the model size. We demonstrate superior style fidelity and consistency across all tasks, validating CLIPGaussian as a universal and efficient solution for multimodal style transfer.

---

[*]Equal contribution

[1]Jagiellonian University, Faculty of Mathematics and Computer Science; [2]Jagiellonian University, Doctoral School of Exact and Natural Sciences; [3]IDEAS Research Institute.

Correspondence to: Kornel Howil `<kornel.howil@student.uj.edu.pl>`, Przemysław Spurek `<przemyslaw.spurek@uj.edu.pl>`

Project page: `https://kornelhowil.github.io/CLIPGaussian/`

39th Conference on Neural Information Processing Systems (NeurIPS 2025).

# 1 Introduction

The past year has seen an explosive rise in user-driven content generation, particularly in the visual domain. Following the launch of OpenAI's GPT-4o [1], users created more than 700 million images within a single week, highlighting the massive demand for generative tools that allow intuitive, direct manipulation of visual content. Although 2D image generation and edition are rapidly becoming mainstream tasks, editing in higher dimensions, such as video, 3D, and 4D content, is significantly more complex [2]. These domains bring challenges in consistency, geometry, temporal coherence, and user control that existing systems are not yet fully equipped to handle.

Gaussian Splatting (GS) [3] is a major advancement in computer graphics, representing 3D scenes as sets of Gaussian components with color and opacity. Its training and rendering are highly efficient and produce realistic images. GS has also been adapted for 4D dynamic scenes [4, 5, 6], 2D images [7, 8], and videos [9, 10], using slightly modified Gaussian components to model 2D content.

One of the ways to edit objects represented by 3D Gaussian primitives is style transfer [11], which alters the global appearance of objects and scenes. A possible approach to address this issue is the use of plug-in-type models. This refers to a family of components that can be inserted into existing architectures without requiring complete retraining [12], allowing them to adapt to new tasks such as style transfer.

In this paper, we introduce CLIPGaussian, a plug-in type model suitable for Gaussian Splatting-based architectures. Our model stylizes content represented by Gaussian primitives, conditioned on either a reference image or a text prompt, across 2D, video, 3D, and 4D data, see Fig. 1. As a plug-in component, CLIPGaussian integrates into existing pipelines without requiring retraining of the base model.

When conditioned on a text prompt, models based on Gaussian Splatting primarily focus on editing rather than style transfer [13, 14, 15]. In the context of conditioning on a reference image, models typically concentrate on modifying only color and opacity. Examples of this include StyleGaussian [16], ReGS [17], InstantStyleGaussian [18], Style3D [19], and StyleSplat [20]. G-Style [21] employs a two-phase process, consisting of stylization followed by refining the scene's geometry, but increasing the size of the model significantly. Since our CLIPGaussian uses the architecture of the base model, we can optimize all parameters, without changing the model size. By operating directly within the architecture of the Gaussian Splatting-based model, our method enables end-to-end optimization of the full set of Gaussian parameters including position and scale, rather than being restricted to color-based edits, see Fig. 2. Crucially, we retain the original number of Gaussian primitives, thereby preserving the memory and computational characteristics of the object's reconstruction.

In this work, we make the following primary contributions:

- **Universal Multimodal Stylization**: We propose CLIPGaussian, the first plug-in style transfer model for Gaussian Splatting, enabling image- and text-guided stylization across 2D, video, 3D, and 4D data without retraining the base model.

- **Joint Appearance and Geometry Optimization**: Our method allows end-to-end optimization of all Gaussian parameters, enabling geometric transformations and temporally consistent results, while preserving the original model size.

- **Generalizable GS-based Stylization Framework**: We demonstrate that Gaussian Splatting is a versatile substrate for style transfer, achieving strong performance across tasks and modalities with a unified architecture.

# 2 Related Works

CLIPGaussian operates across diverse data modalities, including images, videos, and 3D or 4D scenes. To our knowledge, it is the first method capable of style transfer across such a wide range. As a result, direct comparison with existing work is challenging, so we evaluate our model separately for each modality.

**3D** Our model uses Gaussian Splatting to represent various data modalities. Therefore, the closest solutions are dedicated to 3D scenes. In the case of classical Gaussian Splatting, style transfer methods usually work only on colors [16, 17, 18, 19, 20]. It means that such algorithms do not change the geometry of objects. Therefore, the only modification is to the colors and opacities of Gaussian

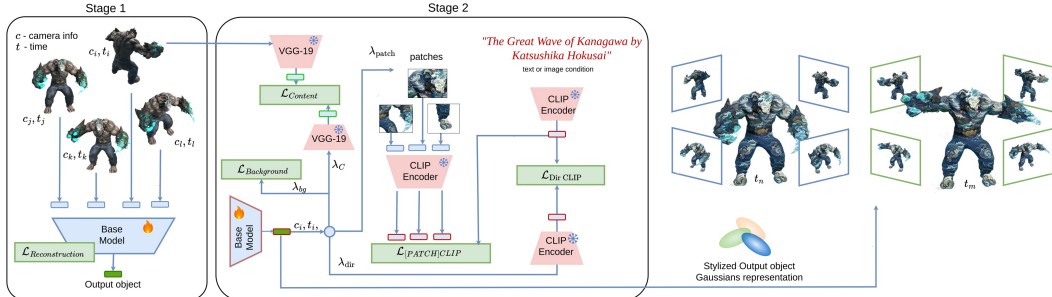

Figure 2: CLIPGaussian architecture in the case of a 4D dynamic scene. The method operates in two main stages. In the first stage, we train a Gaussian Splatting model tailored to a specific data modality. In the second stage, during training, we leverage training images, randomly sampled patches, and conditioning inputs (either an image or a text) in the feature spaces of VGG-19 and CLIP models. We optimize the Gaussian parameters using a composite loss function with four terms: content preservation, background preservation, local style transfer, and global style transfer. Notably, CLIPGaussian integrates with GS-based systems as a plug-in module.

primitives. StyleGaussian [16] embeds 2D VGG features into 3D Gaussians, transforms them based on a style image, and decodes them with a novel KNN-based 3D CNN. Such a model can be seen as an adaptation of AdaIN [22] to 3D Gaussian Splatting. StyleSplat [20] segments individual objects, then fine-tunes their appearance using feature matching with a style image, allowing customizable, multi-object stylization with strong visual consistency and efficiency. ReGS [17] introduces a texture-guided control mechanism for fine-grained appearance editing. It enhances style detail by replacing selected Gaussians with denser ones, guided by texture cues and regularized by depth to preserve geometry. G-Style [21] is a 3D scene stylization algorithm that enhances Gaussian Splatting by optimizing both appearance and geometry to match a reference style image. It improves visual quality by preprocessing out problematic Gaussians, applying multi-scale style losses, and refining geometry through gradient-based Gaussian Splatting.

Alternatively, we can use a large diffusion model for the style transfer of 3D models. InstantStyleGaussian [18] uses diffusion models and an iterative dataset update strategy. It stylizes pre-reconstructed scenes by generating target-style images, updating the training data, and optimizing the scene efficiently, achieving high-quality results with improved speed and consistency. Style3D [19] introduces MultiFusion Attention to align structural and stylistic features across views, ensuring spatial coherence and visual fidelity, and uses a large 3D reconstruction model for high-quality, efficient stylization. Morpheus [23] introduces a novel autoregressive 3D Gaussian Splatting stylization method that enables controllable stylization of both appearance and geometry in 3D scenes. Instruct-GS2GS (I-GS2GS) [13] performs style transfer by applying a diffusion-based model to each of the input images used to train a Gaussian Splatting. Each splat is then fine-tuned using these stylized images. In contrast CLIPGaussian instead of stylizing the training images, directly optimizes the Gaussian representation using losses derived from CLIP embeddings. This approach enables high-quality style transfer and supports conditioning on both images and text prompts, whereas I-GS2GS supports only text-based conditioning. Importantly, CLIPGaussian does not rely on any pretrained diffusion models for image stylization; the entire process is guided solely by CLIP-based objectives. The key contribution is an RGBD diffusion model allowing users to adjust stylization strength over shape and look. Such task is closer to 3D scene editing than style transfer like in DGE [14], GaussCltr [15], ProGDF [24] or EditSplat [25].

**4D**    One of the most underexplored modalities is the 4D scene stylization which aims to modify the appearance of the 3D scene over time. Existing approaches remain limited. Currently, to our knowledge there are only just a few 4D style transfer models.

4DStyleGaussian [26] uses a reversible neural network to train 4D embedded Gaussians, preserving content fidelity while reducing feature distillation artifacts. A learned 4D style transformation matrix enables consistent stylization across views and time. Instruct 4D-to-4D [27] is a NeRF-based method for instruction-guided editing of dynamic scenes. It enhances 2D diffusion models with 4D awareness and spatial-temporal consistency. Treating 4D scenes as pseudo-3D combines anchor-aware attention, optical flow-guided appearance propagation, and depth-based projection to enable consistent, high-quality edits across time and viewpoints. While both methods focus on scenes, we show that our

method also works very well on objects without backgrounds, see Fig. 3. StyleDyRF [28] is also a NeRF-based method. The Canonical Feature Volume and Canonical Style Transformation are introduced to create a compact representation of the 4D scene, and learn the linear transformation to reflect the reference style respectively. As opposed to our method, it supports only image based style conditioning.

**Images** In contrast to 4D scenes, 2D image style transfer is a very popular task. The foundational work in neural style transfer [29] introduced an optimization-based method to match the content features of one image and the style captured using Gram matrices of another, using a pre-trained convolutional neural model. This demonstrated that deep features could disentangle content and style, but the approach was computationally expensive due to its iterative nature. To address speed limitations, feed-forward networks were introduced to enable real-time stylization by applying a fixed style in a single pass [30, 31]. Arbitrary style transfer was later made possible by aligning feature statistics using Adaptive Instance Normalization [22], further refined through Whitening and Coloring Transform [32]. More recent methods have explored more flexible conditioning mechanisms. For image-guided style, transformer-based architectures like StyTr$^2$ leverage both local and global context to improve stylization quality and content preservation [33]. Alternatively, text prompts have emerged as an intuitive way to specify style. Style transfer guided by CLIP embeddings enables stylization from a textual description [34], while more efficient approaches learn lightweight style representations for feed-forward transfer [35].

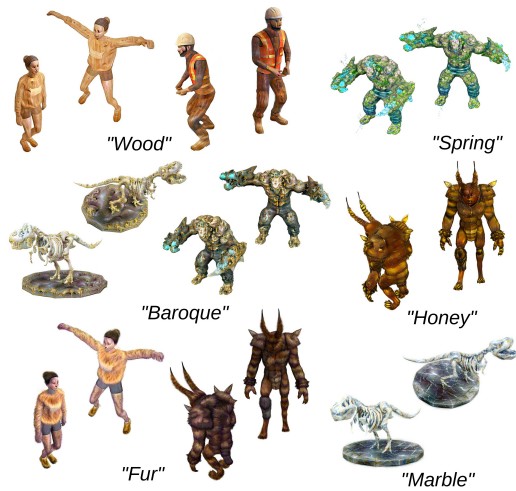

Figure 3: Results of text-based 4D style transfer. Our model modifies both the color and geometry of Gaussian primitives.

**Videos** The majority of image-style transfer methods [36, 37, 38] are employed for video-style transfer. Linear [36] proposes a fast and flexible style transfer method using a learned transformation matrix. It replaces costly or handcrafted operations with a feed-forward model. CCPL [37] uses a novel Contrastive Coherence Preserving Loss to reduce local inconsistencies, maintaining temporal coherence without harming style quality. UniVST [38] is a unified, training-free video style transfer framework based on diffusion models that focuses on localized stylization. It uses a point-matching mask propagation strategy to avoid the need for tracking models. AdaAttN [39] introduces Adaptive Attention Normalization, which addresses the problem of local distortions by performing attentive normalization on a per-point basis by jointly considering shallow and deep features of content and style images. ReReVST [40] introduces a zero-shot video style transfer framework with a novel relaxation of the style loss and a new temporal regularization. ViSt3D [41] introduces the first video style transfer method to use 3D CNNs directly, which works by disentangling motion and appearance features to apply the style only to the appearance before re-adding the motion.

Alternatively, we can use large generative models for style transfer on videos [42, 43, 44]. In [45], the authors adapt large text-to-image diffusion models for video generation, addressing the challenge of temporal consistency. The framework consists of two parts: key frame translation and full video translation. Style-a-video [46] is a zero-shot video stylization method that uses a pre-trained image latent diffusion model and a generative transformer for text-guided style transfer. UniST [47] introduces the Domain Interaction Transformer (DIT), which enables cross-domain learning by sharing contextual information between images and videos. StyleMaster [48] improves style consistency and temporal coherence by filtering content-related patches while retaining style ones. The method also supports image-to-video stylization by employing a lightweight motion adapter trained on still videos.

Each of the above approaches has issues with optical flow, so a dedicated mechanism is necessary to create smooth style transitions between frames. Our method works directly on Gaussian components, which inherently addresses these problems.

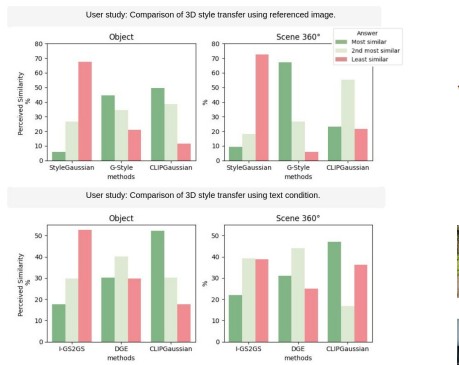

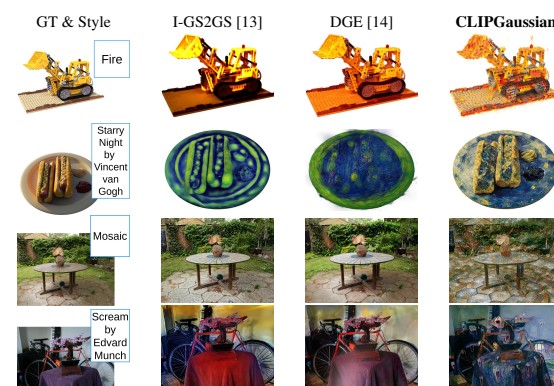

Figure 4: We conduct a user study, comparing our model against baseline methods. CLIPGaussian achieves scores comparable to G-Style with image conditioning and outperforms all models when using text prompts.

Figure 5: Comparison of various 3D style transfer methods involving text conditioning. CLIPGaussian applies style transfer with more significant shape changes. Our model captures details by attending to local regions through patches processing.

# 3 CLIPGaussian: Universal and Multimodal Style Transfer Based on Gaussian Splatting

This section introduces our CLIPGaussian model, which is designed for style transfer across different data modalities: images, videos, and 3D or 4D scenes. CLIPGaussian is a universal method that works as a *plug-in* for Gaussian Splatting-based representations. The core idea is to fine-tune the parameters of the Gaussian components using the CLIP model [49], which evaluates the similarity between natural language and images. As a result, our model is agnostic not only to the modality of data, but also to the conditioning mechanism, allowing the style to be transferred from either an image or a text.

The core of our approach is to represent all data modalities using Gaussian components. This unified representation facilitates style transfer across images, videos, and 3D or 4D scenes by modifying the corresponding Gaussian primitives. Although each modality introduces specific variations— such as temporal embeddings for videos and 4D scenes—our framework supports a single, general architecture for style transfer.

**Training Dataset**    The input data for CLIPGaussian are similar to those of typical style transfer models, but they vary slightly depending on the specific task. When applying a style to an image, the input is simply a single image. For videos, the input consists of all frames, which can be treated as a sequence of 2D images. For 3D scenes, views from different camera positions are used. Finally, for 4D objects, temporal indices for these views are also included. Despite the mentioned differences, we can unify our framework by assuming that the training dataset consists of a set of images $\mathcal{I} = \{I_1, \ldots, I_k\}$. Additionally, we have a style given by an image or a text prompt, which can be seen as a conditioning factor $\mathcal{S}$.

**Base Model: Gaussian Splatting Representation of Various Data Modalities**    Our model can be seen as *plug-in* for Gaussian Splatting representations. For this purpose, we work on a GS-based general representation consisting of a set of Gaussians:

$$\mathcal{G} = \mathcal{G}_{m_i, \Sigma_i, \sigma_i, c_i, \theta_i} = \left\{ (\mathcal{N}(m_i, \Sigma_i), \sigma_i, c_i, \theta_i) \right\}_{i=1}^{n}, \tag{1}$$

where $m_i$ are the mean positions, $\Sigma_i$ are the covariance matrices, $\sigma_i$ are the opacities, and $c_i$ are the Spherical Harmonics (SH) colors of the Gaussian components—these are standard GS parameters. On the other hand, $\theta_i$ are additional parameters dedicated to specific data modalities related to an applied base model, which can be any of the available GS-based architectures. In this paper, we use the classical 3DGS [3] for 3D scenes, D-MiSo [6] for 4D scenes, MiRaGe [6] for 2D images, and VeGaS [10] for videos. A detailed description of each case can be found in Appendix A.

The first stage of CLIPGaussian is to train an established base model on the set of input images $\mathcal{I}$, which boils down to optimizing all the parameters occurring in Eq. (1). Note that this varies depending on the chosen data modality. Once training is complete, the resulting pre-trained model

Table 1: Quantitative comparison of style transfer, compared against baseline methods.

| Model | CLIP-S ↑ | CLIP-SIM ↑ | CLIP-F↑ | CLIP-CONS ↑ | Memory size ↓ |
|---|---|---|---|---|---|
| *Text-conditioned* | | | | | |
| I-GS2GS [13] | 16.80 | 12.03 | 99.19 | **13.53** | **-36%** |
| DGE [14] | 17.59 | 12.27 | **99.31** | 12.46 | -5% |
| **CLIPGaussian-Light** | 23.14 | 22.30 | 99.17 | **8.51** | +0% |
| **CLIPGaussian** | **26.86** | **26.31** | 98.80 | 2.34 | +0% |
| *Image-conditioned* | | | | | |
| StyleGaussian [16] | 63.69 | 13.07 | 98.87 | 1.36 | **+0%** |
| SGSST [50] | 66.57 | 16.24 | 97.54 | 0.91 | **+0%** |
| ABC-GS [51] | 68.68 | 16.29 | 99.10 | 2.11 | **+0%** |
| G-Style [21] | **76.94** | **24.94** | 98.94 | 1.31 | +126% |
| **CLIPGaussian-Light** | 69.27 | 17.02 | **99.16** | **5.26** | **+0%** |
| **CLIPGaussian** | 72.65 | 20.72 | 98.78 | 1.77 | **+0%** |

becomes the input for the second stage of our algorithm, in which a final style transfer procedure is performed.

**CLIPGaussian: Final Style Transfer** In the subsequent stage, we implement style transfer by further fine-tuning the parameters of the base model (see Eq. (1)). The key idea behind CLIP-Gaussian is to work on whole objects and small patches at the same time. The training procedure begins by selecting a single image $I_l$ from the training dataset $\mathcal{I}$, which is then used along with its reconstruction $R_{\mathcal{G}}(I_l)$ produced by the base model $\mathcal{G}$, Then, a collection of random patches $\mathcal{P} = \{p_1(R_{\mathcal{G}}(I_l)), \ldots, p_m(R_{\mathcal{G}}(I_l))\}$ from the rendered output $R_{\mathcal{G}}(I_l)$ is extracted, to which random perspective augmentations are further applied (these cropped patches maintain fixed dimensions). Eventually, the parameters of $\mathcal{G}$ are updated according to the multi-component loss function, which is elaborated in the following paragraph, and the procedure is repeated from the beginning. It is important to note that CLIPGaussian does not perform densification or alter the number of Gaussian components. As a result, the stylized objects retain the same size as the original. Moreover, this property enables style interpolation by linearly interpolating the parameters of each Gaussian component.

**CLIPGaussian: Loss Function** Our loss function contains several components, which are detailed below. However, it should be noted that our approach relies on two pre-trained architectures: CLIP and VGG-19, which we further denote as $\Phi_{\text{CLIP}}$ and $\Phi_{\text{VGG}}$, respectively. We use the CLIP model for style transfer and the VGG-19 model to ensure that the stylized output resembles the original object.

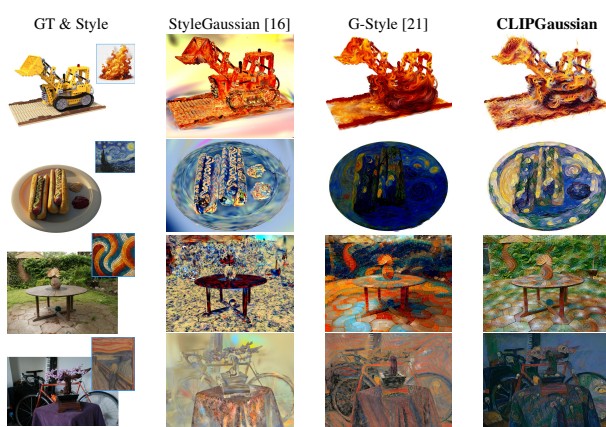

Figure 6: Comparison of 3D style transfer obtained by image conditioning. Results generated by CLIPGaussian are more detailed.

*Content loss* ($L_c$) measures the similarity between input views and elements after style transfer. To maintain the content information of the input object, following [29], we calculate $L_c$ as the mean squared error (MSE) between the `conv4_2` and `conv5_2` features of the original image $I_l$ and rendered image $R_{\mathcal{G}}(I_l)$, extracted from a pre-trained VGG-19 model, i.e.,

$$L_c(R_{\mathcal{G}}(I_l), I_l) = MSE(\Phi_{VGG}(R_{\mathcal{G}}(I_l)), \Phi_{VGG}(I_l)). \qquad (2)$$

*Directional CLIP loss* ($L_d$) is responsible for global style transfer. Similar to [52], we define it as follows:

$$L_d(R_{\mathcal{G}}(I_l), I_l) = 1 - \cos(\Phi_{CLIP}(R_{\mathcal{G}}(I_l)) - \Phi_{CLIP}(I_l), \Phi_{CLIP}(\mathcal{S}) - \Phi_{CLIP}(\text{"Photo"})). \qquad (3)$$

It should be noted that in this case, CLIP embeddings of the original image $I_l$ and the reconstructed image $R_{\mathcal{G}}(I_l)$, in conjunction with outputs produced by the CLIP model for a given style factor $S$ (either a style image or a style prompt) and a simple universal negative prompt "Photo", are utilized. Both negative and positive text prompts are combined with an ImageNet prompt template [53].

*Patch CLIP loss* ($L_p$) is responsible for local style transfer. It is defined as follows:

$$L_p(R_{\mathcal{G}}(I_l), I_l) = \frac{1}{n} \sum_{i=1}^{n} L_d(p_i(R_{\mathcal{G}}(I_l)), I_l). \tag{4}$$

However, we emphasize that, as proposed in [34], for all patches $p_i(R_{\mathcal{G}}(I_l)) \in \mathcal{P}$ we apply random perspective augmentations before calculating the CLIP directional loss.

*Background loss* ($L_b$) is assigned to fixed background elements. For 3D objects and videos, a mask is typically used to distinguish between the foreground and background. We incorporate background loss to prevent the style transfer process from introducing unwanted background artifacts. Therefore, $L_b$ is defined as the mean L1 distance between the designated background color and the pixel values corresponding to background regions in the rendered image.

*Total loss* ($L_{total}$), which is used to fine-tune the parameters of the base model, is defined as a weighted sum of the multiple loss components described above. This leads to the following formula:

$$L_{total} = \lambda_d L_d + \lambda_p L_p + \lambda_c L_c + \lambda_b L_b, \tag{5}$$

where $\lambda_d$, $\lambda_p$, $\lambda_c$, and $\lambda_b$ are empirically chosen weighting parameters. Details of the ablation study are provided in Appendix B. For $L_p$ and $L_d$ we use `ViT-B/32` CLIP model [53].

## 4  Experiments

The experiments section is divided into four parts, each corresponding to a specific modality for which the model has been applied. We note that because CLIPGaussian operates across multiple data modalities, baselines differ by task. Within each task, we standardize the setup for fair comparisons. For comparison, we include two versions of our method which differ in hyperparameters. CLIPGaussian with standard parameters and CLIPGaussian-Light which produces lighter stylization. Details of hyperparameters can be found in Appendix A. Additional experiments using other datasets and ablation studies are presented in Appendix B. For all 2D, 3D, Videos, 4D-objects experiments, we used the NVIDIA RTX 4090 GPU, For 4D-scenes we used NVIDIA DGX A100 GPU. The code is available on GitHub [2].

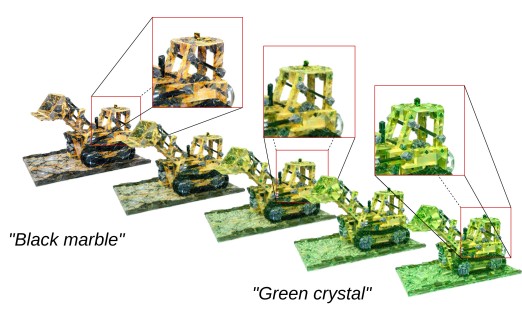

"Black marble"

"Green crystal"

Figure 7: Style interpolation between "Black marble" and "Green crystal" styles on the *lego* object.

**3D** As our approach is designed as a plug-in model compatible with Gaussian Splatting-based models, we begin our evaluation using the standard, vanilla Gaussian Splatting framework. The experiments evaluate both 3D object-level and scene-level performance. The NeRF-Synthetic dataset [55], comprising object-centric synthetic scenes with clean geometry and no background, enables controlled assessment of reconstruction and style transfer on isolated objects. Mip-NeRF 360 [56] provides photorealistic 360-degree real-world scenes with wide baselines, occlusions, and varying depth scales.

Style    4DStyleGaussian [26]    **CLIPGaussian**

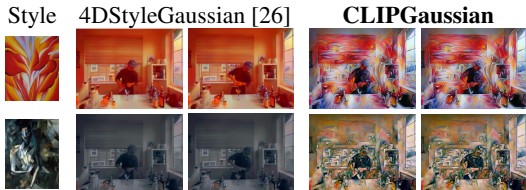

Figure 8: Comparative analysis of style transfer using referenced style image of novel views across various time frames and styles on the Neural 3D Video dataset (DyNeRF) [54].

---

[2] https://github.com/kornelhowil/CLIPGaussian

To assess performance on both image- and text-based stylization, we employ CLIP-based metrics. CLIP Directional Similarity [52] (CLIP-SIM) and CLIP-S [57] evaluate transfer quality, while CLIP Directional Consistency [58] (CLIP-CONS) and CLIP-F [59] assess temporal consistency and content similarity. We also report changes in model size (number of Gaussians) after stylization. Metric definitions and detailed evaluation settings are provided in Appendices B.1.1 and B.2.

Tab. 1 compares our method with Instruct-GS2GS [13] and DGE [14] for text-conditioned style transfer and StyleGaussian [16], SGSST [50], ABC-GS [51] and G-Style [21] for image-conditioned style transfer. Experiments use two objects (*lego*, *hotdog*) and two scenes (*garden*, *bonsai*), each evaluated under both image- and text-driven conditions. CLIPGaussian method achieves state-of-the-art results under text-guided conditions. It significantly outperforms all baselines with respect to the CLIP-S and CLIP-SIM metrics. We also achieve high-quality results under image-guided conditions. Compared to G-Style, our method offers a balance between performance and efficiency. This makes CLIPGaussian a practical and scalable alternative for style transfer tasks.

For selected baseline methods (Instruct-GS2GS, DGE, StyleGaussian and G-Style) and the standard version of CLIPGaussian, we conducted a formal online user study on the same dataset as in the quantitative comparison. It evaluated the quality of the style transfer and identified which method best conveyed the intended style. We conducted four surveys to evaluate different methods of 3D content stylization: (1) 3D objects stylization with images, (2) 3D scenes stylization with images, (3) 3D objects stylization with text, and (4) 3D scenes stylization with text. Each survey had 30 randomly selected participants, one user could participate in multiple surveys. The surveys were conducted formally using the CLICKworker platform[3]. A detailed description of the questions is provided in Appendix B.2.3.

Fig. 4 shows the results from four evaluated scenarios. In case (1), our method was most frequently selected for producing objects most similar to the reference image. In case (2), G-Style performed better on larger scenes, but our method was rated as the least similar to the reference image less often than StyleGaussian. Considering that G-Style uses over twice as many Gaussians, our method remains a competitive alternative (see Table 1). In case (3), users most often rated our approach as the best for text-based object stylization. Case (4), involving 3D scene stylization from text, yielded mixed results. Users gave high and low ratings, likely due to the subjective nature of prompt interpretation. A post hoc Conover-Friedman test showed significant differences in "perceived similarity" rankings across methods, except between our method and G-Style for object stylization.

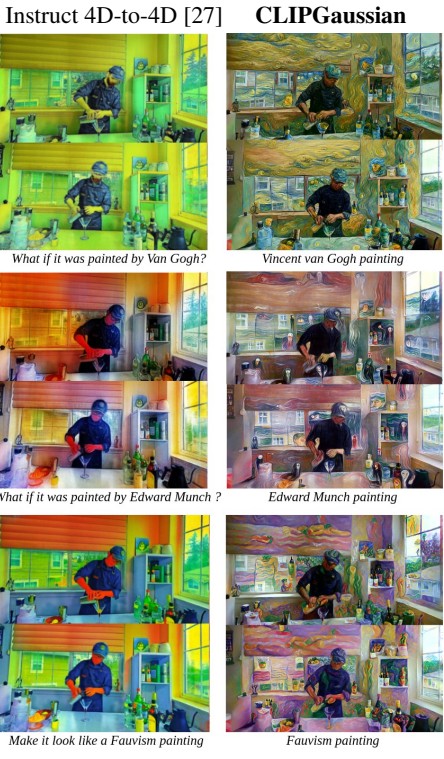

Instruct 4D-to-4D [27]     **CLIPGaussian**

*What if it was painted by Van Gogh?*     *Vincent van Gogh painting*

*What if it was painted by Edward Munch ?*     *Edward Munch painting*

*Make it look like a Fauvism painting*     *Fauvism painting*

Figure 9: Comparative analysis of style transfer using text condition on the *coffee_martini* dataset [54].

Fig. 5 and Fig. 6 show a qualitative comparison with selected baseline methods, on a subset of data used for quantitative evaluation and user study. Full comparison is provided in Appendix B.2. Unlike other methods, CLIPGaussian does not change the number of Gaussians, which also enables style interpolation (see Fig. 7). Additional results and train time can be found in Appendix B.2.

**4D** To evaluate performance on dynamic 4D scenes, we used D-MiSo [6], which integrates the Multi-Gaussian Splatting representation with a deformation network. All components of the pipeline actively contribute to the learning of style transfer.

---

[3]https://www.clickworker.com

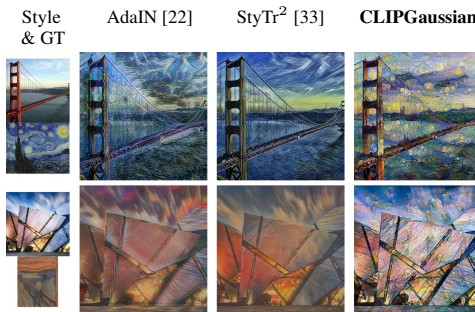

Figure 10: Comparison of 2D style transfer using image condition.

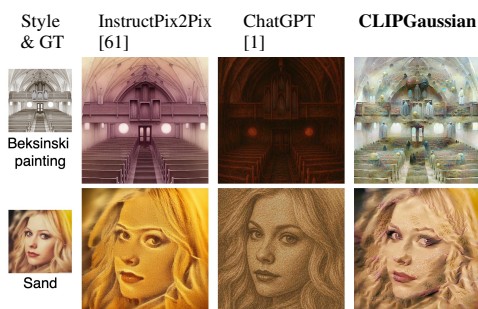

Figure 11: Comparison of 2D style transfer using text condition.

Empirical evaluations were performed on well-established benchmark datasets. The neural 3D video dataset *coffee_martini* (DyNeRF) [54] provides time-synchronized and calibrated multiview video sequences capturing complex 4D dynamic scenes. The D-NeRF dataset [60] consists of seven moving objects, with the constraint that only one camera view is accessible at any given time step.

For comparative evaluation, we examined CLIPGaussian against 4DStyleGaussian [26]. However, at the time of submission, the official implementation of this method was not publicly available. Consequently, visual comparisons were made using qualitative results extracted directly from the paper. Fig. 8 shows a comparison of style editing results produced by both methods, using the referenced image as the style source. Our method captures much more details in the style, especially in patterns and textures. Fig. 9 presents a comparison with Instruct 4D-to-4D [27], a first instruction-guided 4D scene editing method based on diffusion models and NeRFs, which imposes significant hardware requirements. In this comparison, we evaluate style transfer using a text prompt instead of a reference image. Instruct 4D-to-4D focuses mainly on the color palette, while our approach additionally modifies the geometry of the scene. Consequently, the results obtained by CLIPGaussian contain more details. Fig. 3 shows objects from D-NeRF stylization using text prompts, demonstrating that our model performs well not only on 4D scenes. Additional quantitative results, train time comparison and discussion about background loss can be found in Appendix B.3.1.

**2D** We evaluate performance of CLIPGaussian on 2D images using MiRaGe [8] on a subset of MS-COCO [62], which is commonly used for evaluating style transfer [29, 22]. Fig. 10 shows a qualitative comparison with AdaIN [22] and StyTr$^2$ [33], demonstrating image-guided style transfer. Fig. 11 provides a visual comparison with InstructPix2Pix [61] and ChatGPT [1], using text-based stylization. While we acknowledge that our method may not achieve the same level of visual quality as large-scale or diffusion-based models, these models often alter the identity or structure of the stylized subject. In contrast, CLIPGaussian preserves the original content of the image. Additionally, GS-based 2D style transfer benefits from properties such as realistic and localized editing in 3D space, as demonstrated in [8]. Notably, in 2D, repeating structures are more visible. This is particularly evident in Fig. 10 (bottom row) and Fig. 11 (top row). In both cases, stylizations aligned with the chosen references. The *patch_size* parameter controls the local stylization. For example for *patch_size = 128*, each patch of size 128x128 is stylized. For this reason, detailed formation can be controlled through the *patch_size* parameter. Additional comparisons are provided in Appendix B.4.

**Video** To evaluate our algorithm on the video style transfer task, we employed the VeGaS model [10], which represents videos using 3D Gaussian Splatting. The evaluation was conducted on the DAVIS dataset [63], a high-quality, high-resolution video collection commonly used for video object segmentation. The dataset comprises numerous videos, each containing fewer than 100 frames. Additionally, to test performance on a longer videos we use the Ultra Video Group (UVG) dataset [64]. The dataset is composed of 16 versatile high-resolution test video sequences captured at 50/120 fps. Fig. 12 presents a qualitative comparison with existing baseline models CCPL [37] and UniST [47], showcasing an example of image-based style transfer. The figure displays the first and last frames of the respective videos to illustrate the style transformation over time. Our method produces visually more coherent and aesthetically pleasing results, better preserving the style details while maintaining high fidelity to the video content. Fig. 13 provides a visual comparison with Text2Video [42] and RerenderAVideo [45], illustrating an example of text-conditioned style transfer; here too, the first and last frames are shown. Compared to RerenderAVideo, our method demonstrates significantly better temporal consistency, reducing flickering and preserving motion coherence. Compared to

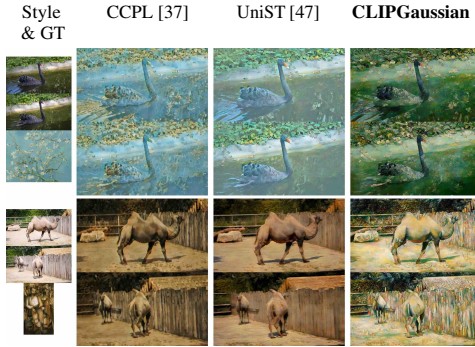

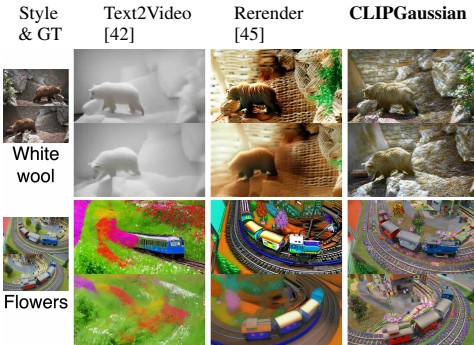

Figure 12: Comparison of video style transfer using image condition on the DAVIS dataset [63]. CLIPGaussian produces more detailed results.

Figure 13: Comparison of video style transfer using text condition on the DAVIS dataset [63]. CLIPGaussian shows better temporal consistency than baseline methods.

Text2Video, we achieve higher visual quality with more faithful adherence to the intended style prompt. All comparisons were conducted using the default settings for each method. Additional comparisons and quantitative evaluations are provided in Appendix B.5.

Our experiments show that CLIPGaussian achieves comparable or superior style transfer. Moreover, CLIPGaussian maintains significantly better temporal consistency than prior methods. This is because video content is modeled as a set of Gaussian primitives, where each Gaussian spans multiple frames. As a result, style modifications applied to the Gaussians naturally propagate coherently over time.

## 5    Conclusions

CLIPGaussian is designed as a plug-in model, compatible with Gaussian Splatting-based architecture. Our method enables effective style transfer from either a text prompt or an image to reconstructed objects. This flexibility allows CLIPGaussian to operate seamlessly across various data modalities, including 2D images, videos, 3D objects, and 4D dynamic scenes.

In the 3D domain, user study experiments demonstrate a strong preference for the stylized outputs produced by CLIPGaussian. This suggests that our approach offers not only technical effectiveness but also practical appeal for creative applications. Quantitative results indicate that our method achieves comparable or superior visual quality relative to existing baselines.

**Limitations**    Given the maturity of existing 2D image stylization methods, we acknowledge that our results in 2D may not match the visual quality achieved by large models or diffusion-based methods. Although CLIPGaussian is designed for universal applicability, its performance is contingent upon the quality of the underlying base models. Moreover, when the selected base fails to reconstruct the scene with high fidelity, style transfer may become intractable or produce perceptually implausible results.

**Social impact**    CLIPGaussian enables cross-modal generality as a plug-in-based on Gaussian Splatting models. Using CLIP as a multimodal vision and language model allow it to enable high-quality stylization and expand creative possibilities. Gaussian-based representation enables the creation of high-quality real-time renders which is a key challenge in computer vision, especially in broad applications in AR, gaming, and digital content creation.

## Acknowledgments

The work of K. Howil, J. Waczyńska, P. Borycki, and P. Spurek was supported by the project *Effective Rendering of 3D Objects Using Gaussian Splatting in an Augmented Reality Environment* (FENG.02.02-IP.05-0114/23), carried out under the First Team programme of the Foundation for Polish Science and co-financed by the European Union through the European Funds for Smart Economy 2021–2027 (FENG). The work of M. Mazur was supported by the National Centre of Science, Poland Grant No. 2021/43/B/ST6/01456. We gratefully acknowledge Polish high-performance computing infrastructure PLGrid (HPC Centers: ACK Cyfronet AGH, CI TASK, WCSS) for providing computer facilities and support within computational grant no. PLG/2025/018296.

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

# A  Base Model Choice: Implementation Details

**3D** The vanilla Gaussian Splatting [3] framework represents a scene using a dense set of 3D Gaussians, denoted as $\mathcal{G} = \{(\mathcal{N}(\mathrm{m}_i, \Sigma_i), \sigma_i, c_i)\}_{i=1}^n$,, where $m$ specifies the mean position, $\Sigma$ the covariance matrix, $\sigma$ the opacity and $c$ the colors of the Gaussians spherical harmonics. The optimization procedure iteratively renders images from these Gaussians and refines their parameters by minimizing the discrepancy between the rendered outputs and the corresponding ground-truth training views.

Our method refines Gaussian primitives by optimizing their position, color, scale, rotation, and opacity parameters, allowing for both appearance and geometry modifications. Our model is trained for 5000 steps without densification or pruning using loss function described in the section 3. If not stated otherwise we set $\lambda_b = 1000$, $\lambda_p = 90$, $\lambda_d = 5$, $\lambda_c = 0.8$, `patch_size` $= 128$, `num_patch` $= 64$ and `feature_lr` $= 0.01$. For the CLIPGaussian-Light variant we set `feature_lr` $= 0.002$ and $\lambda_p = 35$. The remaining hyperparameters follow default values of original GS implementation.

**4D** Many models in the literature are dedicated to 4D (3D dynamic) scenes [4, 65, 66]. Our approach can be adapted to all models since they all use General Gaussian components and an additional model for modeling time dependence. To present our concept for such models, we choose the D-MiSo model [6], which extends the Gaussian Splatting framework through a Multi-Gaussian structure and deform networks. A multi-Gaussian structure uses a hierarchy based on the relationship between the Core and sub-Gaussian. Core Gaussians are designated to capture and model motion dynamics:

$$\mathcal{G}_{core} = \{(\mathcal{N}_{core}(\mathrm{m}_i, R_i, S_i), \sigma_i, c_i)\}_{i=1}^p, \tag{6}$$

where $m$ specifies the mean position, $R$ the rotation matrix, $S$ the scaling parameters, $\sigma$ the opacity and $c$ the colors of the Gaussians. Sub-Gaussians serve to enhance rendering quality, and is defined as

$$\mathcal{G}_{sub} = \{(\mathcal{N}_{sub}(\mathrm{m} + R\boldsymbol{\alpha^{i}}^T, R^i, S^i), \sigma^i, c^i)\}_{i=1}^k, \tag{7}$$

where $\mathrm{m}$, $R$ is Core-Gaussian position and rotation; and $\boldsymbol{\alpha^i}$ are trainable parameters used to define the positions of the Sub-Gaussian relative to the Core-Gaussian.

CLIPGaussian style transfer jointly optimizes the Gaussian primitive parameters and the deformation networks. Our first stage is a classical D-MiSo framework, which gives us scenes represented by trainable parameters of D-MiSo, see Fig. 2. The second phase is dedicated to style transfer. Similar to 3D as default we set $iterations = 5000$, $\lambda_p = 90$, $\lambda_d = 5$, $\lambda_c = 0.8$, `patch_size` $= 128$, `num_patch` $= 64$. We use `feature_lr` $= 0.025$. The remaining hyperparameters follow default values of original D-MiSo implementation. The background loss weighting parameter $\lambda_b$ was set to a default value of 0, which is suitable for most styling experiments with text prompts. In special cases, it was adjusted to 500.

**2D** Adapting 3D Gaussian Splatting frameworks, originally designed for 3D scene representation, to the domain of single-image 2D reconstruction presents significant challenges [7]. MiraGe [8] models 2D image $I$ by embedding flat, parametric Gaussian primitives within a perceptually motivated 3D latent space, aligning with human perception. Each Gaussian is defined as $\mathcal{G} = \{(\mathcal{N}(\mathrm{m}_i, R_i, S_i), \sigma_i, c_i)\}_{i=1}^n$, where $R$ is rotation matrix and $S = \mathrm{diag}(s_1, s_2, s_3)$ is a diagonal matrix containing the scaling parameters, where $s_1 = \epsilon$. To improve generalization and capture symmetries inherent in natural 2D images, MiraGe employs mirrored input image $\mathcal{M}(I)$ during training as a form of data augmentation. While the standard 3D Gaussian Splatting framework employs a loss function of the form $\mathcal{L} = (1 - \lambda \mathcal{L}_1(I, GS(I)) + \lambda \mathcal{L}_{D-SSIM}(I, GS(I))$, the MiraGe model extends this by incorporating both the original image $I$ and its mirrored $\mathcal{I}$ utilizes a cost function $\mathcal{L}(I) + \mathcal{L}(\mathcal{M}(I))$.

We optimize position, color, scale, rotation, and opacity parameters of the Gaussian primitives. We train our model for 5000 steps without densification or pruning, using loss function described in the section 3 (modified by using mirror camera as in MiraGe). If not stated otherwise we set $\lambda_b = 0$, $\lambda_p = 90$, $\lambda_d = 5$, $\lambda_c = 0.8$, `patch_size` $= 128$, `num_patch` $= 64$ and `feature_lr` $= 0.0025$. The remaining hyperparameters follow default values of original MiraGe implementation.

**Video** For a video representation, we use the VeGaS model [10], which models a video as a collection of 3D Folded Gaussians, denoted as

$$\mathcal{G}_{\mathrm{VeGaS}} = (\mathcal{FN}(\mathrm{m}, \Sigma, a, f), \rho, c) \tag{8}$$

Here, $a$ and $f$ are temporal folding functions that modulate the Gaussian shape over time. Each video frame is rendered as a 2D projection of these Gaussians, positioned and shaped according to their trained spatial-temporal parameters.

CLIPGaussian maintains temporal consistency. This is because video content is modeled as a set of Gaussian primitives, where each Gaussian spans multiple frames. As a result, style modifications applied to the underlying Gaussians naturally propagate coherently over time. To preserve it, our framework optimizes only colors of the Gaussians spherical harmonics. Our model is trained for 5000 steps without densification or pruning using loss function described in the section 3. If not stated otherwise we set $\lambda_p = 90$, $\lambda_d = 5$, $\lambda_c = 0.5$, `patch_size` $= 128$, `num_patch` $= 64$ and `feature_lr` $= 0.02$. For the CLIPGaussian-Light variant we set `feature_lr` $= 0.002$ and $\lambda_p = 45$. The remaining hyperparameters follow default values of original VeGaS implementation.

# B Extended Experimental Results and Evaluation

This section presents additional experimental results and analyses. We begin with a detailed description of the evaluation metrics used to assess the effectiveness of our approach. Subsequently, we provide extended results across various data modalities, including images, videos, 3D objects, and 4D dynamic scenes. The supplementary materials contain additional *.mp4* files.

## B.1 Evaluation Metrics

### B.1.1 CLIP-based metrics

In the quantitative evaluation, we report four CLIP-based metrics. *CLIP Directional Similarity* [52] and *CLIP-S* [57] measure the quality of style transfer while *CLIP Directional Consistency* [58] and *CLIP-F* [59] measure the consistency and similarity to the original content. For all metrics, we use `ViT-L/14` CLIP model [53].

Let $E_{pos}$ denote the CLIP embedding of a style (either a style image or a style prompt), and $E_{neg}$ denote the embedding of a negative prompt. In the case of the NeRF Synthetic dataset, we use object names (e.g., "a lego" for *lego* object) with ImageNet prompt templates [53]. Let $E_{render}(i)$ and $E_{gt}(i)$ represent the CLIP embeddings of the stylized render and the ground truth of the $i$-th test image, respectively. $N$ is the size of a test set. We define the *CLIP Directional Similarity* and *CLIP-S* as follows:

$$\textit{CLIP-SIM} = \frac{1}{N} \sum_{i=1}^{N} \cos(E_{render}(i) - E_{gt}(i), E_{pos} - E_{neg}), \tag{9}$$

$$\textit{CLIP-S} = \frac{1}{N} \sum_{i=1}^{N} \cos(E_{render}(i), E_{pos}) \tag{10}$$

Assuming that testing frames come from a video, we can also define *CLIP Directional Consistency* and *CLIP-F* as follows:

$$\textit{CLIP-CONS} = \frac{1}{N-1} \sum_{i=1}^{N-1} \cos(E_{render}(i+1) - E_{render}(i), E_{gt}(i+1) - E_{gt}(i)) \tag{11}$$

$$\textit{CLIP-F} = \frac{\sum_{i=1}^{N-1} cos(E_{render}(i+1), E_{render}(i))}{\sum_{i=1}^{N-1} cos(E_{gt}(i+1), E_{gt}(i))} \tag{12}$$

### B.1.2 Additional consistency metrics

For additional consistency evaluation, we use short-range and long-range consistency following the implementation and settings from StyleRF [67]. Specifically we warp one view to the other according to the optical flow using softmax splatting, and then compute the masked RMSE and LPIPS scores to measure the stylization consistency. For short-range we average over pairs of $i$-th and $(i+1)$-th frames and for long-range we take $i$-th and $(i+7)$-th frames.

In addition to these metrics, for evaluation of video consistency, we employ the Farneback optical Flow based metric, following the setup proposed in the ViSt3D [41]. Accordingly, in the case of video:

$$m_{FoF,k} = \frac{1}{N} \sum_{i=1}^{N} |FoF(GT_i, GT_{i+k}) - FoF(Style_i, Style_{i+k})|, \tag{13}$$

where $FoF$ is Farneback optical Flow function, $GT_i$ is $i$-th frame from the original video and $Style_i$ is $i$-th frame from the stylized video.

## B.2 3D: Additional Results and Explanation

This section provides a detailed description of the quantitative evaluation setup.

During experiments we used two objects (*lego* and *hotdog*) from the NeRF-Synthetic dataset [55], and two scenes (*garden* and *bonsai*) from the Mip-NeRF 360 dataset [56]. Each was stylized using four text prompts ("Fire", "Mosaic", "Starry Night by Vincent van Gogh", and "Scream by Edvard Munch") and four style images (as shown in Fig. 6).

We evaluated CLIPGaussian (our method), DGE [14], SGSST [50], ABC-GS [51] and G-Style [21] using the same base models trained with the original Gaussian Splatting codebase[4]. Since Instruct-GS2GS [13] is incompatible with this codebase, it was evaluated on models trained using Nerfstudio[5]. Due to the high VRAM requirements of StyleGaussian [16], it was evaluated on models trained with its own training script, which limits the number of Gaussians to $10^5$. As Instruct-GS2GS [13] and DGE [14] employ different prompt templates, we prepended our text prompts with the prefixes used in their original papers: "Turn it into a "for Instruct-GS2GS and "Make it look like a" for DGE. Baseline methods were trained using their default configurations. CLIPGaussian models were trained according to the settings described in Appendix A, except for the "Fire" condition, which used a higher feature learning rate (0.02). Additionally, for the *lego* object with the "Fire" condition, we set $\lambda_{bg} = 100$.

### B.2.1 Training Time

We evaluate training time using the same dataset as in the user study and quantitative analysis. For each scene, we report the average training time across all styles. Table 2 shows how the number of Gaussian primitives affects stylization time.

Table 2: Number of Gaussian primitives impact on stylization time.

| Scene | Number of Gaussians | Avg. stylization time |
|---|---|---|
| *hotdog* | 0.14M | 11m29s |
| *lego* | 0.31M | 11m36s |
| *bonsai* | 1.35M | 11m37s |
| *garden* | 4.48M | 21m03s |

### B.2.2 Additional consistency evaluation

The consistency of the stylized objects was evaluated using the metrics described in Appendix B.1.2. Table 3 reports the resulting short-range and long-range consistency metrics, demonstrating that CLIPGaussian achieves results comparable to the original data.

### B.2.3 User Study: Details

We conducted a formal survey using the CLICK-worker platform [6]. This allowed us to recruit a demographically balanced participant pool in terms of gender, age. Each participant was presented with a questionnaire in which they evaluated stylized objects or scenes stylized using a text prompt or reference image. We wanted the survey for both conditional text and image to be comparable, so we chose prompts corresponding to images.

The survey was divided into two parts: a main evaluation section and additional question in the

Table 4: Friedman statistic test $F_r$ and $p$ values for user study *Question 1*. In case of image condition we consider StyleGaussian G-Style, CLIPGaussian and CLIPGaussian, in case text condition we consider I-GS2GS, DGE, CLIPGaussian

| | condition | $F_r$ | $p$ |
|---|---|---|---|
| object | image | 139.3 | 5.62e-31 |
| scene | image | 1178.8 | 1.44e-39 |
| object | text | 54.70 | 1.32e-12 |
| scene | text | 8.40 | 0.014 |

---

[4] https://github.com/graphdeco-inria/gaussian-splatting
[5] https://github.com/nerfstudio-project/nerfstudio
[6] https://www.clickworker.com [access: 12.05.2025]

Table 3: Quantitative comparison of style transfer, compared against baseline methods.

| Model | Short-range consistency | | Long-range consistency | |
|---|---|---|---|---|
| | LPIPS ↓ | RMSE ↓ | LPIPS ↓ | RMSE ↓ |
| Original Videos | 0.053 | 0.047 | 0.140 | 0.123 |
| *Text-conditioned* | | | | |
| I-GS2GS [13] | **0.048** | **0.043** | 0.148 | 0.131 |
| DGE [14] | 0.055 | 0.044 | 0.148 | **0.120** |
| **CLIPGaussian-Light** | 0.051 | **0.043** | **0.132** | **0.117** |
| **CLIPGaussian** | 0.057 | 0.051 | 0.148 | 0.125 |
| *Image-conditioned* | | | | |
| StyleGaussian [16] | 0.052 | 0.061 | 0.145 | 0.144 |
| SGSST [50] | **0.040** | 0.050 | 0.139 | 0.142 |
| ABC-GS [51] | 0.048 | **0.043** | 0.141 | 0.126 |
| G-Style [21] | 0.054 | 0.049 | 0.145 | 0.136 |
| **CLIPGaussian-Light** | 0.055 | 0.045 | **0.139** | **0.120** |
| **CLIPGaussian** | 0.064 | 0.055 | 0.162 | 0.132 |

end. The main section included eight comparative examples, each involving stylized outputs from three different methods (see Fig. 5, 6). Fig. 18 illustrates a typical evaluation interface. In each example, participants were first shown the reference image or text prompt used for styling. This was followed by a GIF animation showing the original object or scene, and then the stylized results from each method. For each evaluation we also asked about stylized factors like *Color Palette, Lighting, Detail Level, Mood or Atmosphere, Brushstrokes*".

**Main evaluation consisted of four questions:**

- Question 1: *"Please rank the generated results according to how closely they match the style from most similar(1) to least similar(3)"* focused on style transfer quality. Participants ranked the three methods based on how well they reflected the reference style. Each method could be selected only once per ranking. For example, Method *X* might be ranked 3rd, Method *Y* 1st, and Method *Z* 2nd.

- Question 2: *"On a scale from "Very Low" to "Very High", how would you rate the visual appeal of each generated result?"* assessed visual appeal. For each method, participants selected one of several options (*"Very Low", "Low", "Medium", "High", "Very High"* ). This allowed users to express their perception of the aesthetic quality of each result independently.

- Question 3: *"On a scale from "Very difficult" to "Very easy", How well can you recognize the original scene content in each generated result"* focused on recognizability of content. The users evaluated how clearly the original object or scene was preserved after stylization. Options (*"Very Difficult", "Difficult", "Medium", "Easy", "Very Easy"*) independently rated for each method.

- Question 4: "On a scale from "Not important at all" to "Extremely important" how much does this factor influence your judgment of the styled 3D object?", asked about the factors influencing the user's decision. Participants chose from predefined options (*"Not important at all", "Not important", "Important", "Extremely important"*). Each factor was rated separately, following the same format as questions 2 and 3.

**Additional question:**

- In the context of *"Starry Night by Vincent van Gogh"* and "*Scream by Edvard Munch*". We asked an additional question: *"Were you familiar with the following images before?"*. Users could answer "Yes" or "No".

The poll format was consistent across all object/scene and text/image conditions. For evaluations based on text prompts, we wanted to avoid introducing visual bias, and we didn't attach any referenced image. Therefore, we included only those participants who were already familiar with the references, such as "*Scream by Edvard Munch*" and "*Starry Night by Vincent van Gogh*".

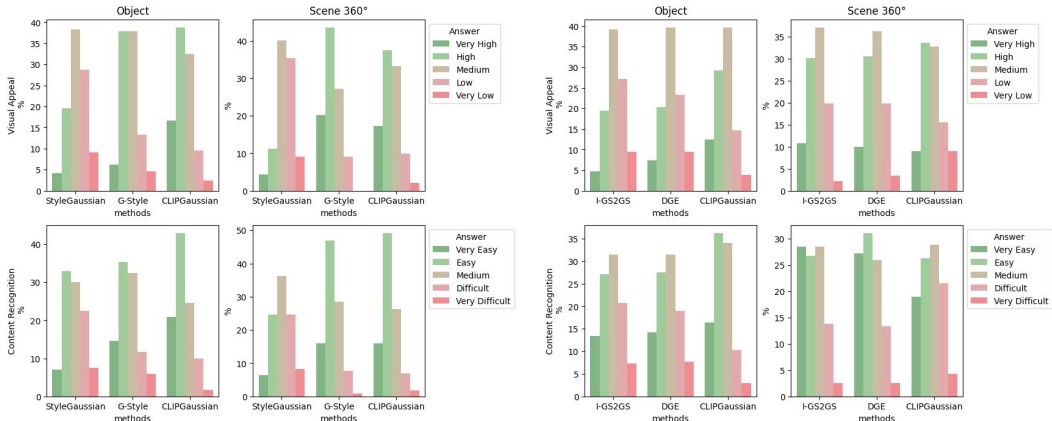

Figure 14: User study: Comparison of 3D style transfer using referenced image.

Figure 15: User study: Comparison of 3D style transfer using text condition.

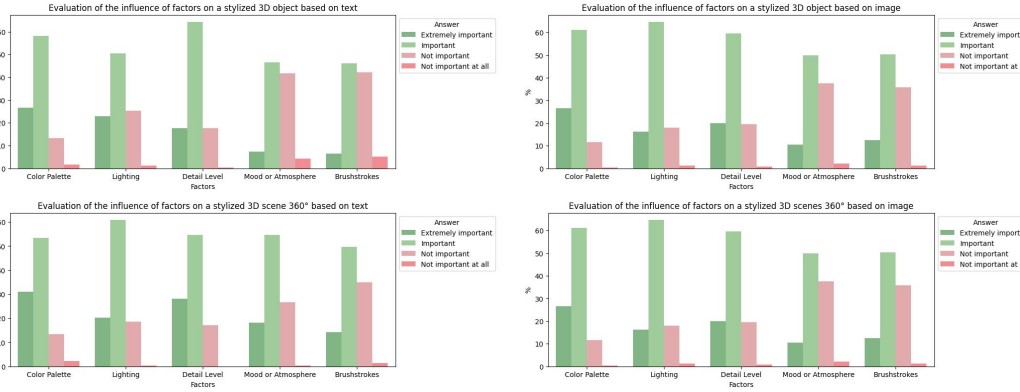

Figure 16: User study: Comparison of 3D style transfer using text condition.User ratings of factors influencing their judgment of styled 3D objects/scenes by text.

Figure 17: User study: Comparison of 3D style transfer using text condition.User ratings of factors influencing their judgment of styled 3D objects/scenes by image.

Tab. 4 shows Friedman statistic test $F_r$ and $p$ values for *Question 1* user study. For the image condition, we evaluated three methods: StyleGaussian, G-Style, and CLIPGaussian. For the text condition, the evaluated methods were I-GS2GS, DGE, and CLIPGaussian. In all scenarios (1-4) we observe significant differences between methods (reject H0). Additional results for image-based style transfer are presented in Fig. 14, while results for text-based style transfer are shown in Fig. 15. The post hoc Conover Friedman test for a user study on stylized objects/scenes by image/text revealed statistically significant differences between methods ($p < 0.05$), indicating if the null hypothesis can be rejected is shown in the Tabs. 5, 6, 7 ,8. In almost every case we can reject H0, the exception is the comparison of our model with G-Style in the case of evaluation on objects stylized with a reference image.

Fig 16 and Fig. 17 show that the majority of participants identified color and lighting as highly influential factors in their assessment of styled 3D objects/scenes. In comparison, brushstroke details were generally regarded as less important to their judgments. A high-resolution comparison across object/scenes styled by text-based methods is shown in Fig. 20 and Fig. 21. A corresponding comparison for image-based methods is presented in Fig. 22 and Fig. 23.

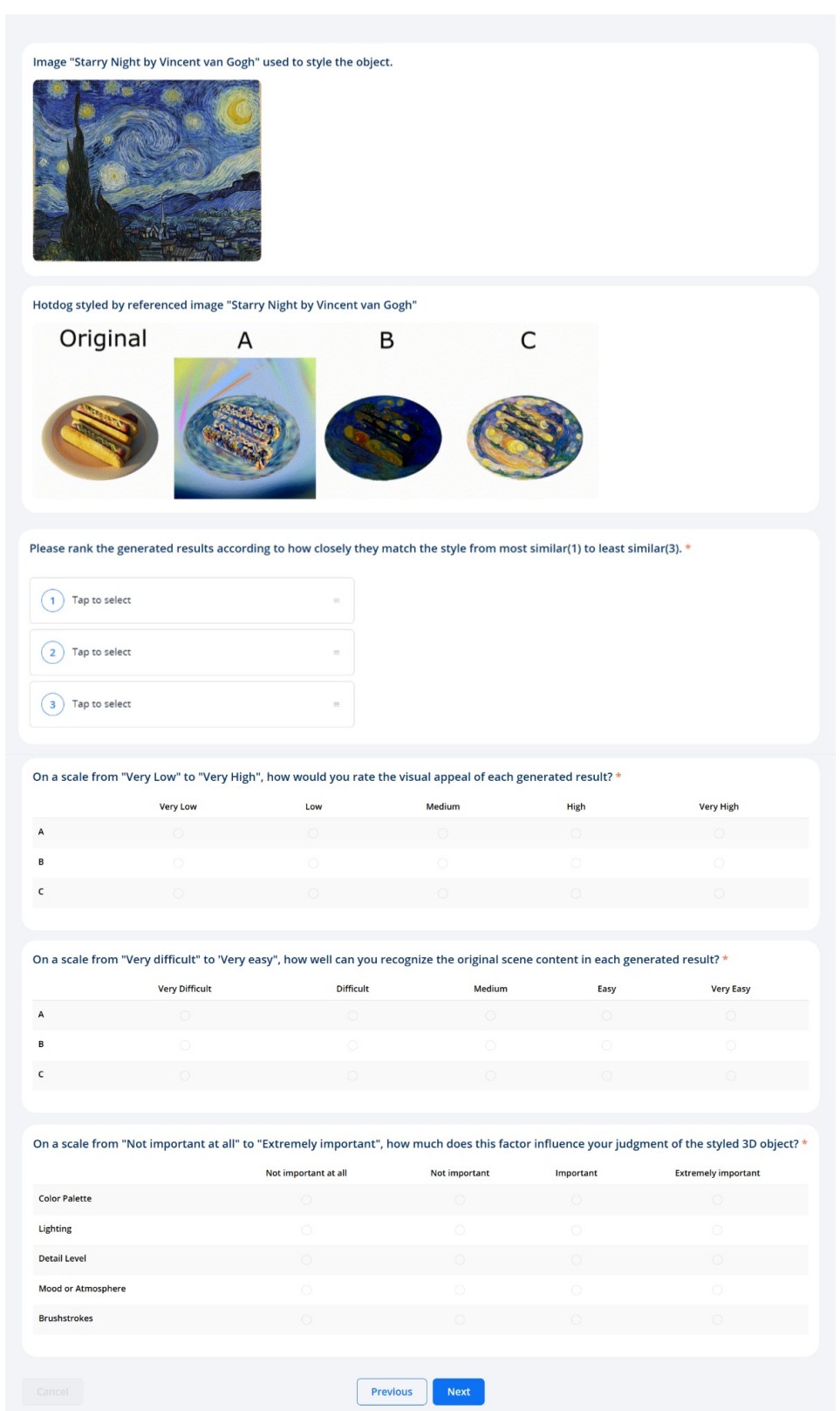

Figure 18: Survey interface view from the ClickWorker website, users evaluate stylization of the 3D object *hotdog* conditioned on the painting *Starry Night by Vincent van Gogh*.

| Style | CLIPGaussian-Light | CLIPGaussian |

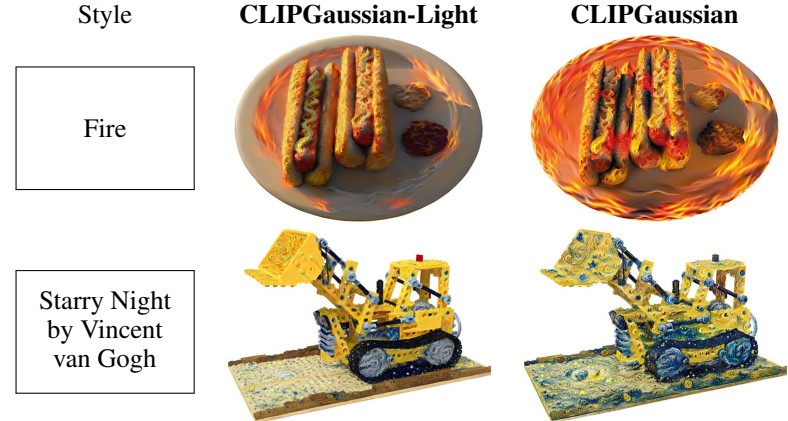

Figure 19: Visual comparison of CLIPGaussian and CLIPGaussian-Light in 3D style transfer, conditioned by text, on *hotdog* and *lego* objects from NeRF-Synthetic dataset [55].

Table 5: Calculated $p$-value Posthoc Conover Friedman test for user study in case of objects stylized by image.

|  | StyleGaussian | G-Style | CLIPGaussian |
|---|---|---|---|
| StyleGaussian | 1.00e+00 | 1.45e-25 | 5.71e-33 |
| G-Style | 1.45e-25 | 1.00e+00 | 6.66e-02 |
| CLIPGaussian | 5.71e-33 | 6.67e-02 | 1.00e+00 |

Table 6: Calculated $p$-value Posthoc Conover Friedman test for user study in case of scene stylized by image.

|  | StyleGaussian | G-Style | CLIPGaussian |
|---|---|---|---|
| StyleGaussian | 1.00e+00 | 8.90e-51 | 1.68e-17 |
| G-Style | 8.90e-51 | 1.00e+00 | 3.32e-15 |
| CLIPGaussian | 1.68e-17 | 3.32e-15 | 1.00e+00 |

Table 7: Calculated $p$-value Posthoc Conover Friedman test for user study in case of objects stylized by text.

|  | I-GS2GS | DGE | CLIPGaussian |
|---|---|---|---|
| I-GS2GS | 1.000e+00 | 0.000067 | 2.39e-14 |
| G-Style | 6.67e-05 | 1.000000 | 1.24e-04 |
| CLIPGaussian | 2.39e-14 | 0.000125 | 1.00e+00 |

Table 8: Calculated $p$-value Posthoc Conover Friedman test for user study in case of scene stylized by text.

|  | I-GS2GS | DGE | CLIPGaussian |
|---|---|---|---|
| I-GS2GS | 1.00e+00 | 0.000067 | 2.39e-14 |
| DGE | 6.67e-05 | 1.00e+00 | 1.24e-04 |
| CLIPGaussian | 2.39e-14 | 0.000125 | 1.00e+00 |

### B.2.4 Hyperparameters Analysis

We evaluate the visual quality of 3D style transfer with respect to the parameters `feature_lr` and `patch_size` on three objects from the NeRF-Synthetic dataset: *lego*, *hotdog*, and *mic*. These objects are stylized using prompts such as *"Starry Night by Vincent van Gogh"*, *"Fire"*, and *"The Great Wave off Kanagawa by Katsushika Hokusai"*. The quantitative impact of `feature_lr` is shown in Tab. 9, while the influence of `patch_size` is reported in Tab. 10. Our observations indicate that both parameters contribute to enhancing the stylistic expressiveness of the output. However, excessively large values introduce greater flexibility in the spatial distribution of the Gaussians, resulting in a noticeable loss of content detail and reduced spatial consistency (see Fig. 24).

Additionally, we evaluate the effect of the weighting factors of the loss components, $\lambda_p$ and $\lambda_d$. The quantitative impact of $\lambda_p$ is shown in Tab. 11, while the influence of $\lambda_d$ is reported in Tab. 12. Qualitative results are presented in Fig. 25. A higher $\lambda_p$ increases the expressiveness of local style transfer; however, excessively high values may lead to overstylization of the object. Similarly, increasing $\lambda_d$ enhances the global style characteristics. Quantitative evaluations suggest that $\lambda_d$ parameter has limited impact on spatial consistency and content similarity.

Table 9: Effect of a feature learning rate on the performance of CLIPGaussian in terms of the CLIP metrics.

| patch_size | CLIP-S ↑ | CLIP-SIM ↑ | CLIP-F ↑ | CLIP-CONS ↑ |
|---|---|---|---|---|
| 32 | 18.53 | 19.51 | **98.08** | **15.02** |
| 64 | 21.73 | 19.31 | 97.70 | 11.06 |
| 128 | 25.60 | 29.78 | 97.87 | 6.93 |
| 256 | **27.45** | **32.65** | 97.96 | 4.42 |

Table 10: Effect of a feature learning rate on the performance of CLIPGaussian in terms of the CLIP metrics.

| feature_lr | CLIP-S ↑ | CLIP-SIM ↑ | CLIP-F ↑ | CLIP-CONS ↑ |
|---|---|---|---|---|
| 0.0025 | 25.24 | 29.31 | **97.68** | **7.91** |
| 0.005 | 25.60 | 29.77 | 97.99 | 6.93 |
| 0.01 | 25.61 | 29.93 | 97.61 | 6.26 |
| 0.02 | **26.27** | **30.09** | 97.54 | 6.35 |

Table 11: Effect of $\lambda_p$ parameter on the performance of CLIPGaussian in terms of the CLIP metrics.

| $\lambda_p$ | CLIP-S ↑ | CLIP-SIM ↑ | CLIP-F ↑ | CLIP-CONS ↑ |
|---|---|---|---|---|
| 0 | 16.00 | 14.12 | **98.74** | **11.03** |
| 90 | 23.67 | 25.07 | 97.95 | 2.36 |
| 180 | **24.36** | **25.27** | 97.78 | 1.48 |

Table 12: Effect of a $\lambda_d$ parameter on the performance of CLIPGaussian in terms of the CLIP metrics.

| $\lambda_d$ | CLIP-S ↑ | CLIP-SIM ↑ | CLIP-F ↑ | CLIP-CONS ↑ |
|---|---|---|---|---|
| 0 | 22.20 | 22.76 | 97.71 | 2.52 |
| 5 | 23.67 | 25.07 | 97.95 | 2.32 |
| 10 | **23.55** | **25.15** | **98.04** | **2.61** |

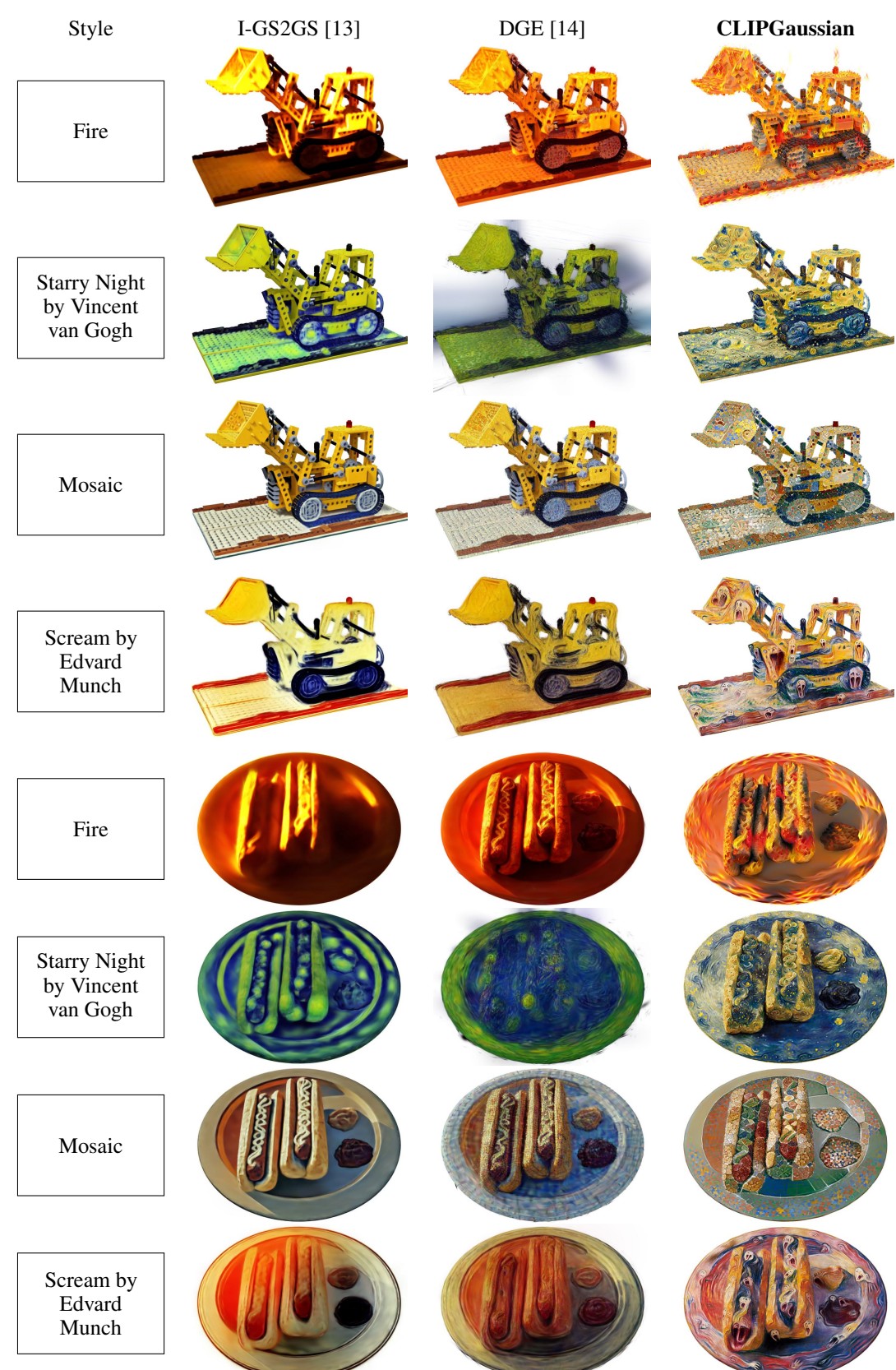

Figure 20: Full comparison of CLIPGaussian (our) and baseline models in 3D style transfer, conditioned by text, on *hotdog* and *lego* objects from NeRF-Synthetic dataset [55].

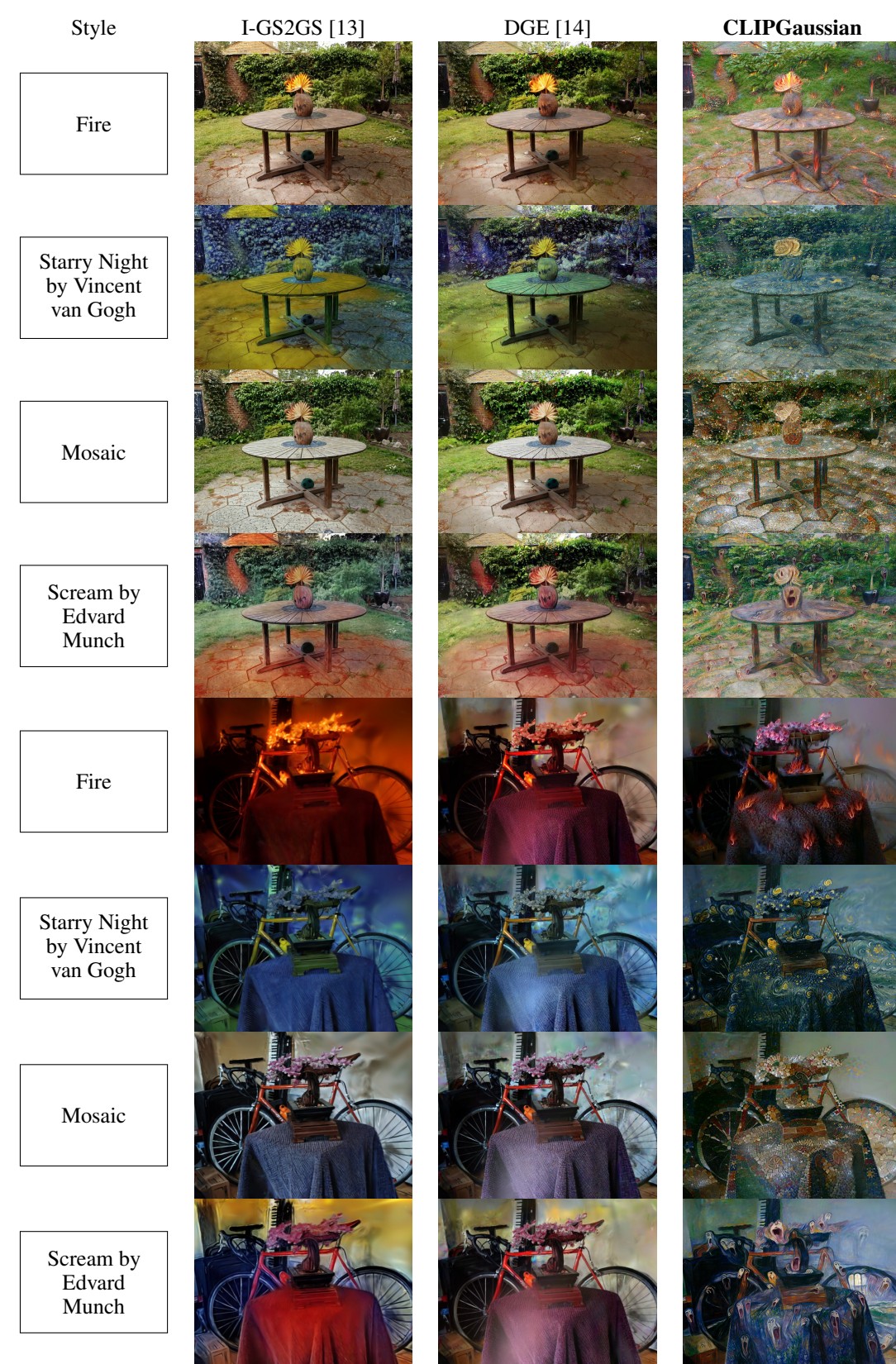

Figure 21: Full comparison of CLIPGaussian (our) and baseline models in 3D style transfer, conditioned by text, on *garden* and *bonsai* objects from Mip-NeRF 360 dataset [56].

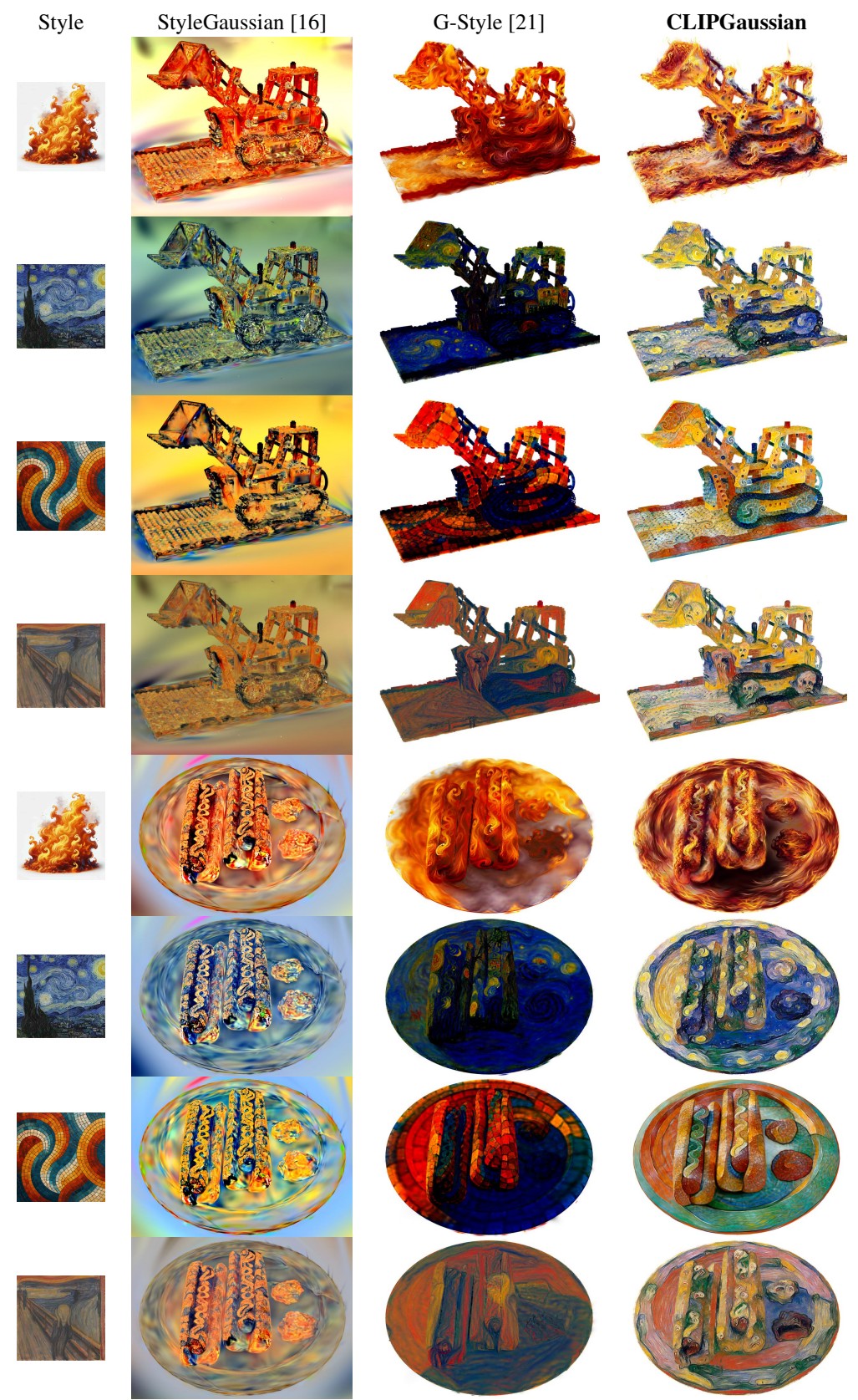

Figure 22: Full comparison of CLIPGaussian (our) and baseline models in 3D style transfer, conditioned by image, on *hotdog* and *lego* objects from NeRF-Synthetic dataset [55].

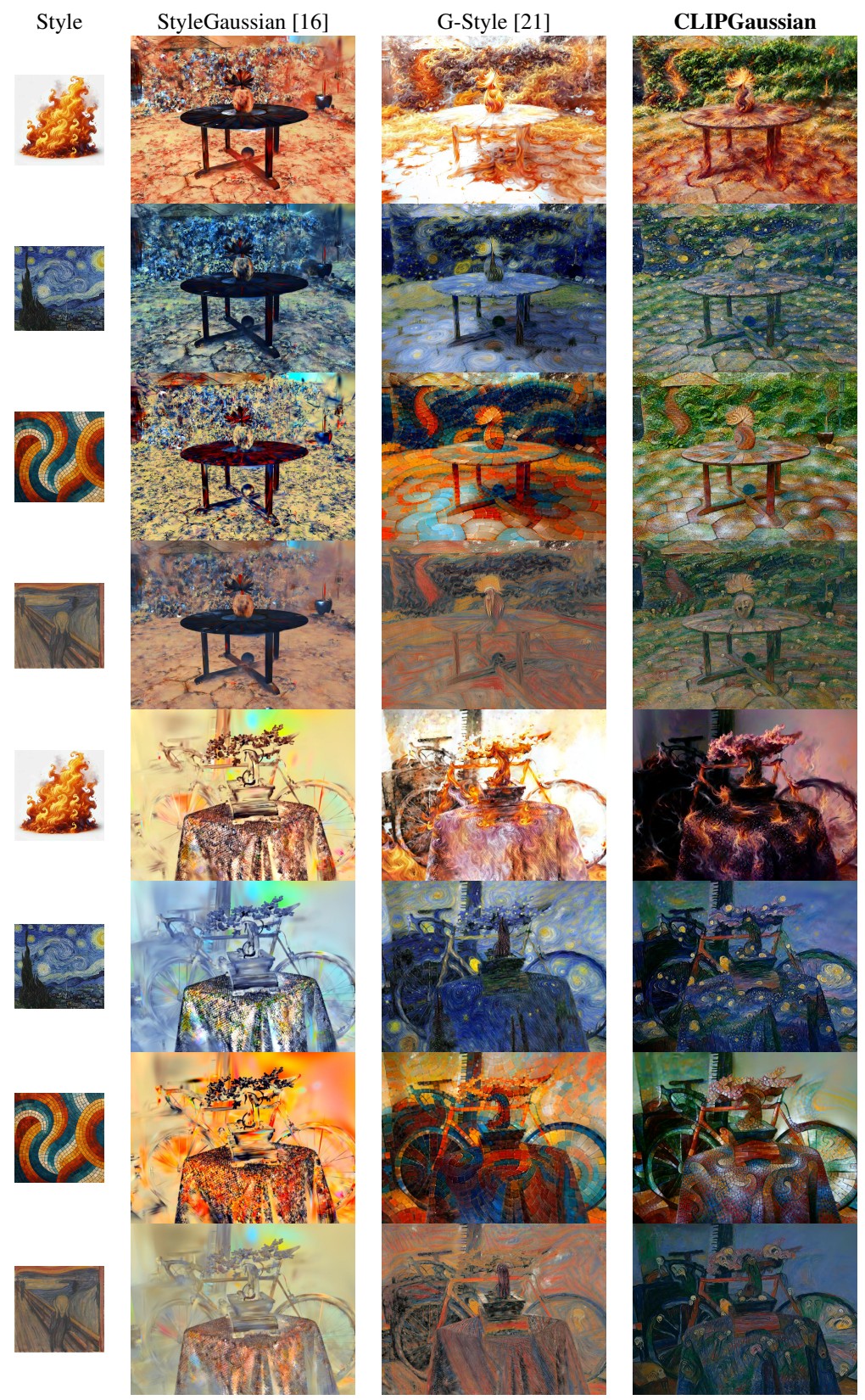

Figure 23: Full comparison of CLIPGaussian (our) and baseline models in 3D style transfer, conditioned by image, on *garden* and *bonsai* objects from Mip-NeRF 360 dataset [56].

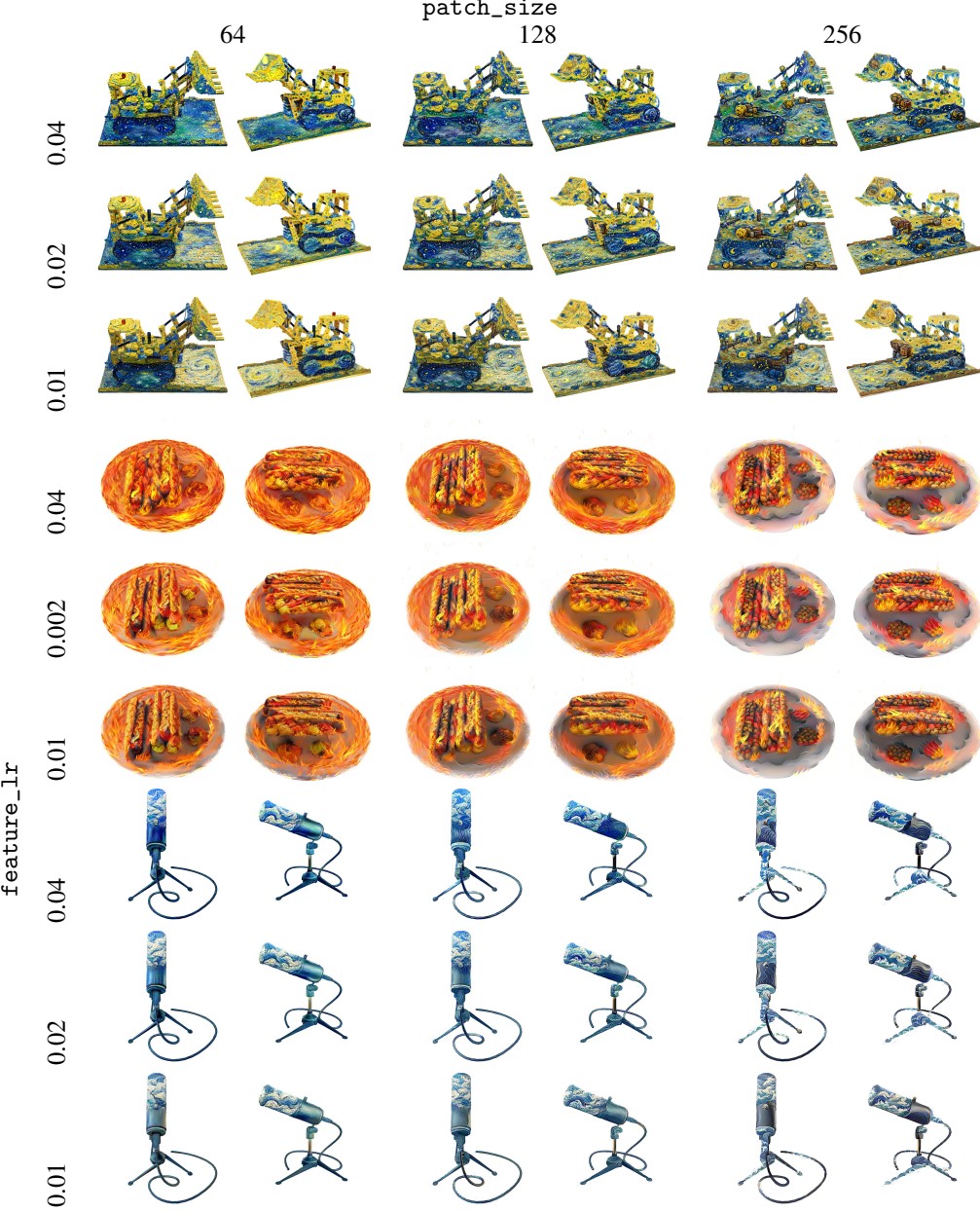

Figure 24: Effect of a patch size and a feature learning rate on the performance of CLIPGaussian on *lego*, *hotdog* and *mic* objects from NeRF-Synthetic dataset [55]. Objects are stylized with "Starry Night by Vincent van Gogh", "Fire" and "The Great Wave off Kanagawa by Katsushika Hokusai" prompts.

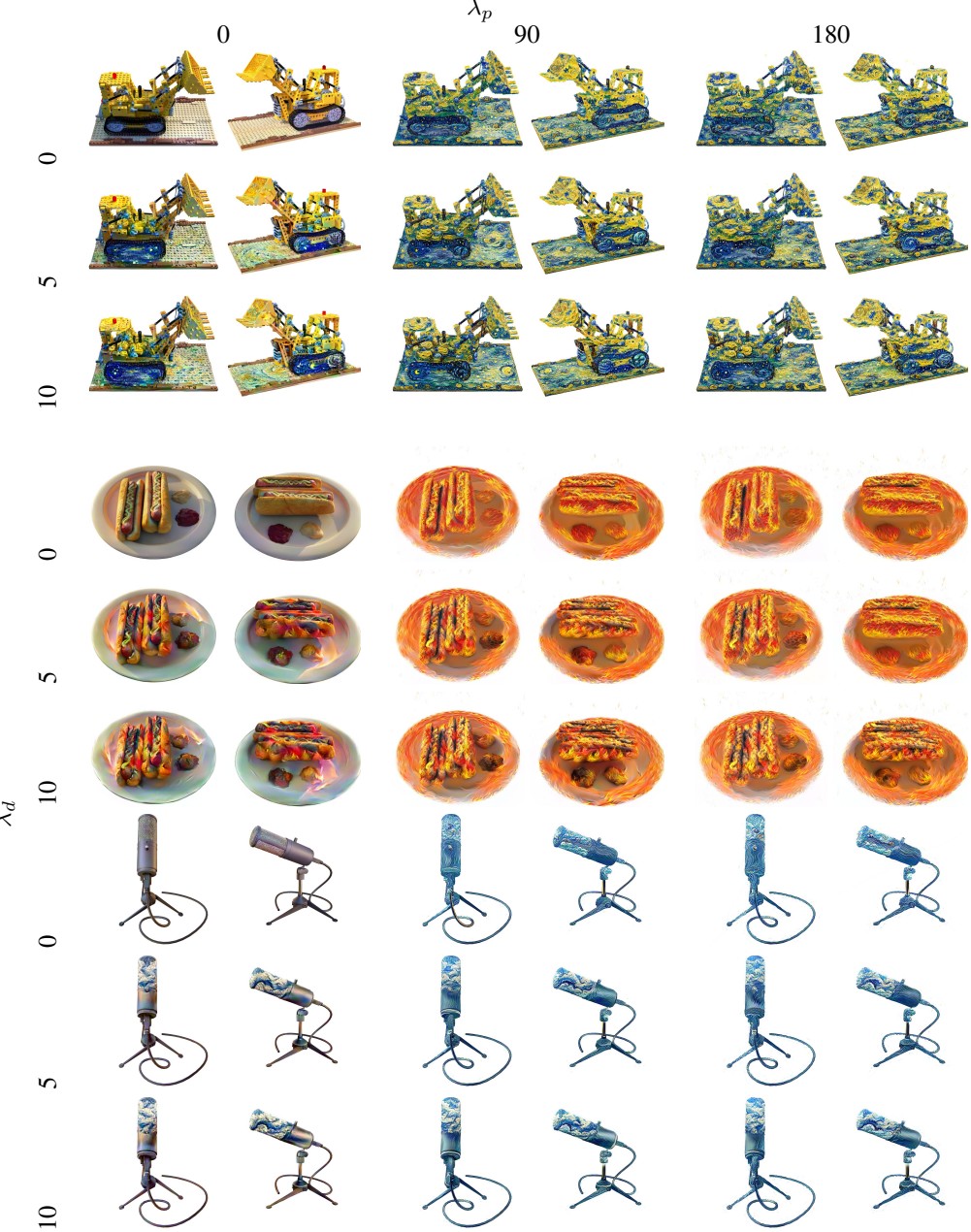

Figure 25: Effect of a $\lambda_p$ and a $\lambda_d$ rate on the performance of CLIPGaussian on *lego*, *hotdog* and *mic* objects from NeRF-Synthetic dataset [55]. Objects are stylized with "Starry Night by Vincent van Gogh", "Fire" and "The Great Wave off Kanagawa by Katsushika Hokusai" prompts.

### B.3 4D: Additional Results and Explanation

This section delves deeper into our experimental analysis of 4D scenes. We begin with a detailed exploration of stage 2 hyperparameters, followed by demonstrating the adaptability of our method on an additional multi-camera dataset.

Fig. 26 presents style transfer on selected objects from the D-NeRF dataset. These results illustrate that CLIPGaussian enables style conditioning via both text prompts and reference images, highlighting its versatility across multiple guidance modalities.

Text condition         Style image

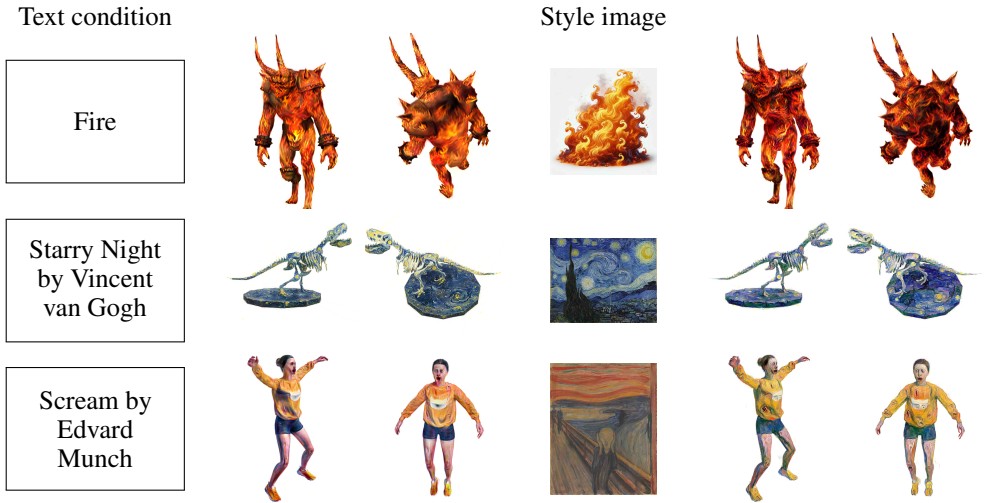

Fire

Starry Night by Vincent van Gogh

Scream by Edvard Munch

Figure 26: Style transfer results on samples from the D-NeRF dataset [60]. Our CLIPGaussian model accommodates both text-based and image-based style inputs.

#### B.3.1 Hyperparameters Analysis

For dynamic 3D scenes using the D-MiSo model, the background parameter was set to the default value of 0 in most cases. However, using the prompt *"Summer"* we could observe the background stylization as well. Fig. 27 shows the effect of background loss on the *Jumpingjacks* object: without background loss ($\lambda_{bg} = 0$) and with background loss ($\lambda_{bg} = 500$). More precisely, we used the alpha channel from the original images available in D-NeRF dataset to create a background mask $m$ for each view. In this case $L_b = \lambda_b \left( |m(I_l) - m(R_{\mathcal{G}}(I_l))| \right)$. The effect of using the background loss is also shown in Fig. 31 and Fig. 32.

We evaluate the visual quality of style transfer in 4D with respect to `feature_lr` and the number of patches $n$. Fig. 28 shows two representative objects from the D-NeRF dataset, each conditioned on a distinct text prompt (*Hellwarrior* - "The Great Wave of Kanagawa by Katsushika Hokusai", *Jumpingjacks* - "Spring"). feature_lr is influences primarily the color saturation of Gaussians. We observe that increasing this parameter enhances the stylistic expressiveness of the output. A higher $n$ generally improves styles details. In *Hellwarrior* example for the larger $n$ we see more "water swirls" appropriate to the style. However, excessively large values (e.g., $n$=128) introduce increased freedom in the

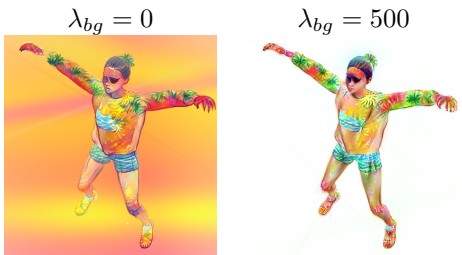

$\lambda_{bg} = 0$     $\lambda_{bg} = 500$

Figure 27: Effect of background loss $\lambda_{bg}$ on stylization *Jumpingjacks* object with the prompt "Summer".

Gaussians' spatial distribution, leading to a notable loss of content detail. This effect is particularly pronounced in the *Jumpingjacks* example, where reconstruction of the hand visibly deteriorates.

Fig. 29 shows that increasing batch size and the number of patches leads to noticeably improved visual fidelity in style transfer. Increasing the batch size enables the preservation of finer visual details.

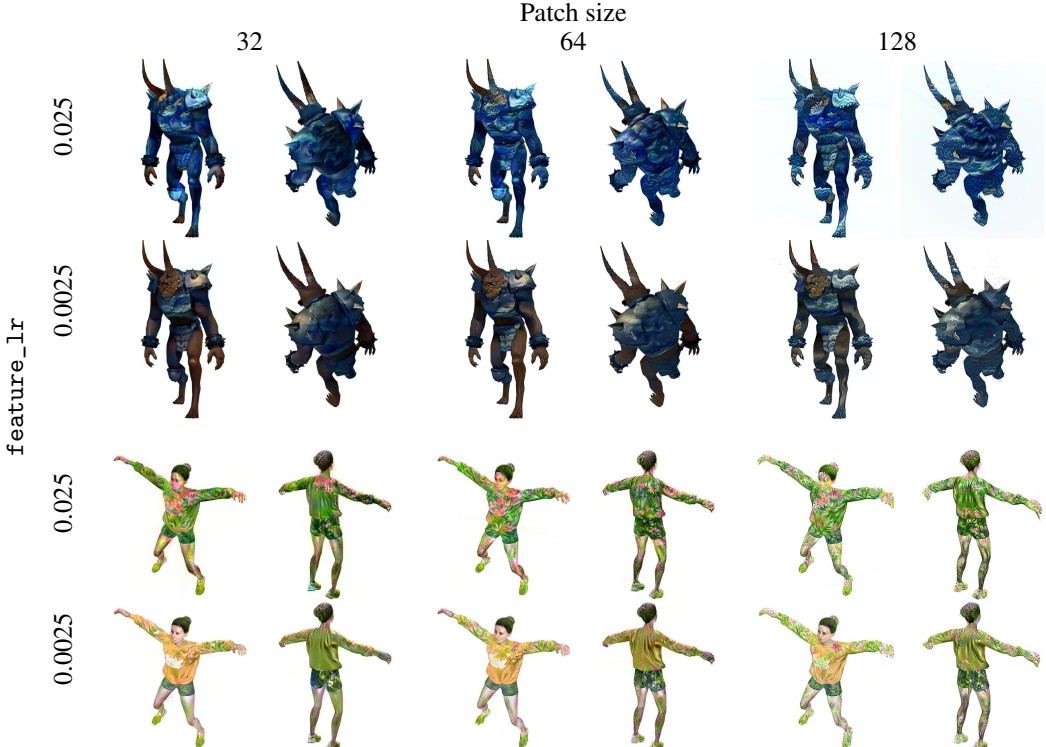

Figure 28: Visual comparison of style transfer results on D-NeRF objects under two text prompts "The Great Wave of Kanagawa by Katsushika Hokusai" and "Spring", showing the impact of `feature_lr` and number of patches.

We check the influence of batch size hyperparameter from the chosen base model (D-MiSo). A larger batch size prevents deformation of the subject's head. This highlights that our method faithfully inherits and reinforces the structural integrity imparted by the base architecture.

Table 13 presents a comparison of numerical training time and *CLIP-S*, *CLIP-SIM* metrics for the selected object under batch size, the text condition "Starry Night by Vincent van Gogh" and number of patch=32. The training time results reflect a dependency on the base architecture. The *CLIP-S* suggests that models trained with smaller batch strategies achieve marginally higher. However, the differences in *CLIP-S* between models remain relatively minor. In particular, despite the small numerical variance, visual comparison (Fig. 29) reveals substantial qualitative differences in output fidelity, highlighting the limitations of current metrics in capturing perceptual quality.

### B.3.2 Multi-Camera Setup

Fig. 8 in main paper shows experiment on The Neural 3D Video dataset (DyNeRF) [54]. It consists of videos captured by 21 cameras for each scene. The multi-view inputs were time synchronized and the images were extracted at 30FPS. In our experiments we use the first 24 frames following the data loader provided in [5] to show the capabilities of CLIPGaussian in real 4D scenes. In this dataset only the frontal views of the scenes are shown. In contrast the PanopticSports datasets [68] is full 360 view dataset, which comprises dynamic scenes featuring significant object and actor movements. Each scene was recorded using 31 cameras over 150 timesteps. Fig. 30 shows the transferred style is prominently expressed on both the primary actor and the background.

Table 13: Numerical comparison of training time and CLIP score, *CLIP-SIM* metrics for the selected object under the text condition "Starry Night by Vincent van Gogh" and number of patch=32, background white, checking influence of batch size hyperparameter from the chosen base model (D-MiSo).

| Batch | Hook | | | Jumpingjacks | | | Trex | | | BouncingBalls | | |
|---|---|---|---|---|---|---|---|---|---|---|---|---|
| | CLIP-S | CLIP-SIM | Train time | CLIP-S | CLIP-SIM | Train time | CLIP-S | CLIP-SIM | Train time | CLIP-S | CLIP-SIM | Train time |
| 1 | 27.73 | 22.68 | 10:12 | 23.15 | 18.32 | 10:35 | 28.00 | 23.57 | 10:41 | 24.77 | 22.81 | 11:43 |
| 2 | 27.75 | 23.11 | 20:59 | 23.19 | 18.68 | 20:32 | 27.54 | 22.94 | 20:53 | 25.27 | 24.36 | 21:49 |
| 4 | 26.88 | 23.42 | 39:31 | 22.05 | 17.64 | 42:25 | 27.47 | 22.13 | 42:01 | 26.35 | 25.37 | 43:21 |

| Batch | Hellwarrior | | | Mutant | | | Standup | | |
|---|---|---|---|---|---|---|---|---|---|
| | CLIP-S | CLIP-SIM | Train time | CLIP-S | CLIP-SIM | Train time | CLIP-S | CLIP-SIM | Train time |
| 1 | 24.58 | 23.52 | 10:17 | 29.76 | 21.58 | 10:43 | 24.55 | 21.43 | 10:28 |
| 2 | 25.40 | 24.53 | 20:56 | 29.76 | 21.99 | 20:08 | 24.21 | 21.42 | 21:18 |
| 4 | 26.02 | 25.85 | 39:37 | 29.43 | 21.86 | 39:57 | 23.02 | 20.70 | 39:17 |

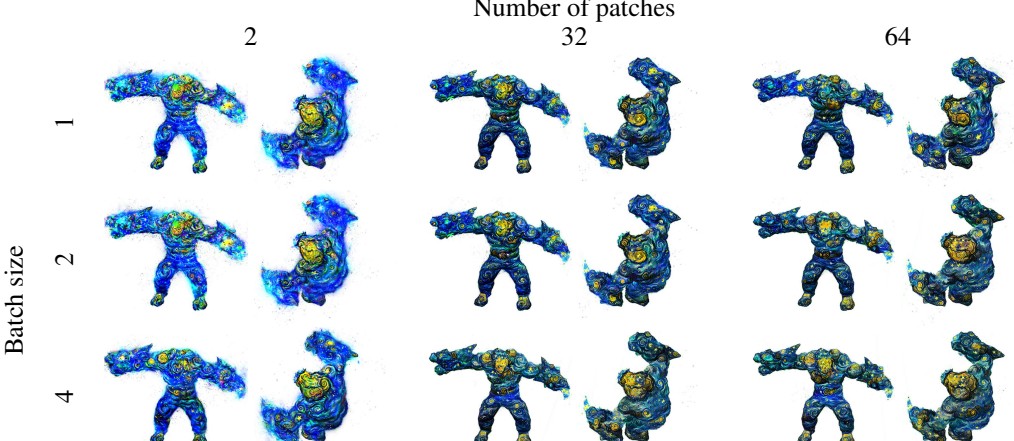

Figure 29: Effect of Batch Sizes and Number of patches using D-MiSo base model. Mutant from D-NeRF is stylized with "Starry Night by Vincent van Gogh".

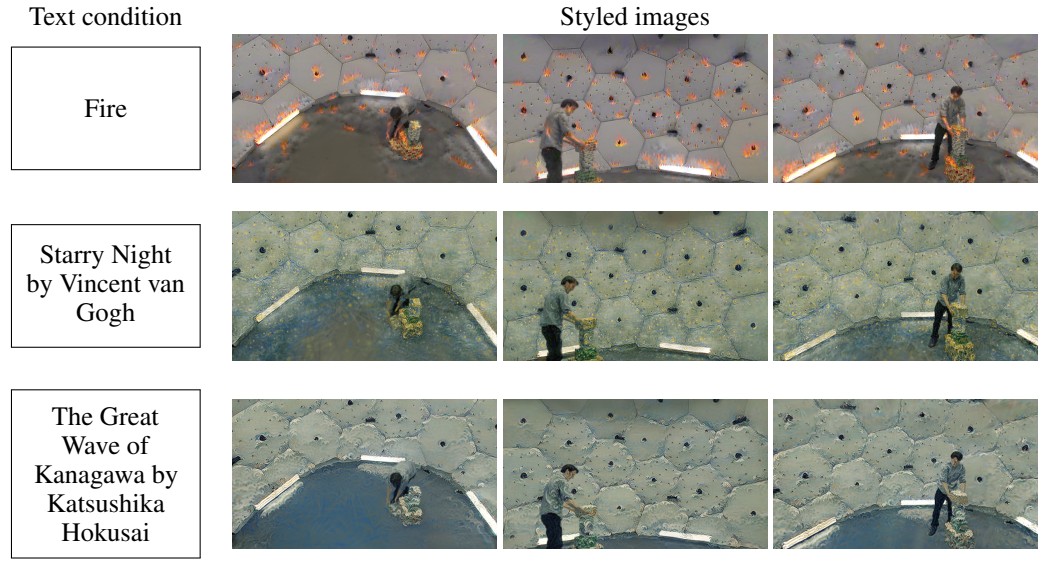

Figure 30: Qualitative style transfer results on samples from the PanopticSports dataset [68].

### B.3.3 Artifacts

The styling of objects may be incorrect or may not meet the user's visual needs, especially in concepts that are difficult to represent, such as *hope*, country names, *random* or *beauty*

Based on our experience, we have also identified common categories of failure cases mostly related to text prompts:

**Too general concepts, less known concepts** Since we are using CLIP representation, the concept should be well represented by CLIP embedding. We considered 3 prompts: We observed that for general prompts such as *"Impressionism"*, *"Painting by Claude Monet"* , *"Woman with a Parasol – Madame Monet and Her Son"*. Where: CosineSimilarity(prompt1, prompt2) = 0.802, CosineSimilarity(prompt1, prompt3) = 0.531, CosineSimilarity(prompt2, prompt3) = 0.382.

We observed that for general prompts such as *"Impressionism"*, the model tends to fill the background, and hollow spaces (e.g. between fingers). This improves the quantitative results, but in our opinion it decreases the visual perception of the stylization. We support this claim using the CLIP-S metric, the difference between choosing the $\lambda_{bg}$ parameter is shown in the Tab. 14 and Fig. 27, 31, 32.

The concepts of *"Impressionism"* and *"Painting by Claude Monet"* are well represented by the CLIP model. In addition, they are closely related, which is visually supported by the models trained using those prompts. On the other hand, concepts such as *"Woman with(...)"* are heavily influenced by details like as an umbrella, which spoils the visual effect in this case. This pattern is shown by the CLIP-SIM metric.

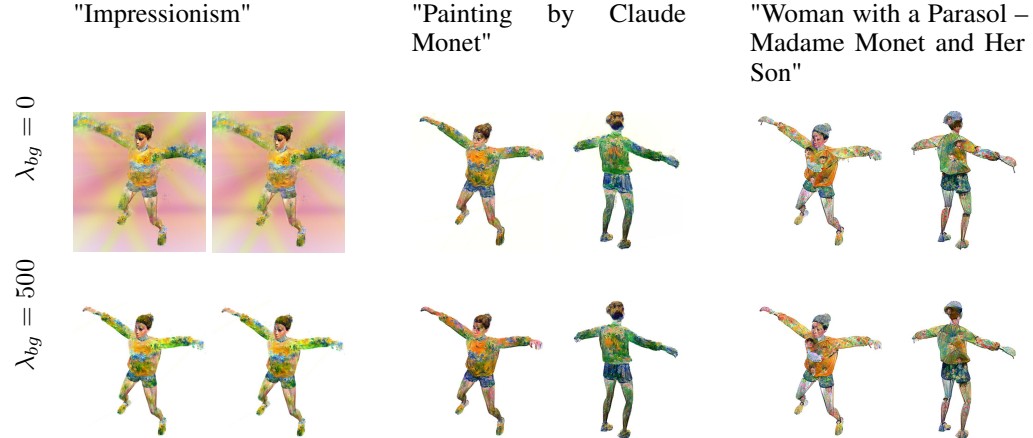

Figure 31: Visual comparison of object styling using three prompts, ranging from the most general to the most specific. A highly general prompt tends to produce background artifacts, which can be mitigated through background loss. Conversely, an overly specific prompt may be poorly represented in the CLIP space, leading to features such as face-like patterns (suggesting *Madame*) or umbrella-like shapes.

Table 14: The impact of prompts on the quality of styling using the CLIP-SIM and CLIP-Smetric. prompt1: *"Impressionism"*, prompt2: *"Painting by Claude Monet"*, prompt3: *"Woman with a Parasol – Madame Monet and Her Son"*.

| *Jumpinjacks* | prompt1 | | prompt2 | | prompt3 | |
|---|---|---|---|---|---|---|
| | CLIP-S | CLIP-SIM | CLIP-S | CLIP-SIM | CLIP-S | CLIP-SIM |
| $\lambda_{bg} = 0$ | 23.04 | 18.63 | 17.77 | 12.18 | 18.97 | 10.88 |
| $\lambda_{bg} = 500$ | 21.51 | 16.02 | 18.02 | 13.45 | 18.35 | 10.41 |
| $\lambda_{bg} = 1000$ | 21.81 | 16.59 | 18.20 | 14.27 | 18.52 | 10.65 |

**Length and detail of the general concept prompts** We conducted an experiment using *Jumpingjacks* in which we considered three prompt lengths, see Tab. 15. We noticed that if a shorter prompt is considered, the CLIP model finds a certain representation of a specific word/short prompt related to

a general concept, see Fig. 3 If a longer prompt is used, we can expect an averaged representation of the concepts found in the prompt, which may cause some inaccuracies in the stylized image, see Fig. 32.

In both cases, the styling emphasizes artifacts, especially in the background. We can mitigate the artifacts in the background using $\lambda_{bg}$ and $patch\_size$. This is particularly evident in detailed areas, e.g. on the hands (see Fig. 27, 28). Visual assessment is very difficult and subjective.

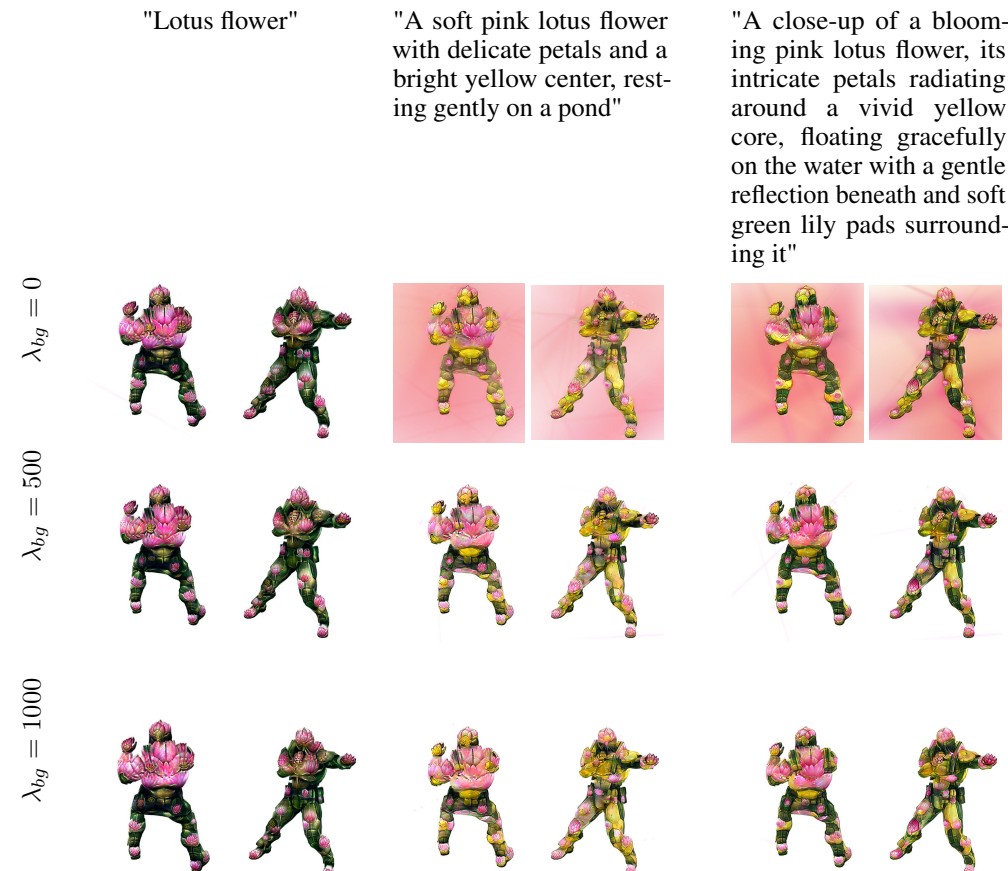

Figure 32: Visual comparison of the impact of length and detail of the prompt on the style. Using a more detailed prompt tends to create artifacts in the background. It is worth noting that some areas tend to change more, e.g. a lotus almost always appeared on the *Hook*'s back. When prompt is too long we observe inaccurate with a good representation using a clip.

Table 15: The impact of length and detail of the overall concept of the prompt. prompt1: *"Lotus flower"*, prompt2: *"A soft pink lotus flower with delicate petals and a bright yellow center, resting gently on a pond"*; prompt3: *"A close-up of a blooming pink lotus flower, its intricate petals radiating around a vivid yellow core, floating gracefully on the water with a gentle reflection beneath and soft green lily pads surrounding it"*.

| *Jumpinjacks* | prompt1 | | prompt2 | | prompt3 | |
|---|---|---|---|---|---|---|
| | CLIP-S | CLIP-SIM | CLIP-S | CLIP- SIM | CLIP-S | CLIP-SIM |
| $\lambda_{bg} = 0$ | 24.97 | 22.24 | 23.36 | 19.57 | 24.03 | 21.88 |
| $\lambda_{bg} = 500$ | 24.85 | 20.78 | 19.46 | 15.87 | 20.34 | 19.15 |
| $\lambda_{bg} = 1000$ | 24.71 | 20.82 | 19.25 | 15.81 | 20.90 | 19.00 |

### B.3.4 Time

The D-NeRF dataset provide GT as only one time point for each camera. For a set of time steps $0.1 \cdot i \mid i \in \{0, 1, 2, \ldots, 10\}$, we used the test cameras from NeRF-Synthetic (200 views) to generate reference images using the base D-MiSo model. Then, we generated the corresponding stylized images and computed the mean CLIP-F and CLIP-CONS metrics for time steps, see. Tab 16. From this experiment, we can see that it is difficult to indisputably determine which styling is best based on the selected metrics.

Table 16: Numerical comparison of the influence of batch on the quality of styling using CLIP-F and CLIP-CONS metrics. We used default parameters and the prompt *"Starry Night by Vincent van Gogh"* for the experiment. D-Miso only means reconstructions of the 4D object.

| dataset | Hook | Jumpingjacks | Trex | Bouncingballs | Hellwarrior | Mutant | Standup |
|---|---|---|---|---|---|---|---|
| CLIP-F | | | | | | | |
| batch=1 | 97.56 | 98.41 | 97.36 | 98.73 | 98.41 | 98.01 | 98.03 |
| 2 | 97.37 | 98.15 | 97.34 | 98.66 | 98.23 | 97.95 | 97.87 |
| 4 | 97.66 | 98.35 | 97.68 | 98.33 | 98.26 | 97.65 | 98.19 |
| CLIP-CONS | | | | | | | |
| 1 | 2.1 | 2.31 | 2.25 | 0.66 | 5.64 | 1.78 | 2.60 |
| 2 | 1.7 | 2.80 | 3.24 | 1.03 | 5.72 | 1.92 | 3.02 |
| 4 | 2.7 | 2.97 | 2.97 | 0.53 | 6.29 | 2.43 | 3.68 |

For each test camera from the D-NeRF dataset, we generated stylized images at $0.05 \cdot i \mid i \in \{0, 1, 2, \ldots, 20\}$ time points. Next, we calculate the average metrics to evaluate the temporal consistency over time for each camera; see the Tab. 17. An interesting observation is that in almost every case, when using selected hyperparameters, we see an increase in metrics compared to the base model. Fig. 33 presents a visual comparison of objects with respect to camera angle and time. The results show that the styling remains consistent and that temporal variations do not introduce artifacts, as long as the object is accurately reconstructed.

Table 17: A numerical comparison of the effect of batch size on temporal consistency over time for each camera, using the prompt *"Starry Night by Vincent van Gogh"*.

| dataset | Hook | Jumpingjacks | Trex | Bouncingballs | Hellwarrior | Mutant | Standup |
|---|---|---|---|---|---|---|---|
| Short-range consistency: LPIPS | | | | | | | |
| 1 | 0.017 | 0.019 | 0.012 | 0.013 | 0.023 | 0.007 | 0.009 |
| 2 | 0.017 | 0.018 | 0.012 | 0.012 | 0.025 | 0.007 | 0.010 |
| 4 | 0.017 | 0.017 | 0.012 | 0.013 | 0.026 | 0.007 | 0.009 |
| D-MiSo | 0.009 | 0.011 | 0.008 | 0.007 | 0.018 | 0.006 | 0.006 |
| Short-range consistency: RMSE | | | | | | | |
| 1 | 0.046 | 0.061 | 0.046 | 0.021 | 0.055 | 0.028 | 0.031 |
| 2 | 0.045 | 0.058 | 0.044 | 0.020 | 0.056 | 0.028 | 0.032 |
| 4 | 0.044 | 0.055 | 0.043 | 0.021 | 0.057 | 0.028 | 0.031 |
| D-MiSo | 0.023 | 0.036 | 0.029 | 0.018 | 0.043 | 0.017 | 0.018 |
| Long-range consistency: LPIPS | | | | | | | |
| 1 | 0.028 | 0.062 | 0.049 | 0.043 | 0.027 | 0.044 | 0.031 |
| 2 | 0.025 | 0.063 | 0.050 | 0.043 | 0.026 | 0.043 | 0.030 |
| 4 | 0.026 | 0.063 | 0.048 | 0.040 | 0.027 | 0.039 | 0.029 |
| D-MiSo | 0.016 | 0.046 | 0.031 | 0.028 | 0.020 | 0.031 | 0.021 |
| Long-range consistency: RMSE | | | | | | | |
| 1 | 0.049 | 0.043 | 0.031 | 0.028 | 0.062 | 0.027 | 0.044 |
| 2 | 0.050 | 0.043 | 0.030 | 0.025 | 0.063 | 0.026 | 0.043 |
| 4 | 0.048 | 0.040 | 0.029 | 0.026 | 0.063 | 0.027 | 0.039 |
| D-MiSo | 0.031 | 0.028 | 0.021 | 0.016 | 0.046 | 0.020 | 0.031 |

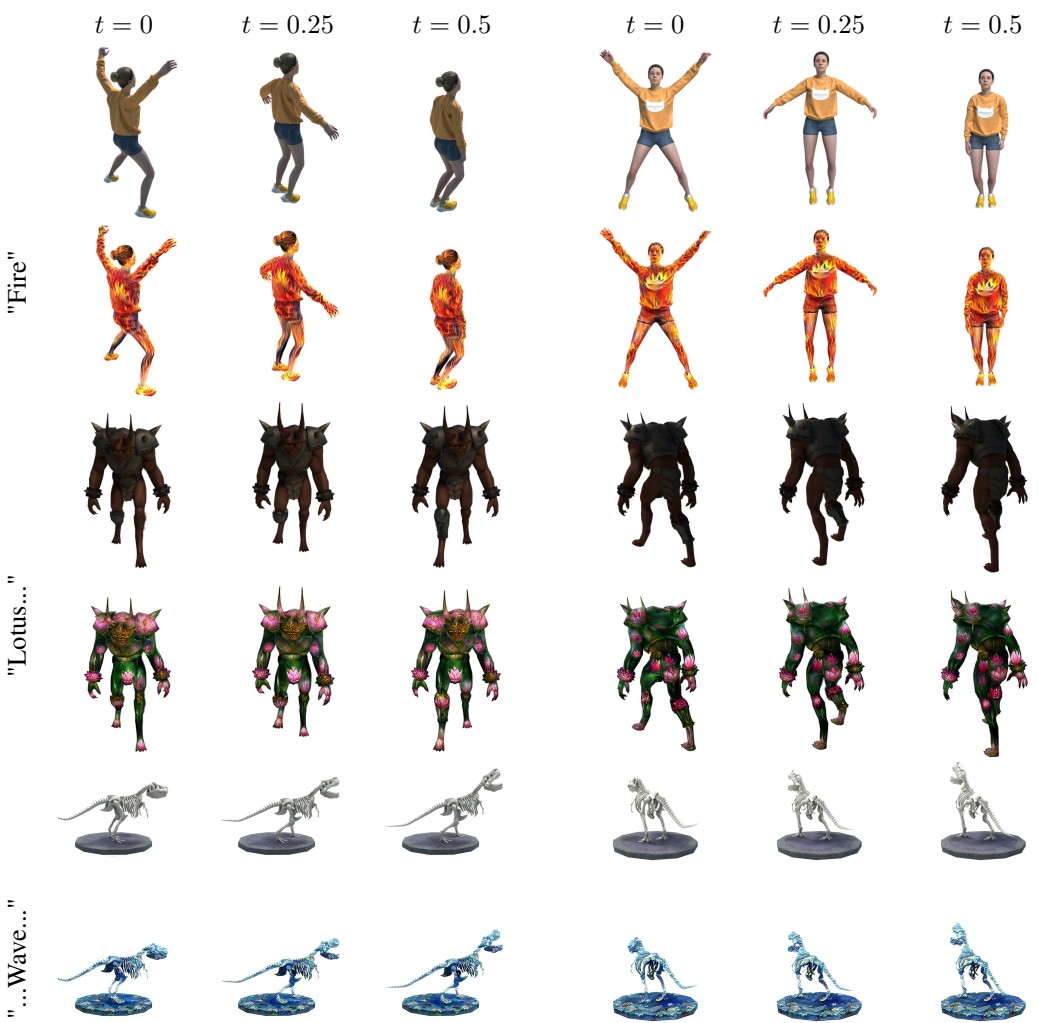

Figure 33: Visual comparison of objects across camera angles and time, demonstrating consistent styling. We used $\lambda_{bg} = 1000$, batch=4.

## B.4  2D: Additional Results and Explanation

This section presents additional qualitative results for 2D style transfer. Fig. 34 shows a qualitative comparison with similar text-based methods: CLIPstyler [34] and FastCLIPstyler [35]. Fig. 35 presents an additional comparison with AdaIN [22] and StyTr$^2$ [33]. CLIPGaussian is a competitive alternative to these methods. We see that our method mainly focuses on details.

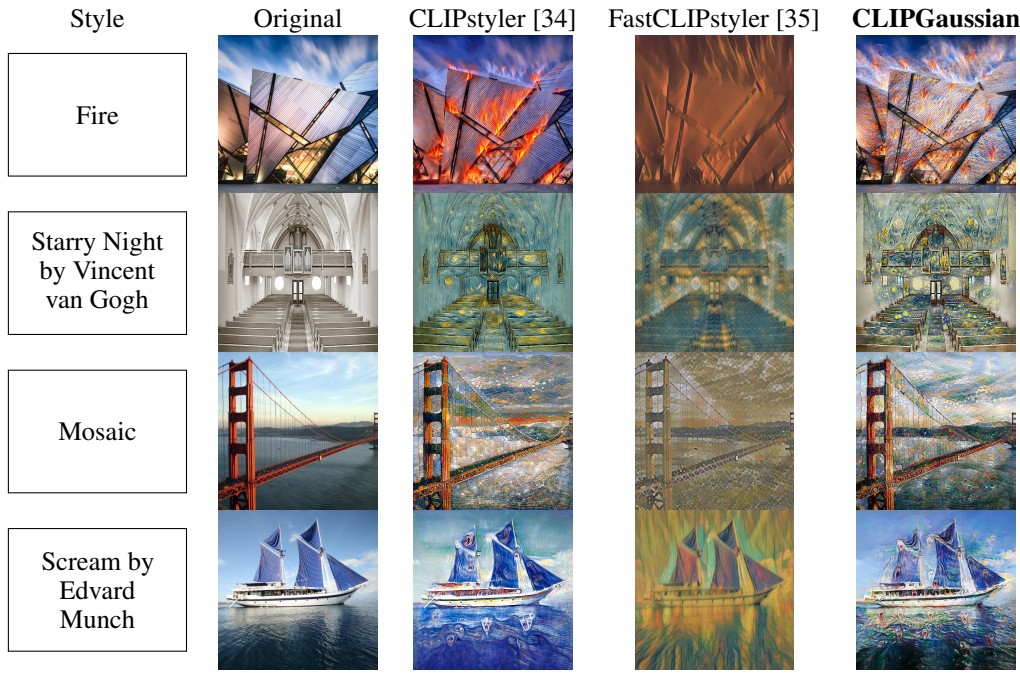

Figure 34: Comparison of image style transfer using text condition.

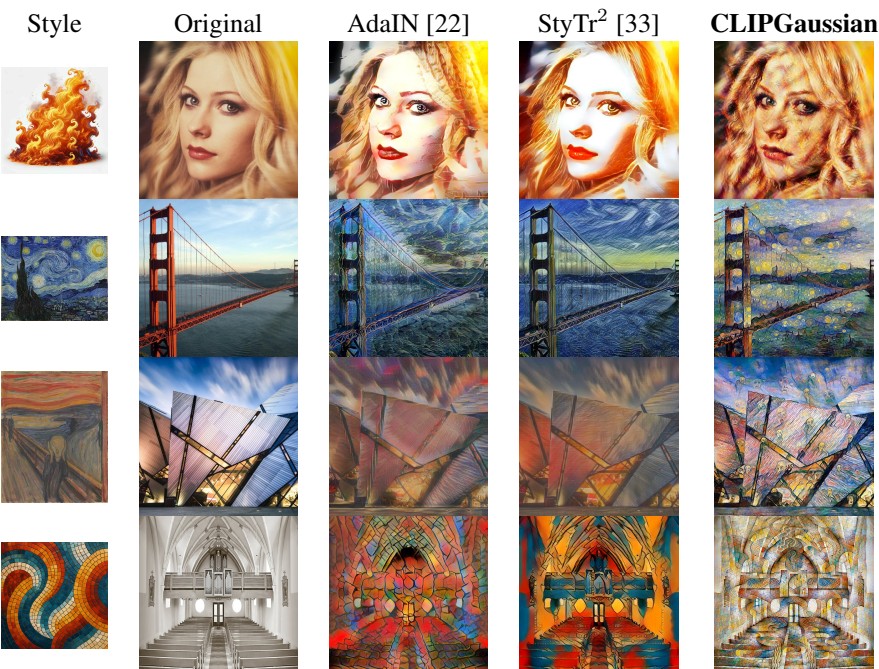

Figure 35: Comparison of image style transfer using image condition.

### B.5 Video: Additional Results and Explanation

This section provides a detailed description of the quantitative evaluation setup.

During experiments we used four videos (*camel*, *bear*, *train* and *blackswan*) from the DAVIS dataset [63] and four videos (*beauty*, *bospho*, *shake* and *yacht*) from the UVG dataset [64]. For DAVIS dataset we used videos with resolution $854\text{px} \times 480\text{px}$ and for UVG we downscaled videos to $960\text{px} \times 540\text{px}$. Similarly to evaluation of 3D style transfer, each was stylized using four text prompts ("Fire", "Mosaic", "Starry Night by Vincent van Gogh", and "Scream by Edvard Munch") and four style images (same as in Fig. 36).

As Text2Video [42] and RerenderAVideo [45] employ different prompt templates, we use "Make it {style} style" for Text2Video and "A {object} in {style} style" for RerenderAVideo, where `style` is a CLIPGaussian style prompt and `object` is a video name i.e. *bear*, *blackswan*, *camel* or *train*. Baseline methods were trained using their default configurations. CLIPGaussian models were trained according to the settings described in Appendix A. On average CLIPGaussian took around 11 minutes to style a video from DAVIS dataset.

We provide example comparisons on a test subset for image (Fig. 36) and text (Fig. 37) conditioning. Quantitative evaluations using CLIP-based metrics are presented for the DAVIS (Tab. 18) and UVG (Tab. 19) datasets. Furthermore, consistency is evaluated using RMSE and LPIPS (Tab. 20 and Tab. 21), while temporal consistency is measured using Farneback optical flow at different time intervals (Tab. 22).

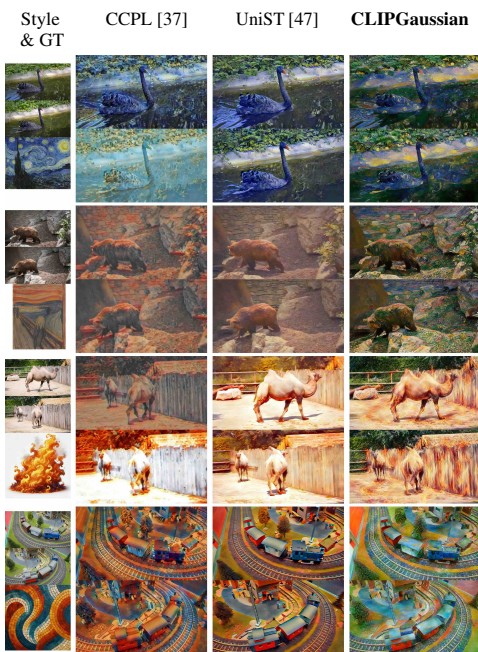

Figure 36: Comparison of video style transfer using image condition on DAVIS dataset [63].

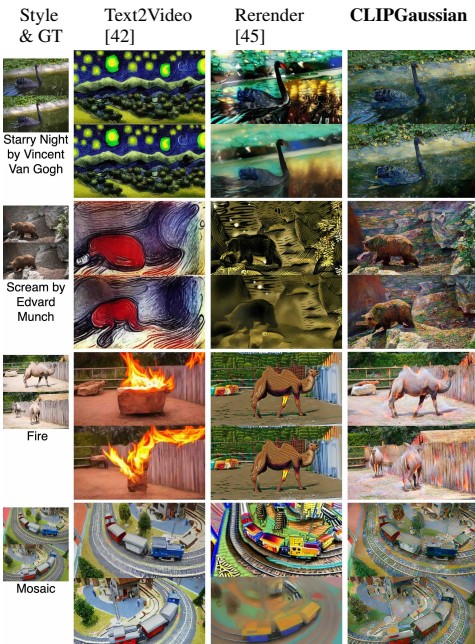

Figure 37: Comparison of video style transfer using text condition on DAVIS dataset [63].

Table 18: Quantitative comparison of video style transfer on the DAVIS dataset [63], using CLIP-based metrics. Larger values are better.

| Model | CLIP-S ↑ | CLIP-SIM ↑ | CLIP-F ↑ | CLIP-CONS ↑ |
|---|---|---|---|---|
| *Text-conditioned* | | | | |
| Rerender [45] | 19.40 | 9.83 | 98.23 | -0.03 |
| Text2Video [42] | 26.05 | **24.99** | 93.63 | 0.03 |
| **CLIPGaussian** | **26.25** | 24.53 | **99.00** | **1.92** |
| *Image-conditioned* | | | | |
| ViSt3D [41] | 55.92 | 2.75 | 99.18 | 3.28 |
| AdaAttN [39] | 57.71 | -1.04 | 97.56 | 1.08 |
| ReReVST [69] | 61.50 | 2.12 | 98.85 | 2.51 |
| UniST [47] | 65.93 | 3.85 | 99.36 | 5.16 |
| CCPL [37] | 68.89 | 8.20 | 97.92 | -0.02 |
| **CLIPGaussian-Light** | 63.60 | 8.06 | **99.50** | **11.99** |
| **CLIPGaussian** | **74.31** | **17.60** | 99.18 | 1.27 |

Table 19: Quantitative comparison of video style transfer on the UVG dataset [64], using CLIP-based metrics. Larger values are better.

| Model | CLIP-S ↑ | CLIP-SIM ↑ | CLIP-F ↑ | CLIP-CONS ↑ |
|---|---|---|---|---|
| *Text-conditioned* | | | | |
| Rerender [45] | 14.08 | 7.11 | 74.38 | 0.01 |
| Text2Video [42] | 19.02 | 13.18 | 70.51 | 0.18 |
| **CLIPGaussian** | **23.88** | **18.38** | **100.39** | **18.38** |
| *Image-conditioned* | | | | |
| ViSt3D [41] | 55.98 | 0.70 | 99.54 | 3.78 |
| AdaAttN [39] | 56.27 | 1.24 | 98.11 | 1.22 |
| ReReVST [69] | | Out of memory | | |
| UniST [47] | 55.75 | 2.04 | 99.44 | **5.29** |
| CCPL [37] | 59.14 | 2.19 | 98.50 | 1.45 |
| **CLIPGaussian-Light** | 63.83 | 10.61 | 100.38 | 4.25 |
| **CLIPGaussian** | **71.64** | **16.68** | **100.42** | 0.94 |

Table 20: Quantitative comparison of video style transfer consistency on the DAVIS dataset [63], using RMSE and LPIPS. Smaller values are better.

| Model | Short-range consistency | | Long-range consistency | |
|---|---|---|---|---|
| | LPIPS ↓ | RMSE ↓ | LPIPS ↓ | RMSE ↓ |
| Original Videos | 0.042 | 0.034 | 0.070 | 0.055 |
| *Text-conditioned* | | | | |
| Rerender [45] | **0.062** | **0.040** | **0.132** | **0.077** |
| Text2Video [42] | 0.261 | 0.183 | 0.235 | 0.166 |
| **CLIPGaussian** | 0.084 | 0.057 | 0.152 | 0.095 |
| *Image-conditioned* | | | | |
| ViSt3D [41] | 0.081 | 0.043 | 0.121 | 0.063 |
| AdaAttN [39] | 0.087 | 0.059 | 0.116 | 0.083 |
| ReReVST [69] | 0.072 | 0.047 | 0.100 | 0.068 |
| UniST [47] | 0.062 | 0.047 | 0.088 | 0.066 |
| CCPL [37] | 0.102 | 0.065 | 0.132 | 0.093 |
| **CLIPGaussian-Light** | **0.049** | **0.035** | **0.083** | **0.061** |
| **CLIPGaussian** | 0.086 | 0.057 | 0.157 | 0.090 |

Table 21: Quantitative comparison of video style transfer consistency on the UVG dataset [64], using RMSE and LPIPS. Smaller values are better.

| Model | Short-range consistency | | Long-range consistency | |
|---|---|---|---|---|
| | LPIPS ↓ | RMSE ↓ | LPIPS ↓ | RMSE ↓ |
| Original Videos | 0.045 | 0.025 | 0.068 | 0.049 |
| *Text-conditioned* | | | | |
| Rerender [45] | 0.027 | 0.025 | 0.064 | **0.049** |
| Text2Video [42] | 0.223 | 0.153 | 0.150 | 0.105 |
| **CLIPGaussian** | **0.017** | **0.018** | **0.053** | 0.054 |
| *Image-conditioned* | | | | |
| ViSt3D [41] | 0.047 | 0.031 | 0.103 | 0.059 |
| AdaAttN [39] | 0.056 | 0.046 | 0.096 | 0.075 |
| ReReVST [69] | | Out of memory | | |
| UniST [47] | 0.043 | 0.038 | 0.077 | 0.065 |
| CCPL [37] | 0.072 | 0.052 | 0.106 | 0.081 |
| **CLIPGaussian-Light** | **0.019** | **0.017** | **0.049** | **0.044** |
| **CLIPGaussian** | 0.020 | 0.019 | 0.060 | 0.056 |

Table 22: Quantitative comparison of video style transfer consistency on the DAVIS dataset [63], using mean absolute difference of Farneback optical Flow, for different intervals (number of frames - $k$).

| Model / $k$ | 1 | 2 | 4 | 8 | 16 |
|---|---|---|---|---|---|
| *Text-conditioned* | | | | | |
| Rerender [45] | 1.08 | 2.07 | 4.40 | 7.95 | 11.34 |
| Text2Video [42] | 5.06 | 5.91 | 7.68 | 9.93 | 11.34 |
| **CLIPGaussian** | **0.41** | **0.79** | **2.02** | **4.00** | **6.58** |
| *Image-conditioned* | | | | | |
| ViSt3D [41] | 0.75 | 1.19 | 2.30 | 3.59 | 4.54 |
| AdaAttN [42] | 0.69 | 1.09 | 2.57 | 3.86 | 5.26 |
| ReReVST [69] | 0.46 | 0.84 | 2.30 | 3.43 | 4.94 |
| UniST [47] | 0.43 | 0.73 | 1.75 | 3.21 | 4.98 |
| CCPL [37] | 1.40 | 2.36 | 4.62 | 6.86 | 8.45 |
| **CLIPGaussian-Light** | **0.20** | **0.35** | **0.83** | **1.46** | **2.32** |
| **CLIPGaussian** | 0.41 | 0.79 | 2.12 | 4.06 | 7.00 |

