# OpenReview forum: "CLIPGaussian: Universal and Multimodal Style Transfer Based on Gaussian Splatting"
_NeurIPS.cc/2025/Conference — NeurIPS 2025 poster_

### Official Review · Reviewer_xZpT · 2025-06-21

**Clarity:** 2
**Significance:** 2
**Originality:** 2
**Rating:** 4
**Confidence:** 4

**Summary:**

The paper introduces CLIPGaussian, which leverages 3D Gaussian splatting (3DGS) as its base model to edit 3D, 4D, and videos based on a simple text prompt or a style image.  The authors achieve this goal by jointly optimizing all the 3D Gaussian attributes. Additionally, the authors include various training supervision such as content loss, directional CLIP loss, patch CLIP loss, and background loss to help obtain the final edited results. Both qualitative and quantitative experiments have been conducted to validate the performance and effectiveness of the proposed method.

**Questions:**

The reviewer am convinced by the quality of the results. However, the reviewer still doesn't get the main contribution of the paper or how to obtain the results based on the materials provided in the submission. More analysis about (1) how each component will affect the results, (2) their motivations, and (3) the differences between previous methods will be needed. From the perspective of the reviewer, it's more like a unified model of Instruct-GS2GS.

Therefore, the reviewer would rate the paper as ''borderline accept'' at the current stage.

**Ethical Concerns:**

["NO or VERY MINOR ethics concerns only"]

**Final Justification:**

The rebuttal addresses most of my concerns. I would like to keep my original rating and encourage the authors to check the issues mentioned in the comments/conversations below.

**Limitations:**

Limitations have been discussed in the paper, while the paper doesn't provide potential societal impact.

**Quality:**

3

**Strengths And Weaknesses:**

Strengths:

+ The results produced by the proposed method demonstrate superior quality over existing methods.

+ The paper is well-written and easy to follow.

Weaknesses:

- Lack many evaluations. (1) While the CLIPGaussian is claimed to perform well for 3D, 4D, and videos, the authors provide comparisons with just the very baseline in each field. More comparisons with more recent methods would be required to demonstrate the performance. (2) Lack of sufficient quantitative evaluations. The two quantitative evaluations are provided in Tables 1 and 2. However, the compared methods are limited. And, the results are not obviously better than existing methods. As for the user studies provided in the appendix, maybe it's also beneficial to rate based on the quality. (3) Ablation studies regarding the loss terms would also be required.

- The introduction is a little bit confusing. The authors introduce texturing, local editing, insertion, and style transfer. It would lead to a conclusion that the proposed method can address these four tast with the proposed plug-in type models. However, the paper only shows  results with style transfer.

---

> ### Author Rebuttal · Authors · 2025-07-30
>
> Thank you to the reviewer for pointing out inaccuracies and offering suggestions. We’re glad the work is considered interesting and easy to follow. We hope our responses and revisions will improve clarity for future readers.
>
> **W1 [More comparisons]:**
>
> We thank the reviewer for the suggestion. We conducted additional comparison with **SGSST** (CVPR 25) and **ABC-GS** (ICME 25) (for 3D) as well as **ViSt3D** and **ReReVST** (for video).  Moreover, we have included results obtained using CLIPGaussian with an alternative set of hyperparameters (lambda_p​=35, feature_lr = 0.002), which produces slightly lighter stylization. We call this variant CLIPGaussian-Lite. As a 4D style transfer is a new field, we already compared CLIPGaussian with all relevant baselines. The updated results for 3D and Video are shown in the tabs below.
>
> **TABLE xZpT.A**
> |Model|CLIP-S|CLIP-SIM|CLIP-F|CLIP-CONS|Memory Size|
> |-----|:----|:----|:----|:----|:----|
> |||Text-conditioned||||
> |I-GS2GS|16.80|12.03|99.19|**13.54**|-36%|
> |DGE|17.59|12.27|**99.31**|12.46|-5%|
> |**CLIPGaussian-Lite**|23.14|22.30|99.17|8.51|0%|
> |**CLIPGaussian**|**26.86**|**26.31**|98.80|2.34|0%|
> |||Image-conditioned||||
> |StyleGaussian|63.69|13.07|98.87|1.36|0%|
> |SGSST|66.57|16.24|97.54|0.91|0%|
> |ABC-GS|68.68|16.29|99.10|2.11|0%|
> |G-Style|**76.94**|**24.94**|98.94|1.31|+126%|
> |**CLIPGaussian-Lite**|69.27|17.02|**99.16**|**5.26**|0%|
> |**CLIPGaussian**|72.65|20.72|98.78|1.77|0%|
>
> CLIPGaussian, in both the base and light variants, outperforms SGSST and ABC-GS in stylization quality metrics (CLIP-S and CLIP-SIM). Notably, CLIPGaussian-Light achieves the highest scores across all consistency metrics.  It is important to highlight that G-Style significantly increases the model size; typically by more than a factor of two, whereas CLIPGaussian offers competitive results without altering the underlying model size.
>
> **TAB. xZpT.B**
>
> |Model|CLIP-S|CLIP-SIM|CLIP-F|CLIP-CONS|
> |-----|:----|:----|:----|:----|
> |||Text-conditioned|||
> |Rerender|19.40|9.83|98.23|-0.03|
> |Text2Video|26.05|**24.99**|93.36|0.03|
> |**CLIPGaussian**|**26.25**|24.53|**99.00**|**1.92**|
> |||Image-conditioned|||
> |CCPL|18.89|8.20|97.92|-0.02|
> |UniST|15.93|3.85|99.36|5.16|
> |ViSt3D|55.92|2.75|99.18|3.28|
> |ReReVST|61.50|2.12|98.85|2.51|
> |**CLIPGaussian-Light**|63.60|8.06|**99.50**|**11.99**|
> |**CLIPGaussian**|**74.31**|**17.60**|99.18|1.27|
>
> CLIPGaussian also outperforms ViSt3D and ReReVST. Additionally version with lighter stylization has the best consistency while still outperforming baselines.
>
> **W2 [Lack of sufficient quantitative evaluations]:**
> Please find quantitative evaluation for 3D in: Tabs. 1-12, 4D: Tab. 13, Video: Tabs. 14, 15. However, as suggested by reviewers **8XY8** and **XWe5**, we conducted additional quantitative evaluation using flow-based consistency metrics for 3D, Video and 4D as well as CLIP-based consistency metrics for 4D.
>
> **TAB. xZpT.C**
>
> |Model|Short-range consistency||Long-range consistency||
> |-----|:----|:----|:----|:----|
> ||LPIPS|RMSE|LPIPS|RMSE|
> |OriginalVideos|0.053|0.047|0.140|0.123|
> |||Text-conditioned|||
> |I-GS2GS|**0.048**|**0.043**|0.148|0.131|
> |DGE|0.055|0.044|0.148|0.120|
> |**CLIPGaussian-Lite**|0.051|**0.043**|**0.132**|**0.117**|
> |**CLIPGaussian**|0.057|0.051|0.148|0.125|
> |||Image-conditioned|||
> |StyleGaussian|0.052|0.061|0.145|0.144|
> |SGSST|**0.040**|0.050|**0.139**|0.142|
> |ABC-GS|0.048|**0.043**|0.141|0.126|
> |G-Style|0.054|0.049|0.145|0.136|
> |**CLIPGaussian-Lite**|0.055|0.045|**0.139**|**0.120**|
> | **CLIPGaussian** | 0.064 | 0.055 | 0.162 | 0.132 |
>
> **TAB. xZpT.D**
>
> |Model|Short-range consistency||Long-range consistency||
> |-----|:----|:----|:----|:----|
> ||LPIPS|RMSE|LPIPS|RMSE|
> |OriginalVideos|0.042|0.034|0.070|0.055|
> |||Text-conditioned|||
> |Rerender|0.062|0.040|0.132|0.077|
> |Text2Video|0.261|0.183|0.235|0.166|
> |**CLIPGaussian-Light**|**0.049**|**0.036**|**0.085**|**0.061**|
> |**CLIPGaussian**|0.084|0.057|0.152|0.095|
> |||Image-conditioned|||
> |CCPL|0.102|0.065|0.132|0.093|
> |UniST|**0.062**|0.047|0.088|0.066|
> |ViSt3D|0.081|0.043|0.121|0.063|
> |ReReVST|0.072|0.047|0.100|0.068|
> |**CLIPGaussian-Lite**|0.049|**0.035**|**0.083**|**0.061**|
> |**CLIPGaussian**|0.086|0.057|0.157|0.098|
>
> In response to the reviewer XWe5, we evaluate consistency of CLIPGaussian in 4D setting (see Tabs. XWe5.B-D).
>
> It is important to note that the interpretation of style transfer metrics is not trivial. CLIP-S and CLIP-SIM scores would be maximized if the generated object were identical to the style image or prompt, which would clearly be an undesirable outcome. On the other hand, consistency metrics are maximized when the output is identical to the original object, which also indicates poor stylization. This trade-off can be easily seen in Tab. 11. We believe that CLIPGaussian strikes a meaningful balance between these two objectives, achieving high-quality style transfer without compromising the consistency of the object or video. Another important factor, which is not measured by any metrics in universality. All other methods work on only one medium (3D, 4D, 2D or video) and support only one modality (text or image). CLIPGaussian is the first method that works in all of these settings while being either superior or at least comparable in quality.
>
> In case of user study, we evaluated “quality” or “visual appeal” in the Question 2: "On a scale from "Very Low" to "Very High", how would you rate the visual appeal of each generated result?”. We will highlight it in the revised version of the main manuscript.
>
> **Q0 Main contribution of the paper**
>
> Our main contribution is a unified framework for various data modalities such as 2D, 3D, 4D, and videos etc. This broad applicability was positively highlighted by the reviewers (e.g., Rev. ed8o and Rev. XWe5). Our framework demonstrates competitive performance across modalities, which can be especially seen in the case of 3D and 4D, where it outperforms existing approaches. We acknowledge that our performance is not yet on par with current state of the art specialized approaches in the case of images, which are often based on large diffusion models or ChatGPT-like architectures. In contrast, our method is the first to propose a unified multimodal GS based style transfer approach.
>
> **Q1/ Q2 Motivation of each component and its effect**
>
> CLIPGaussian operates by directly optimizing gaussians (either in 3D, 4D, Video or Images) using losses based on CLIP embeddings. Analysis of loss terms can be found in the Appendix B.2.3 **(lines 722-736)**. Description of the losses can be found in Section 3 **(lines 217-252)**. The loss consists of four main parts:
>
> 1. **Content Loss (Eq. 2)**: This is a classical loss computed using VGG-19 embeddings. During each optimization step, it encourages the generated view to preserve the content of the ground truth. In the case of video, it ensures that each generated frame retains the original semantic content; e.g., that the subject remains a camel or a bear (see Figs.12 and 13).
> 2. **Directional CLIP Loss (Eq. 3):** This component is responsible for global style transfer. It compares the CLIP embedding of a rendered view with that of the corresponding original view, both conditioned on the style prompt or style image. This encourages the overall appearance of the object to align with the desired style.
> 3. **Patch CLIP Loss (Eq. 4):** This loss is equivalent to the Directional loss but is applied to random patches of the rendered view rather than the entire image. It ensures that each patch of the rendered view is stylized. This loss is responsible for the texture of the stylized object, as it operates on smaller fragments.
> 4. **Background loss (bg loss):** This loss is applied only for 3D and 4D objects. In these cases, we aim to stylize the object itself without affecting its bg. For example, this loss ensures that the white bg behind the Lego set in Fig. 6 remains unchanged. In contrast, StyleGaussian does not enforce bg consistency on the same image.
>
> **W3 [Ablation studies regarding the loss terms]:**
> We have already conducted an ablation study in the context of 3D stylization. The results are presented in Fig. 24 and Tabs. 12 and 13 of the manuscript. The parameters `lambda_d` and `lambda_p` correspond to the weights of their respective loss terms (directional loss and patch loss). In both cases, we examined the effect of setting these weights to zero. The effect of the bg loss can be seen on Fig. 26. We will highlight this fact in the manuscript. We believe that this will improve understanding of losses.
>
> **Q3 [The differences between previous methods will be needed] Instruct-GS2GS:**
>
> CLIPGaussian is not a unified version of Instruct-GS2GS (I-GS2GS). While I-GS2GS stylizes training images using a diffusion model before fine-tuning the Gaussian splats, CLIPGaussian directly optimizes the Gaussian representation using CLIP-based losses. This allows high-quality style transfer with both image and text conditioning, unlike I-GS2GS, which supports only text prompts. Additionally, CLIPGaussian does not rely on any pretrained diffusion models  and is applicable across modalities, whereas I-GS2GS is limited to 3D. A direct comparison with I-GS2GS is provided in Fig. 19 and 20. We will add a description of I-GS2GS to the related work section. Thank you for pointing out this omission.
>
> **L1 [Social Impact]:**
>
> ClipGaussian enables cross-modal generality as a plugin based on GS models. Using CLIP as a multimodal vision and language model motivated  it to enable high-quality stylization and expanding creative possibilities. Gaussian based representation enables the creation of high-quality real-time renders which is a key challenge in computer vision, especially in broad applications in AR, gaming, and digital content creation. We will add this to manuscripts.
>
> We are grateful for your careful review, which contributed to improving the clarity of our paper. If anything remains unclear, we would be glad to answer any questions.

---

> > ### Comment · Reviewer_xZpT · 2025-08-01
> >
> > Thanks for the rebuttal from the authors. However,
> >
> > (1) The quantitative comparisons provided in the rebuttal are not very convincing to demonstrate CLIPGaussian's performance.
> >
> > (2) I check the qualitative comparison regarding the ablation in Fig.24. However, I find that the ablation comparison is obvious for \$\lambda_p\$, but not \$\lambda_d\$. For example, when \$\lambda_p\$ is 90 or 180, different \$\lambda_d\$ values (0, 5, 10) don't make many differences.
> >
> > The quantitative results in Table 12 are not obvious either. The qualitative comparison makes me curious about the detailed setting for experiments in Table 12.
> >
> > (3) It seems tha the authors haven't answered my question regarding this concern:
> > > The introduction is a little bit confusing. The authors introduce texturing, local editing, insertion, and style transfer. It would lead to a conclusion that the proposed method can address these four tast with the proposed plug-in type models. However, the paper only shows results with style transfer.
> >
> > Overall, I am currently not positive regarding this paper

---

> > > ### Author Response · Authors · 2025-08-01
> > >
> > > **(1)**
> > >
> > > **3D**: Table A demonstrates CLIPGaussian performance in 3D object stylization. The only other method that has better stylization than CLIPGaussian or CLIPGaussian-Light (in terms of CLIP-S and CLIP-SIM metrics) is G-Style which increases the size of a model more than two times. In the text-conditioned setting, we always observe improved visual quality and better metric results (see Fig. 19).
> > >
> > > Additionally table C showcases that CLIPGaussian-Light outperforms all methods in terms of long-range consistency while matching original data in short-range consistency (which is also measured by CLIP-CONS and CLIP-F metrics from Table A). That shows that CLIPGaussian (especially in the Light setting) outperforms all other baselines besides G-Style which has other, significant drawbacks.
> > >
> > > **Video**: Moreover, Tables B and D show that CLIPGaussian-Light outperforms all other methods both in stylization and consistency in case of image conditioning. In case of text condition CLIPGaussian matches performance of Text2Video while having significantly better consistency (which can also be visible in videos provided in the supplementary material).
> > >
> > > **4D**: It is also worth noticing that we are one of the first methods supporting 4D style transfer and in visual comparisons (Figures 8 and 9) CLIPGaussian outperforms other methods. We would like to emphasize that, to our knowledge, we are the only method that has demonstrated the transfer style on objects (D-NeRF dataset).
> > >
> > > **Multimodality and universality**: We also want to highlight that our main contribution is enabling image- and text-guided stylization across 2D, video, 3D, and 4D data. CLIPGaussian is the first and only method that allows for multimodal and universal style transfer.
> > >
> > > **(2)**
> > >
> > > You raise a valid and important concern.
> > >
> > > In our opinion it is reasonable that we are observing only small improvements both visually and in the metrics with respect to changes in $\lambda_d$. If the local patches are well optimized, there is no significant global improvement.
> > > - During training, n patches are randomly selected. Since the patches do not have empty intersections, they optimize themselves relative to each other in a common area
> > > - Directional CLIP Loss (global) is therefore not crucial in a visual context. But we could observe that it improves consistency, which is reflected in improved metrics. As you can see from Table 12, changes in $\lambda_d$ had a small, but positive impact on the metrics.  $\lambda_d=10$ compared to $\lambda_d=0$ has CLIP-S +1.35, CLIP-SIM +2.39, CLIP-F +0.33, CLIP-CONS +0.09.
> > >
> > > We run a grid of experiments on parameters $\lambda_d \times \lambda_p =$ {0, 5, 10} $\times$ {0, 90, 180\}. In the case of Table 12 we set $\lambda_p=90$ which is a default setting of CLIPGaussian, `patch_size= 128`,  `num_patch = 64` and `feature_lr = 0.01`, which follow default settings.
> > >
> > > Thank you for pointing out the absence of this discussion. We will include a more detailed description of this topic in the manuscript, as we agree that it's important to address it thoroughly.
> > >
> > > **(3)**
> > >
> > > Thank you for your follow-up comment and for highlighting the ambiguity in the introduction.
> > > You're correct that the way we initially presented the tasks: texturing, local editing, insertion, and style transfer, may have suggested that our method explicitly addresses all four. We apologize for not making this distinction more transparent in the previous revision.
> > >
> > > To clarify:
> > >
> > > - **Our contribution is limited to style transfer**, which is the only task we experimentally validate in the paper.
> > > - The other tasks (texturing, local editing, and insertion) were mentioned to provide a broader context, not as a demonstration of our method's capabilities.
> > >
> > > To avoid confusion, we will rephrase this paragraph. We appreciate your feedback, which helped us clarify the scope and presentation of our work.
> > >
> > > We would be glad to provide further clarification if any questions remain.

---

> > > > ### Comment · Reviewer_xZpT · 2025-08-01
> > > >
> > > > Thanks for the prompt reply.
> > > >
> > > > I have no further questions about the quantitative results. It's interesting to see that the model is somewhat sensitive to the \$lambda_p\$. The authors may add more analysis regarding this sensitivity as well.
> > > >
> > > > I would also highly recommend the authors to analyze more about the \$\lambda_d\$, to demonstrate (1) whether it's useful or not, and (2) whether the network needs the related modules or not.
> > > >
> > > > Besides rephrasing the introduction to avoid confusion, I would like to suggest including more important analyses/experiments from the supplementary to the main paper.
> > > >
> > > > In conclusion, I would like to maintain my positive rating

---

> > > > > ### Author Response · Authors · 2025-08-01
> > > > >
> > > > > Thank you for the constructive feedback. We will include the relevant experiments and clarifications, and revise the introduction to address the raised concerns.

---

### Official Review · Reviewer_8XY8 · 2025-07-02

**Clarity:** 3
**Significance:** 3
**Originality:** 4
**Rating:** 4
**Confidence:** 4

**Summary:**

The paper introduces a universal stylization module, CLIPGaussian, capable of performing style transfer across multiple modalities, including 2D images, video, 3D objects, and 4D dynamic scenes. The authors instantiate their approach using Gaussian Splatting (GS) as the underlying 3D representation. For content preservation, they employ a perceptual loss computed using the conv4_2 and conv5_2 layers of VGG-19. The style supervision is two-fold: (i) a Directional CLIP loss, which captures global style alignment by enforcing directional similarity between CLIP embeddings of the content and style prompts, and (ii) a Patch CLIP loss, which encourages local style fidelity by aggregating directional CLIP losses over spatial patches of the stylized output. This modular formulation enables consistent and flexible stylization across diverse input domains.

**Questions:**

Can you provide stronger cross-modality comparisons—especially in 2D, video, and 4D domains?
 The claim of universal stylization across 2D, video, 3D, and 4D is central to the paper. However, the empirical support is heavily skewed toward 3D, with limited or missing comparisons in other domains. For 2D, please consider comparisons with AdaAttN, StyTr², Style Injection in Diffusion. For 4D, results for image-based style transfer seem missing. For video, methods like ViSt3D, ReReVST, MCCNet could serve as strong baselines.

 ➤ Clarifying this and extending comparisons could significantly improve the perceived quality and significance of the work.


Can you report standard temporal consistency metrics (e.g., optical flow-based) for video stylization?
 Relying solely on CLIP directional similarity may overlook frame-to-frame consistency issues. Reporting temporal smoothness metrics, averaged over 15–20 video examples (as is standard in video stylization literature), would help substantiate the method's applicability in real-world video scenarios.

 ➤ A strong quantitative temporal evaluation would likely lead to a higher score in “Quality.”


Could you improve the user study design and provide clearer analysis of results?
 The current user study includes only the 3D modality and fewer than 15 examples, which is insufficient given the paper’s multi-modal scope. Also, rating-based user scoring is hard to interpret compared to preference voting, which is standard in prior work. A clearer textual interpretation of plots (e.g., Fig. 14–17) is also missing.

 ➤ An improved and more interpretable user study across all modalities could raise both “Quality” and “Clarity.”


Do you plan to address the hallucinated artifacts that conflict with content preservation claims?
 In some stylized results (e.g., Fig. 10 and 11), there appear to be hallucinated face-like structures that contradict the claim of content preservation. Is this a limitation of CLIP guidance or an artifact of the Gaussian Splatting representation? Could it be mitigated with an additional loss or mask?

 ➤ Clarifying this would strengthen the reliability of your approach and could improve the score.


Can you correct the figure presentation inconsistencies and table caption errors?
 Figures showing stylized results often omit or downscale the content image, making visual assessment difficult. Also, the captions of Table 9 and 10 are reversed. These issues hurt clarity and reader experience.

 ➤ Addressing these would improve the “Clarity” rating of the paper.


Score Revision Criteria
If the authors provide thorough comparisons across all modalities, improve quantitative evaluation (especially for video), and clarify or resolve content preservation concerns, the “Quality” and “Significance” scores could be raised.

Improvements in user study design and presentation clarity may lead to a higher “Clarity” score as well.

**Ethical Concerns:**

["NO or VERY MINOR ethics concerns only"]

**Final Justification:**

Based on the newly reported **quantitative results** and the authors’ **commitment to address visual and qualitative aspects in the final submission**, I find the rebuttal satisfactory and convincing. I have accordingly **improved my score** to reflect this update.

**Limitations:**

yes

**Paper Formatting Concerns:**

In multiple figures (e.g., Fig. 1, 5, 6, 10, 11), the content image is either missing or shown at a significantly smaller scale than the stylized output, making comparison difficult. For clarity and fairness, all input and output visuals should be consistently scaled and properly aligned.

The captions for Table 9 and Table 10 are reversed, which may confuse readers.

Some figure references and captions lack sufficient detail, especially in the supplementary material (e.g., Fig. 14–17), where plots are shown without textual interpretation.

**Quality:**

3

**Strengths And Weaknesses:**

### **Strengths**

* The paper proposes an interesting and non-trivial idea of **universal stylization** using **Gaussian Splatting** that supports multiple modalities, including **2D images, video, 3D objects, and 4D dynamic scenes**. This unified formulation is both timely and impactful.

* The experimental setup demonstrates that the proposed method is **easy to train and reproducible**, with results obtainable on **reasonable hardware**. This enhances the practical usability of the approach.

* The **supplementary results in the 3D domain** are qualitatively strong, especially under both **text- and image-based style prompts**, suggesting the method’s effectiveness in this modality.

### **Weaknesses**

#### **Major Weaknesses**

1. **Lack of consistent and appropriately scaled content visualization in figures**: In most figures (e.g., Fig. 1, 5, 6, 10, 11), the content image is shown at a reduced scale compared to the stylized output, and in others (Fig. 3, 7, 8, 9), it is completely missing. A direct visual comparison is difficult, undermining the reader’s ability to evaluate content preservation and style transfer quality.

2. **Contradiction between content preservation claims and visual evidence**: At line 327, the authors claim that CLIPGaussian preserves the original content. However, in Fig. 10 (bottom row) and Fig. 11 (top row), the stylized outputs contain **hallucinated human-face-like structures**, which clearly contradict the content fidelity claim.

3. **Insufficient modality-wise comparisons**:

   * **2D**: The paper compares with only two methods on very limited examples. A comprehensive comparison with **SOTA methods** such as *AdaAttN*, *StyTr²*, and *Style Injection in Diffusion* is missing.
   * **3D**: No comparisons are provided against relevant works such as *SGSST* (which is directly related), *StylizedNeRF*, *Style-NeRF2NeRF*, or *MM-NeRF*.
   * **Video**: The evaluation is limited to **short, low-motion clips** (e.g., from DAVIS), with no comparison to methods such as *AdaAttN*, *StyleMaster*, *ViSt3D*, *MCCNet*, or *ReReVST*.
   * **4D**: The supplementary includes **only text-guided** results; **image-guided stylization results are missing**, and comparisons with *StyleDyRF*, *Instruct 4D-to-4D*, etc., are not discussed.
     This undermines the core claim of **universal applicability** and requires a much more thorough evaluation across all supported modalities.

4. **Limited and insufficiently designed user study**: The user study is based on **fewer than 15–20 examples** and is restricted to the **3D object domain**, ignoring other modalities. Additionally, the methodology (rating-based scoring) deviates from **standard forced-choice voting** in literature, making the results harder to interpret. Moreover, no **qualitative discussion** is provided for the results in Fig. 14–17.

5. **Inadequate evaluation of temporal consistency in videos**: The paper relies on **CLIP directional similarity**, which suffers from key limitations:

   * **Semantic granularity loss** (loss of fine details),
   * **Motion insensitivity**, and
   * Inability to capture **localized artifacts**.

   To ensure temporal coherence, authors should report **optical flow-based metrics** averaged over **15–20 examples**, as is common in the video stylization literature.

#### **Minor Weaknesses**

6. **Caption mismatch in tables**: The captions of **Table 9** and **Table 10** are swapped, which could lead to reader confusion.

---

> ### Author Rebuttal · Authors · 2025-07-30
>
> Thank you for your very insightful review.
>
> **Q4/W2 [hallucinated human-face-like structures]:**
>
> Thank you for this very interesting question. The appearance of human-face-like structures in the output is an expected outcome, resulting from the combination of the chosen `patch_size` and the specific style prompt.
>
> For example:
> - in Fig. 10 (bottom row), the scream-face-like patterns are a natural result of the characteristic visual elements in the refereed image "The Scream by Edvard Munch".
> - in Fig. 11 (top row) the prompt “Beksinski painting” caused the creation of human-like structures.  This is consistent with Beksinski’s style, known for dark landscapes and ghostly, human-like characters.
>
> In both cases, these structures are not hallucinations but rather stylizations aligned with the chosen references. The`patch_size` parameter controls the local stylization. For example for  `patch_size=128`, each patch of size 128x128 is stylized. For this reason, detailed repetition can be controlled through the patch_size param. This effect is illustrated in Fig. 23 (Lego scene). Increasing the`patch_size` results in more star-like structures caused by the influence of the prompt “Starry Night by Vincent van Gogh.”
>
> We will add a clarifying note to the experimental description
>
> **Q1/W3:**
>
> - **2D:** The paper compares **only two methods**...
>
> We compare the performance of our model on 2D images with AdaIN and **StyTr²** in Fig. 10 and 31, with InstructPix2Pix and ChatGPT (DALL·E) in Fig. 11, and with CLIPStyler and FastCLIPStyler in Fig. 30. In total, **we compare our model against six other methods, not just two**. Moreover, in the manuscript, we explicitly acknowledge that, given the highly developed state of 2D image stylization methods, our results in 2D may not match the visual quality achieved by large or diffusion-based models (lines 358–359). AdaAttN (2021) is an older method and can no longer be considered state of the art. While we do not directly compare to "Style Injection in Diffusion", we do evaluate our model against ChatGPT (DALL·E), which is also based on diffusion.
>
> - **3D:** No comparisons are provided against relevant works such as **SGSST** (which is directly related), StylizedNeRF, Style-NeRF2NeRF, or MM-NeRF.
>
> Our goal was to compare style transfer on fully trained 3D objects. NeRF-based models have significantly higher computational requirements and cannot be used in the same setting as Gaussian splats. For these reasons, we decided to compare CLIPGaussian only with other GS-based techniques. We emphasize that the base models were identical across all evaluated methods (with the exception of Instruct-GS2GS, which was incompatible). However, in response to the reviewer's suggestion, we additionally compare our model against two recent baselines: **SGSST (CVPR 2025)** and **ABC-GS (ICME 2025)**. Moreover, we have included results obtained using CLIPGaussian with an alternative set of hyperparameters (lambda_p​=35, feature_lr= 0.002), which produces lighter stylization. We call this variant **CLIPGaussian-Lite**.
> The updated results are shown in the table below.
>
> **TABLE 8XY8.A**
>
> |Model|CLIP-S|CLIP-SIM|CLIP-F|CLIP-CONS|Memory Size|
> |-----|:----|:----|:----|:----|:----|
> |||Text-conditioned||||
> |I-GS2GS|16.80|12.03|99.19|**13.54**|-36%|
> |DGE|17.59|12.27|**99.31**|12.46|-5%|
> |**CLIPGaussian-Lite**|23.14|22.30|99.17|8.51|+0%|
> |**CLIPGaussian**|**26.86**|**26.31**|98.80|2.34|+0%|
> |||Image-conditioned||||
> |StyleGaussian|63.69|13.07|98.87|1.36|+0%|
> |SGSST|66.57|16.24|97.54|0.91|+0%|
> |ABC-GS|68.68|16.29|99.10|2.11|+0%|
> |G-Style|**76.94**|**24.94**|98.94|1.31|+126%|
> |**CLIPGaussian-Lite**|69.27|17.02|**99.16**|**5.26**|+0%|
> |**CLIPGaussian**|72.65|20.72|98.78|1.77|+0%|
>
> CLIPGaussian, in both the base and light variants, outperforms SGSST and ABC-GS in stylization quality metrics (CLIP-S and CLIP-SIM). Notably, CLIPGaussian-Light achieves the highest scores across all consistency metrics. It is important to highlight that G-Style significantly increases the model size; typically by more than a factor of two, whereas CLIPGaussian offers competitive results without altering the underlying model size.
>
> - **Video:** ...no comparison to methods such as AdaAttN, StyleMaster, ViSt3D, MCCNet, or ReReVST.
>
> Since MCCNet (2020), ReReVST (2020), and AdaAttN (2021) are relatively older methods, we chose to compare our model with more recent baselines. We acknowledge that StyleMaster would be a valuable baseline; however, as its code was unavailable at the time of the submission, it was not possible to include it in our comparison. Nevertheless, we will add them to the related work section. In response to the reviewer's suggestion, we additionally compare our model with ReReVST and ViSt3D using the same dataset and experimental setting as described in the manuscript. The updated results are shown in the table below.
>
> **TABLE 8XY8.B**
>
> |Model|CLIP-S|CLIP-SIM|CLIP-F|CLIP-CONS|
> |-----|:----|:----|:----|:----|
> |||Text-conditioned|||
> |Rerender|19.40|9.83|98.23|-0.03|
> |Text2Video|26.05|**24.99**|93.36|0.03|
> |**CLIPGaussian**|**26.25**|24.53|**99.00**|**1.92**|
> |||Image-conditioned|||
> |CCPL|18.89|8.20|97.92|-0.02|
> |UniST|15.93|3.85|99.36|5.16|
> |ViSt3D|55.92|2.75|99.18|3.28|
> |ReReVST|61.50|2.12|98.85|2.51|
> |**CLIPGaussian-Light**|63.60|8.06|**99.50**|**11.99**|
> |**CLIPGaussian**|**74.31**|**17.60**|99.18|1.27|
>
> - **4D:**...image-guided stylization results are missing, and comparisons with StyleDyRF, Instruct 4D-to-4D, etc., are not discussed
>
> Instruct 4D-to-4D has been discussed in main manuscript lines 114-121 in section Related Works and line 315-318 in section Experiments. Instruct 4D-to-4D focuses mainly on the color palette, while our approach additionally modifies the geometry of the scene. Consequently, the results obtained by CLIPGaussian contain more details, which in our opinion better resembles the referenced style. We will add this discussion. The supplementary also includes image-guided results for 4D, see Fig. 25. We will add more examples to the main manuscript.
>
> We thank the reviewer for pointing out the StyleDyRF method, which is a relevant NeRF based zero shot style transfer approach that stylizes dynamic scenes represented by Dynamic NeRF. In contrast to our method, it supports only image based style conditioning. In our work we compared against more recent img-conditioned zero shot approaches such as 4DStyleGaussian, which similarly to our method works on a GS scene representation. We will add the discussion to the related work section.
>
> **Q2/W5 [optical flow-based metrics for video]:**
> We conducted an additional evaluation of CLIPGaussian using short-range and long-range consistency metrics, following the implementation and settings from StyleRF. Specifically, we warp one view to another using optical flow and softmax splatting, and then compute the masked RMSE and LPIPS scores to assess stylization consistency. For short-range consistency, we average the scores over pairs of the i-th and (i+1)th frames; for long-range consistency, we use pairs of the i-th and (i+7)th frames. We evaluate Video-CLIPGaussian and the baselines on the same dataset used in the paper. The results are presented in the table below.
>
> **TABLE 8XY8.C**
>
> |Model|Short-range consistency||Long-range consistency||
> |-----|:----|:----|:----|:----|
> ||LPIPS|RMSE|LPIPS|RMSE|
> |OriginalVideos|0.042|0.034|0.070|0.055|
> |||Text-conditioned|||
> |Rerender|0.062|0.040|0.132|0.077|
> |Text2Video|0.261|0.183|0.235|0.166|
> |**CLIPGaussian-Light**|**0.049**|**0.036**|**0.085**|**0.061**|
> |**CLIPGaussian**|0.084|0.057|0.152|0.095|
> |||Image-conditioned|||
> |CCPL|0.102|0.065|0.132|0.093|
> |UniST|**0.062**|0.047|0.088|0.066|
> |ViSt3D|0.081|0.043|0.121|0.063|
> |ReReVST|0.072|0.047|0.100|0.068|
> |**CLIPGaussian-Lite**|0.049|**0.035**|**0.083**|**0.061**|
> |**CLIPGaussian**|0.086|0.057|0.157|0.098|
>
> The interpretation of these metrics is not trivial. Consistency metrics are maximized when the output is identical to the original object, which suggests poor stylization. Notably, only Text2Video deviates significantly from the other methods in terms of consistency. It is also worth noting that while Rerender achieves low metrics values, the stylization and content of the videos decreases in time **(see suplementary_material/Video/comparisons/comparison_white_wool.mp4)**. CLIPGaussian offers superior stylization quality compared to the baselines, and we believe it achieves high-quality style transfer without compromising the consistency of the object or video.
> Additionally in the response to the reviewer **XWe5**, we evaluate 3D spatial consistency **(Table XWe5.A)** as well as 4D spatial and temporal consistencies using the same metrics **(Tables XWe5.B-D)**.
>
> **Q3/W4 [user study]:**
>
> The user study is described in detail in App. B.2.2. It was conducted carefully using the CLICKworker platform, which enabled us to recruit a balanced group of participants (lines 641-644). The structure of the survey closely follows that used in the G-Style paper[21], including the preference ranking question described in line 660. The questions were constructed identically to the reference work. We additionally proposed question 4 (line 674) and additional question (line 680). This alignment allowed us to make meaningful comparisons while building on validated recent prior work.
> We would like to emphasize that the user study is a supporting evaluation, not the core of our contribution. The primary contribution of our work lies in presenting a general and flexible stylization framework, which can be applied to any Gaussian-based model and supports both text- and img-driven conditioning. To our knowledge, no other existing method offers this level of adaptability
>
> **Q5, Q6/W1, MW1[Paper formatting]:**
> Thank you for highlighting the editorial corrections and typos we will address them carefully.
>
> We would be happy to address any further questions and provide clarifications.

---

> ### Comment · Reviewer_8XY8 · 2025-08-02
>
> I thank the authors for the detailed and constructive rebuttal.
>
> **Content Fidelity and Stylization Hallucinations (Q4/W2):**
>
> The clarification on the appearance of face-like patterns (e.g., due to patch size and the semantic content of the style prompt) was helpful. I now agree these are not hallucinations in the negative sense but a result of stylization driven by strong structural priors in the reference styles. The authors' suggestion to add explanatory notes in the manuscript would improve clarity.
>
> ---
>
> **Cross-Modality Evaluation (Q1/W3) and Temporal Consistency (Q2/W5):**
> While the rebuttal adds comparisons across 2D, 3D, video, and 4D modalities, I remain concerned about the **evaluation in video stylization**, especially for real-world, high-motion scenarios.
>
> Specifically:
> * Most video stylization experiments are conducted on low-motion datasets (e.g., DAVIS) or synthetic datasets like D-NeRF, which feature **clean geometry, low texture variance, and minimal deformation**.
>
> * These settings do not reflect real-world stylization challenges such as background flicker, motion blur, and viewpoint inconsistencies.
>
> Regarding temporal consistency evaluation, I also note that the entire video frame—**including both foreground and background—should be considered** during optical-flow-based evaluation. Restricting evaluation to foreground regions only (as implied by masked metrics in 3D/4D settings) underestimates flickering and inconsistency, which commonly arise in real-world backgrounds.
>
> The **temporal consistency metrics** (short- and long-range LPIPS and RMSE) reported by the authors are informative **but insufficient to assess long-term perceptual quality** in high-motion scenarios:
>
> * LPIPS, while good for short-term perceptual similarity, cannot capture motion-induced artifacts across large frame gaps.
>
> * RMSE is sensitive to natural motion and may penalize valid stylization changes, thus leading to misleading conclusions.
>
> I appreciate the inclusion of baselines such as ViSt3D and ReReVST. However, **methods like AdaAttN**, although older, show evaluation in motion of the video contexts and could offer a valuable comparison reference for robustness assessment.
> Moreover, while CLIPGaussian-Lite shows improved consistency scores, its stylization quality remains unverified as no visual results are provided in the paper or supplementary. Given the stylization-consistency trade-off, this is a non-trivial omission.
>
> ---
>
> **Q3 / W4: User Study**
>
> While the authors clarify their user study setup and note the influence from G-Style, it is still limited in scope (3D domain only) and uses a less interpretable rating-based approach. The lack of **forced-choice comparisons** and the absence of study results across other modalities reduces the significance of this evaluation.
>
> ---
>
> **Overall**
>
> The paper introduces a promising universal stylization framework with clear strengths in modularity and multi-modality support. However, I believe that the *video modality*, which is central to the paper’s "universal" claim, is still *under-evaluated*, particularly in challenging, real-world, high-motion scenarios.
>
> I would be willing to raise my score if the following are clearly addressed in the rebuttal and supplementary material:
>
> * Reporting **optical flow-based metrics** for temporal consistency and direct comparison with baseline video stylization methods (e.g., AdaAttN, ViSt3D, ReReVST).
>
> * Inclusion of visual results and consistency metrics on **real-world, high-motion videos**, using datasets beyond DAVIS and D-NeRF. If authors commit to doing it in a camera-ready revision.
>
> * Presentation of **CLIPGaussian-Lite visual outputs**. If authors commit to doing it in a camera-ready revision.
>
> Unless the above is satisfied, I will maintain my current rating.
>
> ---

---

> > ### Author Response · Authors · 2025-08-03
> >
> > We thank the reviewer for valuable observations and we are pleased that the potential of our work, particularly its “universal stylization framework”, has been acknowledged.
> >
> > **1)**
> >
> > We also thank the reviewer for the additional clarification regarding temporal consistency. In addition to previously used metrics, we employ the Farneback optical Flow based metric, following the setup proposed in the ViSt3D paper (NeurIPS 2023). Accordingly, in the case of video:
> >
> > FoF \= Farneback optical Flow
> > $m_{\text{FoF}, k}= \frac{1}{N}\sum_{i=1}^N |FoF(GT_i, GT_{i+k}) - FoF(Style_i, Style_{i+k})|$
> >
> > We also took AdaAttN into account for quantitative comparison. For video results on the DAVIS dataset, the metrics are as follows:
> >
> > **TABLE 8XY8.B2 Updated results for Video stylization**
> >
> > |Model|CLIP-S|CLIP-SIM|CLIP-F|CLIP-CONS|
> > |-----|:----|:----|:----|:----|
> > |||Text-conditioned|||
> > |Rerender|19.40|9.83|*98.23*|\-0.03|
> > |Text2Video|26.05|**24.99**|93.36|*0.03*|
> > |**CLIPGaussian**|**26.25**|*24.53*|**99.00**|**1.92**|
> > |||Image-conditioned|||
> > |CCPL|18.89|*8.20*|97.92|\-0.02|
> > |UniST|15.93|3.85|*99.36*|*5.16*|
> > |ViSt3D|55.92|2.75|99.18|3.28|
> > |AdaAttN|57.71|\-1.04|97.56|1.08|
> > |ReReVST|61.50|2.12|98.85|2.51|
> > |**CLIPGaussian-Light**|*63.60*|8.06|**99.50**|**11.99**|
> > |**CLIPGaussian**|**74.31**|**17.60**|99.18|1.27|
> >
> > **TABLE 8XY8.C2 Updated results consistency metrics for Video**
> >
> > ||$m\_{\\text{FoF,k}}$|||||
> > |-----|:----|:----|:----|:----|:----|
> > |Model/k|1|2|4|8|16|
> > |||Text-conditioned||||
> > |Rerender|*1.08*|*2.07*|*4.40*|*7.95*|*11.34*|
> > |Text2Video|5.06|5.91|7.68|9.93|11.34|
> > |**CLIPGaussian**|**0.41**|**0.79**|**2.02**|**4.00**|**6.58**|
> > |||Image-conditioned||||
> > |CCPL|1.40|2.36|4.62|6.86|8.45|
> > |UniST|0.43|*0.73*|*1.75*|*3.21*|*4.98*|
> > |ViSt3D|0.75|1.19|2.30|3.59|4.54|
> > |AdaAttN|0.69|1.09|2.57|3.86|5.26|
> > |ReReVST|0.46|0.84|2.30|3.43|4.94|
> > |**CLIPGaussian-Light**|**0.20**|**0.35**|**0.83**|**1.46**|**2.32**|
> > |**CLIPGaussian**|*0.41*|0.79|2.12|4.06|7.00|
> >
> > The results suggest that CLIPGaussianLight consistently outperforms the baseline methods in text- as well as image-conditioned stylization in terms of FoF-based metric. Moreover, the base version of CLIPGaussian also achieves highly competitive results, which fall short of only UniST. This relative performance against UniST mirrors the results achieved on CLIP-based consistency metrics.
> >
> > **2)**
> >
> > We would like to emphasize that we agree that evaluation on a real-world dataset is very important. However, we would like to highlight that accurate reconstruction remains a challenging task, even with current SOTA approaches.
> >
> > Our model works as a plugin to the selected base model. Consequently, our performance is closely related to the reconstruction quality of the underlying base model. It is worth noting that as a result our model benefits from the advancements of Gaussian Splatting-based reconstruction models.
> >
> > Nevertheless, we believe that this is an important task and we promise to include both visual and numerical comparisons on real-world datasets.
> >
> > For Video: In the camera-ready version, visual and numerical results will also be shown on the open Ultra Video Group (UVG) dataset, which is also used to evaluate video-Gassian-Splatting based models. The dataset is composed of 16 versatile high-resolution test video sequences captured at 50/120 fps.
> >
> > For 4D, we present an example of styling on a challenging real-world PanopticSports dataset (Figure 29). Due to the limited rebuttal time, below we present a numerical comparison for four prompts for the `boxes` dataset; for the camera ready version, we are preparing a comparison for other datasets. In the case of 4D, the metric $m_{\text{FoF}, k=1}$ is additionally averaged across all cameras.
> >
> > **TABLE 8XY8.D Consistency metrics for 4D**
> >
> > |  | boxes |  |  |  |  |
> > | :---- | :---- | :---- | :---- | :---- | :---- |
> > |  |  | Short-range Consistency |  | Long-range Consistency |  |
> > |  |  $m\_{\\text{FoF, k=1}}$ | LPIPS | RMSE | LPIPS | RMSE |
> > | D-MiSo | 0.062 | 0.011          | 0.010   | 0.018  | 0.018 |
> > | Fire | 0.069 | 0.011        | 0.014 | 0.017  | 0.022 |
> > | Starry Night by Vincent van Gogh | 0.079 | 0.019           | 0.019 | 0.028 | 0.027 |
> > | the great wave off kanagawa by katsushika hokusai | 0.074 | 0.020           | 0.021 | 0.028 | 0.030 |
> > | blooming flowers, lush green leaves, intertwining branches, serene garden, floral patterns, leafy textures | 0.065 | 0.009           | 0.012  | 0.014  | 0.019 |
> >
> > **3)**
> >
> > Due to new rules, we are not allowed to include visual comparisons in the rebuttal. However, we commit to add them to the camera-ready version.
> >
> > ****
> >
> > As we consider these discussions valuable, we will supplement the manuscript with additional experiments and analysis of visual results, focusing primarily on the effects of patch size and the semantic content of the style prompt.
> >
> > We are happy to provide further clarification should any questions remain.

---

> ### Comment · Reviewer_8XY8 · 2025-08-04
>
> Thank you to the authors for their detailed follow-up and appreciate the additional clarifications provided.
>
> **Q1: Temporal Consistency and Optical Flow Evaluation**
>
>  I thank the authors for incorporating **optical flow-based metrics**, following the ViSt3D setup, and for reporting performance against baselines such as **AdaAttN**, **ViSt3D**, and **ReReVST**. Based on the numbers in Tables **8XY8.B2** and **8XY8.C2**, the temporal stability of **CLIPGaussian-Lite** is now much clearer and quantitatively substantiated. The results on the DAVIS dataset show that CLIPGaussian-Lite consistently achieves favorable frame-to-frame consistency and competitive performance across both short and long temporal windows.
>
> I am assuming that the authors have used **stylized video results**, and there might be a typo in the mathematical description of the **FoF** metric, where "Style" should likely be corrected to "Stylized" in the optical flow equation.
>
> **Q2: Evaluation on Real-World, High-Motion Videos**
>
>  I also appreciate the authors’ acknowledgement of the importance of testing on real-world, high-motion sequences and their commitment to reporting results on the **Ultra Video Group (UVG)** dataset in the camera-ready version. This dataset, with its high frame rate and diverse content, is a strong candidate for validating video stylization models under realistic conditions.
>
> Additionally, the use of **PanopticSports** for 4D stylization evaluation is a valuable step forward. The inclusion of consistency metrics for different prompts (Table 8XY8.D) improves the empirical grounding for 4D results.
>
> **Remaining Concerns**
>
>  While the authors have made credible commitments for extended evaluations and visual results in the camera-ready version, the current rebuttal still lacks:
> * Rich **qualitative visual analysis** due to NeurIPS 2025 media-sharing constraints.
> *  A sufficiently broad and standardized **user study** using forced-choice protocols.
>
> These aspects limit the interpretability of stylization quality and user preference, especially across modalities.
>
> **Overall Evaluation**
>
>  Based on the newly reported **quantitative results**, and the authors’ **commitment to address visual and qualitative aspects in the final submission**, I find the rebuttal **satisfactory and convincing**. I have accordingly **improved my score** to reflect this update.

---

> > ### Author Response · Authors · 2025-08-04
> >
> > Thank you for dedicating your time to reviewing our manuscript. We believe that the revisions have given the work greater context, enhancing its value for the reader. We sincerely appreciate the decision to increase the score.

---

### Official Review · Reviewer_XWe5 · 2025-07-02

**Clarity:** 3
**Significance:** 3
**Originality:** 3
**Rating:** 5
**Confidence:** 4

**Summary:**

This paper aims to handle the style transfer problem of Gaussian Splatting (GS) based representations. This paper proposes CLIPGaussian, the first unified style transfer framework that supports text-/image-based style guidance and multiple modalities, including: 2D images, videos, 3D-based GS and 4D-based GS. The experimental results demonstrate superior style fidelity and consistency across all tasks (image/video/3D/4D style transfer).

**Questions:**

1. **[Metric of 3D Consistency].** In line 270-274, the authors briefly introduce the utilized metrics: CLIP-SIM, CLIP-S, CLIP-CONS, and CLIP-F. CLIP-CONS is used to evaluate the temporal consistency. From Appendix B.1, I find that the cross-frame consistency is evaluated via the CLIP embeddings. However, it is not convincing enough because the consistency of local regions or pixels should also be considered. Could the authors provide short-range / long-range consistency following [1,2]?
2. **[Quantitative Evaluation of Consistency in 4D Style Transfer].** As discussed in Weakness #2, the quantitative evaluation of temporal consistency in 4D style transfer is important for proving whether CLIPGaussian can effectively preserve the temporal consistency.
3. **[Spatial-Temporal Consistency].** For 4D style transfer, both the spatial and temporal consistency should be considered. In practical, the temporal consistency might be a troublesome issue. Could CLIPGaussian effectively handle this issue? From the methodology part, I can only find the modified multi-view rendering loss which can ensure the spatial consistency, but the temporal consistency among different time in local regions lack explicit regularization.

[1] StyleRF: Zero-shot 3D Style Transfer of Neural Radiance Fields.

[2] StyleDyRF: Zero-shot 4D Style Transfer for Dynamic Neural Radiance Fields.

**Ethical Concerns:**

["NO or VERY MINOR ethics concerns only"]

**Final Justification:**

The authors have addressed my previous concerns about the consistency issues. I have no further questions and keep my positive rating for this work.

**Limitations:**

yes

**Quality:**

3

**Strengths And Weaknesses:**

**Strengths:**
1. CLIPGaussian can be simply extended to style transfer tasks of different modalities including: images, videos, 3D and 4D scenes.
2. CLIPGaussian can support both text-based style guidance and image-based style guidance.
3. The authors conduct solid experiments on 3D, image, video style transfer, demonstrating the great performance of CLIPGaussian.

**Weakness:**
1. The quantitative metrics of temporal consistency (line 270-274) are based on CLIP embeddings. The evaluation of temporal consistency in local regions/pixels should also be considered.
2. Though the authors provide the quantitative evaluation of consistency in 3D style transfer, but the consistency evaluation in 4D style transfer is missing. Since 4D style transfer is also an important task that CLIPGaussian can solve, it is not convincing enough with only the results of 3D style transfer.

---

> ### Author Rebuttal · Authors · 2025-07-30
>
> We would like to thank the Reviewer for the kind words and interesting indication of gaps, we believe that the newly designed experiments will indeed help to improve the work.
>
> **W1/Q1 \[Metric of 3D Consistency\]**
>
> As suggested, we evaluated the CLIPGaussian using short-range and long-range consistency following the implementation and settings from StyleRF. Specifically we warp one view to the other according to the optical flow using softmax splatting, and then compute the masked RMSE and LPIPS scores to measure the stylization consistency. For short-range we average over pairs of i-th and (i+1)-th frames and for long-range we take i-th and (i+7)-th frames. We evaluate 3D-CLIPGaussian and baselines on the same data as in the paper. Additionally, as asked by the reviewers **8XY8** and **xZpT** we evaluate against two more baselines, SGSST (CVPR 2025\) and ABC-GS (ICME 2025). Moreover, we have included results obtained using CLIPGaussian with an alternative set of hyperparameters (lambda\_p​=35, feature\_lr \= 0.002), which produces lighter stylization. We call this variant “**CLIPGaussian-Lite**”. It is important to note that this model still outperforms other methods in terms of stylization quality (CLIP-S and CLIP-SIM), with the exception of G-Style, which achieves slightly higher scores at the cost of more than doubling the model size (see table Tab. 8XY8.A in response to reviewer 8XY8).
>
> **TABLE XWe5.A: Consistency metrics for 3D**
>
> | Model | Short-range consistency |  | Long-range consistency |  |
> | ----- | :---- | :---- | :---- | :---- |
> |  | LPIPS | RMSE | LPIPS | RMSE |
> | Original Videos | 0.053 | 0.047 | 0.140 | 0.123 |
> |  |  |  |  |  |
> |  |  | Text-conditioned  |  |  |
> | I-GS2GS | **0.048** | **0.043** | 0.148 | 0.131 |
> | DGE | 0.055 | 0.044 | 0.148 | 0.120 |
> | **CLIPGaussian-Lite** | 0.051 | **0.043** | **0.132** | **0.117** |
> | **CLIPGaussian** | 0.057 | 0.051 | 0.148 | 0.125 |
> |  |  |  |  |  |
> |  |  | Image-conditioned |  |  |
> | StyleGaussian | 0.052 | 0.061 | 0.145 | 0.144 |
> | SGSST | **0.040** | 0.050 | **0.139** | 0.142 |
> | ABC-GS | 0.048 | **0.043** | 0.141 | 0.126 |
> | G-Style | 0.054 | 0.049 | 0.145 | 0.136 |
> | **CLIPGaussian-Lite** | 0.055 | 0.045 | **0.139** | **0.120** |
> | **CLIPGaussian** | 0.064 | 0.055 | 0.162 | 0.132 |
>
> Additionally we also measure short-range and long-range consistency for video style transfer. The table can be found in response to reviewer **8XY8 (TABLE 8XY8.C)**.
>
> **W2/Q2 \[Quantitative Evaluation of Consistency in 4D Style Transfer\] / Q3 \[Spatial-Temporal Consistency\]:**
>
> CLIPGaussian is built on top of specialized architectures tailored to each modality. For 4D data, our framework is implemented on top of D-MiSo. As input, we use a fully trained D-MiSo model, which already addresses temporal consistency through a dedicated deformation network that models changes in the Gaussians throughout time. During stylization, we keep the weights of this network fixed to ensure that the stylization process does not directly affect temporal consistency. We assume that D-MiSo (or any other 4D model adapted for use with CLIPGaussian) handles this aspect. Nevertheless, in response to the reviewer's suggestion, we evaluate two types of consistency in 4D data:
>
> 1\. The first is consistency across different views at a fixed moment in time. This is similar to the 3D setup, as described in Equations 11 and 12\. To evaluate this GT is required for each view at a given time
> 2\. The second is temporal consistency, where we fix the camera and analyze how the object changes over time.
>
> In the supplementary material, we included visual examples that reflect both camera changes and temporal change for example:
> supplementary\_material/4D/examples/jumpingjacks\_cubism.mp4
>
> Unfortunately, the D-NeRF dataset does not provide GT across time for each camera. Only one time point is available in the data for each camera. However, we agree with the reviewer that in both cases, considering the metrics is very important. Therefore, we decided to conduct two experiments, supplementing Tab. 13:
>
> 1\. For a set of time steps {0, 0.1, 0.2, ..., 1}, we used the test cameras from NeRF-Synthetic (200 views, as in the jumpingjacks\_cubism.mp4) to generate reference images using the base D-MiSo model. Then, we generated the corresponding stylized images and computed the mean CLIP-F and CLIP-CONS metrics for time steps
>
> **TABLE XWe5.B:**
>
> | dataset | hook | jumpingjacks | trex | bouncingballs | hellwarrior | mutant | standup |
> | :---- | :---- | :---- | :---- | :---- | :---- | :---- | :---- |
> |  |  |  |  |  |  |  |
> |  |  |  | CLIP-F |  |  |  |  |
> | batch=1 | 97.56 | 98.41 | 97.36 | 98.73 | 98.41 | 98.01 | 98.03 |
> | 2 | 97.37 | 98.15 | 97.34 | 98.66 | 98.23 | 97.95 | 97.87 |
> | 4 | 97.66 | 98.35 | 97.68 | 98.33 | 98.26 | 97.65 | 98.19 |
> |  |  |  |  |  |  |  |
> |  |  |  | CLIP-CONS  |  |  |  |  |
> | 1 | 2.1 | 2.31 | 2.25 | 0.66 | 5.64 | 1.78 | 2.60 |
> | 2 | 1.7 | 2.80 | 3.24 | 1.03 | 5.72 | 1.92 | 3.02 |
> | 4 | 2.7 | 2.97 | 2.97 | 0.53 | 6.29 | 2.43 | 3.68 |
>
> From this experiment, we can see that it is difficult to indisputably determine which styling is best based on the selected metrics. We emphasize this in line 778 that visual comparison (Fig. 28\) reveals substantial qualitative differences in output fidelity. Fig. 8 shows that with a larger batch, the shape of the object is better preserved (this is clearly visible for patch=32; where the mutant's head was more accurately reproduced only when we used batch=4 ). We will add additional metrics to the manuscript and describe our observations clearly.
>
> 2\. For each test camera from the D-NeRF dataset, we generated stylized images at {0, 0.05, 0.1, ..., 1} time points. Next, we calculated the average metrics to evaluate temporal consistency across time for each camera.
>
> **TABLE XWe5.C**
>
> | dataset | hook | jumpingjacks | trex | bouncingballs | hellwarrior | mutant | standup |
> | :---- | :---- | :---- | :---- | :---- | :---- | :---- | :---- |
> |  |  |  |  |  |  |  |  |
> | Short-range consistency: LPIPS |  |  |  |  |  |  |  |
> | 1 | 0.017 | 0.019 | 0.012 | 0.013 | 0.023 | 0.007 | 0.009 |
> | 2 | 0.017 | 0.018 | 0.012 | 0.012 | 0.025 | 0.007 | 0.010 |
> | 4 | 0.017 | 0.017 | 0.012 | 0.013 | 0.026 | 0.007 | 0.009 |
> | D-MiSo | 0.009 | 0.011 | 0.008 | 0.007 | 0.018 | 0.006 | 0.006 |
> |  |  |  |  |  |  |  |  |
> | Short-range consistency: RMSE |  |  |  |  |  |  |  |
> | 1 | 0.046 | 0.061 | 0.046 | 0.021 | 0.055 | 0.028 | 0.031 |
> | 2 | 0.045 | 0.058 | 0.044 | 0.020 | 0.056 | 0.028 | 0.032 |
> | 4 | 0.044 | 0.055 | 0.043 | 0.021 | 0.057 | 0.028 | 0.031 |
> | D-MiSo | 0.023 | 0.036 | 0.029 | 0.018 | 0.043 | 0.017 | 0.018 |
> |  |  |  |  |  |  |  |  |
> | Long-range consistency:  LPIPS |  |  |  |  |  |  |  |
> | 1 | 0.028 | 0.062 | 0.049 | 0.043 | 0.027 | 0.044 | 0.031 |
> | 2 | 0.025 | 0.063 | 0.050 | 0.043 | 0.026 | 0.043 | 0.030 |
> | 4 | 0.026 | 0.063 | 0.048 | 0.040 | 0.027 | 0.039 | 0.029 |
> | D-MiSo | 0.016 | 0.046 | 0.031 | 0.028 | 0.020 | 0.031 | 0.021 |
> |  |  |  |  |  |  |  |  |
> | Long-range consistency: RMSE  |  |  |  |  |  |  |  |
> | 1 | 0.049 | 0.043 | 0.031 | 0.028 | 0.062 | 0.027 | 0.044 |
> | 2 | 0.050 | 0.043 | 0.030 | 0.025 | 0.063 | 0.026 | 0.043 |
> | 4 | 0.048 | 0.040 | 0.029 | 0.026 | 0.063 | 0.027 | 0.039 |
> | D-MiSo | 0.031 | 0.028 | 0.021 | 0.016 | 0.046 | 0.020 | 0.031 |
>
> An interesting observation is that in almost every case, when using selected hyperparameters, we see an increase in metrics compared to the base model. In our case, a lot depends on stage 1; a good idea for future work would be to check the performance of colors represented using RGB instead of spherical harmonics in order to avoid light inconsistencies between views. We will include this observation in the manuscript as well.
>
> **TABLE XWe5.D**
>
> |  | Fire |  |  |  | Lotus flower  |  |  |  | The Great Wave of Kanagawa by Katsushika Hokusai |  |  |  |
> | :---- | :---- | :---- | :---- | :---- | :---- | :---- | :---- | :---- | :---- | :---- | :---- | :---- |
> |  | Short-range Consistency |  | Long-range Consistency |  | Short-range Consistency |  | Long-range Consistency |  | Short-range Consistency |  | Long-range Consistency |  |
> | batch=4; num\_crops=64; | LPIPS | RMSE | LPIPS | RMSE | LPIPS | RMSE | LPIPS | RMSE | LPIPS | RMSE | LPIPS | RMSE |
> | hellwarrior | 0.022 | 0.054 | 0.053 | 0.120 | 0.023 | 0.052 | 0.058 | 0.120 | 0.026 | 0.058 | 0.061 | 0.128 |
> | hook | 0.015 | 0.040 | 0.043 | 0.097 | 0.014 | 0.040 | 0.044 | 0.096 | 0.016 | 0.048 | 0.045 | 0.102 |
> | jumpingjacks | 0.015 | 0.051 | 0.035 | 0.096 | 0.015 | 0.049 | 0.036 | 0.093 | 0.018 | 0.057 | 0.037 | 0.095 |
> | trex | 0.010 | 0.037 | 0.025 | 0.071 | 0.010 | 0.036 | 0.026 | 0.070 | 0.011 | 0.043 | 0.027 | 0.079 |
>
> It is interesting that despite differences in the (well-represented) prompt, the metrics usually behave very similarly for the selected object and the changes are minor. This emphasizes the importance of object reconstruction (stage 1). Nevertheless, it also shows that our model yields consistent results across different prompts.
>
> We sincerely thank the reviewer once again, as the new experiments provide additional insights into our model. We will include both the quantitative and qualitative results in the paper. To further highlight the effect of time and viewpoint changes, we will also include visualizations for additional objects, showing reconstructions and stylized versions at selected moments and camera positions.

---

> > ### Comment · Reviewer_XWe5 · 2025-08-05
> >
> > Thanks for the detailed reply. The authors have addressed my previous concerns about the consistency issues. I have no further questions and keep my positive rating for this work.

---

> > > ### Author Response · Authors · 2025-08-05
> > >
> > > We appreciate your constructive evaluation of our work and we are pleased that your concerns were resolved.

---

### Official Review · Reviewer_ed8o · 2025-07-03

**Clarity:** 3
**Significance:** 3
**Originality:** 3
**Rating:** 5
**Confidence:** 4

**Summary:**

This study presents CLIPGaussian, a plug-in module for style transfer that can be applied to various data types such as 2D images, 3D objects, videos, and 4D dynamical scenes. The core idea is to represent various data types as Gaussian Splatting (GS) representations, and design a unified framework to manipulate this representation to transfer the reference style. The proposed approach incorporates two main steps: (1) authors train the base model which learns how to represent the object or scene using Gaussians (i.e., normal Gaussian Splatting training using existing baselines). Then, CLIPGaussian modifies the properties of the Gaussians so that the final result looks similar to the target style (i.e., a given text or reference image). Authors conduct both qualitative and quantitative evaluations. The evaluation across multiple domains (2D, 3D, 4D scenes) was impressive and comprehensive. Results suggest that this method outperform existing studies in various aspects without increasing initial model size or complexity.

**Questions:**

**1. [Control over geometry]:** Authors can show how much their model is capable of controlling or imitating diverse geometries. Adding diverse examples for geometry control might help to understand model capabilities.

**2. [Failure cases]:** Adding failure cases can help to further understand how this model can be improved and which reference styles can lead to failure. Based on qualitative results, it slightly seemed that the method can better imitate styles given by the text prompts (in comparison with reference images).

**3. [Diversity of reference images or text prompts]:** Text-guided stylization might be subjective and dependent on prompts used for evaluation. What was the diversity of text prompts or reference images used in the evaluation? How much method is sensitive to vague prompts? Is model capable of imitating diverse geometry constraints or is limited to simple control over geometry?

I am on the positive side for this paper, and I am willing to improve my initial score after reading authors comments and replies.

**Ethical Concerns:**

["NO or VERY MINOR ethics concerns only"]

**Final Justification:**

Thanks for the authors effort to provide the rebuttal. The rebuttal tries to address my concerns, providing explanations and examples on geometric control (e.g., “fur” and “cubism” styles), systematic failure case analysis, and prompt diversity; with these additions, I remain positive and support my initial positive acceptance score.

**Limitations:**

It can be improved if authors add more failure cases (e.g., vague prompts, diverse reference styles, sparse reference images, etc.)

**Paper Formatting Concerns:**

I did not notice major formatting concerns.

**Quality:**

3

**Strengths And Weaknesses:**

Thanks for the authors effort to prepare this draft. This paper was among the papers that I really enjoyed reviewing it; considering its clear representation, comprehensive evaluation, detailed supplementary, and innovative ideas behind it.

**- Strengths:**

**1. [Impressive and comprehensive evaluation]:** Authors provide comprehensive qualitative and quantitative evaluations across various modalities (2D, 3D, and 4D scenes). Results suggest that the proposed method outperforms existing ones or it is among top performing ones in various modalities.

**2. [Unified framework across multiple modalities]:** It is interesting how the proposed simple framework can be applied to various data types and outperform existing methods in each domain. It is promising to have a unified framework across various domains.

**3. [Innovative idea]:** I like the basic idea behind the paper. It enables cross-modal generality by focusing on GS representation, converting various domains to a single unified representation, and then using well-known CLIP and VGG losses to transfer the style.

**4. [Clear presentation]:** Paper is well-written. Steps are clearly explained. Method and results are nicely communicated.

**5. [Efficient model]:** The model works as a plug and it does not increase the model size.

-----
 **- Room for improvement:**

**1. [Performance on 2D images]:** 2D image stylization might not as well as other modalities, considering the performance of 2D diffusion models.

**2. [Inherited dependence on base GS model]:** This model performs as a plugin and built upon the base model. If base model fails, style transfer might also fail.

**3. [Adding domain specific loss terms]:** While this method is generalizable across various modalities, the performance in each domain can be improved if loss terms specific to that domain would be added to ensure successful style transfer. For example, for 4D dynamic scenes, it might be beneficial to add loss terms that ensures temporal consistency.

---

> ### Author Rebuttal · Authors · 2025-07-30
>
> We sincerely thank the reviewer for their thoughtful comments and interesting questions. We are especially pleased that the overall pipeline was well understood and that its contributions were clearly recognized. We hope the responses below will address any remaining concerns and help us further improve the quality of the paper.
>
> **W2, W3  \[Inherited dependence on base GS model, Adding domain specific loss terms\]:**
>
> In this work we used background loss as an example of a domain specific loss term. It was only used in case of 3D and 4D objects to ensure that the background won’t be affected by the stylization. In future work, it would be a good idea to provide additional support for the base model, especially in terms of domain-specific consistency. We believe that adding the appropriate loss (for example time related in case of 4D/Video) would contribute to the correctness of the base model and stage 2: the styling itself. We will highlight this as a future work direction.
>
> **Q1 \[Control over geometry\]:**
>
> Our model is based on the Gaussian Splatting representation. We can consider editing the object as in the case of the D-MiSo/VeGaS/MiraGe model (all of these base architectures support object editing), because the stylized representation fully retains the properties of the base model. When it comes to styling, we can modify the position as well as the scaling and rotation of corresponding Gaussian Splats. Hence, It is possible to achieve the effect of “detaching” the Gaussians from the reconstructed object. An interesting example is the stylized object using the text prompt `Fur`, see Fig. 1, Fig. 3, which can result in the effect of fluffy fur.
>
> Another interesting prompt is `cubism`, which can be seen in supplementary materials under /supplementary\_material/4D/examples/jumpingjacks\_cubism.mp4.
> We can see that the character's face has become more angular, typical to the style, and is obtained by changing the geometry of the head (face and hair).
>
> We will add visual examples to the manuscript.
>
> In our work, we focused on hyperparameters related to the base model, but in future work, it is possible to consider a loss responsible for controlling, among other things, the position of the gaussians, which would control the degree of freedom the splats are allowed to move in space; this would allow for greater control over stylization using prompts such as `Fire`.
>
> **Q2 \[Failure cases\]:**
>
> It is difficult to say definitively that “the method can better imitate styles given by the text prompts (in comparison to reference images)”. We believe it strongly depends on the prompt/image and object/scene. For example, in our opinion the “mosaic” effect on the bonsai scene works better when it is conditioned by the selected image (Fig. 21\) than when it is conditioned by (general) text “Mosaic” (Fig. 20). Nevertheless, an important element is how a given text is represented by CLIP. In general it is interesting how the model represents a given textual/visual concept.
>
> Based on our experience we have identified three common categories of failure cases mostly related to text prompts
>
> 1\. Too general concepts, less known concepts:
>
>  Since we are using CLIP representation, the concept should be well represented by CLIP embedding. We considered 3 prompts:
>
> **TABLE ed80.A**
>
> | object: jumpingjacks | Impressionism |  |Painting by Claude Monet | | Woman with a Parasol – Madame Monet and Her Son | |
> | --- | --- | --- | --- | --- | --- | --- |
> || CLIP-S | CLIP-SIM | CLIP-S | CLIP-SIM | CLIP-S | CLIP-SIM |
> |lambda_bg = 0 |23.04| 18.63 | 17.77 | 12.18 | 18.97 | 10.88 |
> |lambda_bg = 500 |21.51| 16.02 | 18.02 | 13.45 | 18.35 | 10.41 |
> |lambda_bg = 1000 |21.81| 16.59 | 18.20 | 14.27 | 18.52 | 10.65 |
>
>
> CosineSimilarity(prompt1, prompt2) \= 0.802
> CosineSimilarity(prompt1, prompt3) \= 0.531
> CosineSimilarity(prompt2, prompt3) \= 0.382
>
> We observed that for general prompts such as “Impressionism”, the model tends to fill the background, and hollow spaces (e.g. between fingers). This improves the quantitative results, but in our opinion it decreases the visual perception of the stylization. We support this claim using the CLIP-S metric, see the difference between lambda_bg=0 and lambda_bg=500 in Tab. ed80.A and Fig. 26 in the manuscript.
>
> The concepts of “Impressionism” and “Painting by Claude Monet” are well represented by the CLIP model. Additionally, they are closely related, which is visually supported by the models trained using those prompts. On the other hand, concepts such as “Woman with(...)” are heavily influenced by details like an umbrella, which spoils the visual effect in this case. This pattern is shown by the CLIP-SIM metric. Due to the rebuttal rules we are not allowed to include visual comparison in our response, we will add it to the manuscript when possible.
>
> 2\. Length and detail of general concept prompts:
> We conducted an experiment using `jumpingjacks` in which we considered three prompt lengths:
>
> **TABLE ed80.B**
>
> |  | Lotus flower |  | A soft pink lotus flower with delicate petals and a bright yellow center, resting gently on a pond |  | A close-up of a blooming pink lotus flower, its intricate petals radiating around a vivid yellow core, floating gracefully on the water with a gentle reflection beneath and soft green lily pads surrounding it |  |
> | :---- | :---- | :---- | :---- | :---- | :---- | :---- |
> |  | CLIP-S | CLIP- SIM | CLIP-S | CLIP- SIM | CLIP-S | CLIP- SIM |
> | lambda\_bg \= 0 | 24.97 | 22.24 | 23.36 | 19.57 | 24.03 | 21.88 |
> | lambda\_bg \= 500 | 24.85 | 20.78 | 19.46 | 15.87 | 20.34 | 19.15 |
> | lambda\_bg \= 1000 | 24.71 | 20.82 | 19.25 | 15.81 | 20.90 | 19.00 |
>
>
> We noticed that if a shorter prompt is considered, the CLIP model finds a certain representation of a specific word/short prompt related to a general concept. Creating a nice and consistent style, similar to Fig. 2\. If a longer prompt is used, we can expect an averaged representation of the concepts found in the prompt, which may cause some inaccuracies in the stylized image. We will add a visual comparison to the manuscripts.
>
> In both cases (1, 2), the styling emphasizes artifacts, especially in the background. We can mitigate the artifacts in the background using `bg_loss` and `patch_size`. This is particularly evident in detailed areas, e.g. on the hands (see Fig. 26, 27). But, visual assessment is very difficult and subjective.
>
> 3\. Concepts that are difficult to represent, such as `hope`, country names, `random` or `beauty`.
>
>  We will add a comparison and additional experiments to the manuscripts.
>
> **Q3 \[Diversity of reference images or text prompts\]:**
>
> For the user study, we prepared prompts and corresponding images, which we presented in Fig. 19, Fig. 20 (text prompts), Fig. 21, Fig. 22 (image prompts). The number of object-stylization pairs was prepared so that the average time to complete the survey was 10 minutes. A detailed description of the user study can be found in B.2.2. For evaluations based on text prompts, we wanted to avoid introducing visual bias, and we did not attach any reference images. In the context of “Starry Night by Vincent van Gogh” and “Scream by Edvard Munch”. We asked an additional question at the end of the survey: “Were you familiar with the following images before?” Users could answer “Yes” or “No.”
>
> However, in our work, we indicate very different text prompts. In particular, supplementary\_material/4D/examples/trex\_many\_styles.mp4 and supplementary\_material/4D/examples/lego\_many\_styles. mp4, you can find over 26 different prompts; we agree that there are fewer for image conditioning (about 8; see Figs. 6, 8, 10, 12\)
>
> We will add this information to the main paper to improve the clarity of the work.
>
> We thank the reviewer once again for feedback. We strongly believe that the additional experiments have further strengthened the contribution of our work. If there are any remaining questions or points requiring clarification, we will be happy to address them.

---

> > ### Comment · Reviewer_ed8o · 2025-08-06
> >
> > Thanks for the authors effort to provide the rebuttal. The rebuttal addresses my concerns, providing explanations and examples on geometric control (e.g., “fur” and “cubism” styles), systematic failure case analysis, and prompt diversity; with these additions, I remain positive and support acceptance.

---

> > > ### Author Response · Authors · 2025-08-07
> > >
> > > Thank you very much for your response. We are glad to hear that we provided a convincing rebuttal. We also appreciate your support for the acceptance of our work.

---

### Decision · Program_Chairs · 2025-09-17

**Decision:**

Accept (poster)

**Comment:**

After rebuttal, all reviewers have consistent positive opinions. The AC also carefully read all comments and agree with all reviewers for acceptance.